# RUDDER: Return Decomposition for Delayed Rewards

**Jose A. Arjona-Medina**\*   **Michael Gillhofer**\*   **Michael Widrich**\*
**Thomas Unterthiner**   **Johannes Brandstetter**   **Sepp Hochreiter**†

LIT AI Lab
Institute for Machine Learning
Johannes Kepler University Linz, Austria
† also at Institute of Advanced Research in Artificial Intelligence (IARAI)

## Abstract

We propose RUDDER, a novel reinforcement learning approach for delayed rewards in finite Markov decision processes (MDPs). In MDPs the $Q$-values are equal to the expected immediate reward plus the expected future rewards. The latter are related to bias problems in temporal difference (TD) learning and to high variance problems in Monte Carlo (MC) learning. Both problems are even more severe when rewards are delayed. RUDDER aims at making the expected future rewards zero, which simplifies $Q$-value estimation to computing the mean of the immediate reward. We propose the following two new concepts to push the expected future rewards toward zero. (i) Reward redistribution that leads to return-equivalent decision processes with the same optimal policies and, when optimal, zero expected future rewards. (ii) Return decomposition via contribution analysis which transforms the reinforcement learning task into a regression task at which deep learning excels. On artificial tasks with delayed rewards, RUD-DER is significantly faster than MC and exponentially faster than Monte Carlo Tree Search (MCTS), TD($\lambda$), and reward shaping approaches. At Atari games, RUDDER on top of a Proximal Policy Optimization (PPO) baseline improves the scores, which is most prominent at games with delayed rewards. Source code is available at https://github.com/ml-jku/rudder and demonstration videos at https://goo.gl/EQerZV.

## 1 Introduction

Assigning credit for a received reward to past actions is central to reinforcement learning [47]. A great challenge is to learn long-term credit assignment for delayed rewards [23, 20, 18, 33]. Delayed rewards are often episodic or sparse and common in real-world problems [30, 25]. For Markov decision processes (MDPs), the $Q$-value is equal to the expected immediate reward plus the expected future reward. For $Q$-value estimation, the expected future reward leads to biases in temporal difference (TD) and high variance in Monte Carlo (MC) learning. For delayed rewards, TD requires exponentially many updates to correct the bias, where the number of updates is exponential in the number of delay steps. For MC learning, the number of states affected by a delayed reward can grow exponentially with the number of delay steps. (Both statements are proved after Supplements theorems S8 and S10.) An MC estimate of the expected future reward has to average over all possible future trajectories, if rewards, state transitions, or policies are probabilistic. Delayed rewards make an MC estimate much harder.

---

The main goal of our approach is to construct an MDP that has **expected future rewards equal to zero**. If this goal is achieved, $Q$-value estimation simplifies to computing the mean of the immediate rewards. To push the expected future rewards to zero, we require two new concepts. The first new concept is **reward redistribution** to create **return-equivalent MDPs**, which are characterized by having the same optimal policies. An optimal reward redistribution should transform a delayed reward MDP into a return-equivalent MDP with zero expected future rewards. However, expected future rewards equal to zero are in general not possible for MDPs. Therefore, we introduce sequence-Markov decision processes (SDPs), for which reward distributions need not to be Markov. We construct a reward redistribution that leads to a return-equivalent SDP with a second-order Markov reward distribution and expected future rewards that are equal to zero. For these return-equivalent SDPs, $Q$-value estimation simplifies to computing the mean. Nevertheless, the $Q$-values or advantage functions can be used for learning optimal policies. The second new concept is **return decomposition** and its realization via **contribution analysis**. This concept serves to efficiently construct a proper reward redistribution, as described in the next section. Return decomposition transforms a reinforcement learning task into a regression task, where the sequence-wide return must be predicted from the whole state-action sequence. The regression task identifies which state-action pairs contribute to the return prediction and, therefore, receive a redistributed reward. Learning the regression model uses only completed episodes as training set, therefore avoids problems with unknown future state-action trajectories. Even for sub-optimal reward redistributions, we obtain an enormous speed-up of $Q$-value learning if relevant reward-causing state-action pairs are identified. We propose RUDDER (RetUrn Decomposition for DElayed Rewards) for learning with reward redistributions that are obtained via return decomposition.

**To get an intuition** for our approach, assume you repair pocket watches and then sell them. For a particular brand of watch you have to decide whether repairing pays off. The sales price is known, but you have unknown costs, i.e. negative rewards, caused by repair and delivery. The advantage function is the sales price minus the expected immediate repair costs minus the expected future delivery costs. Therefore, you want to know whether the advantage function is positive. — Why is zeroing the expected future costs beneficial? — If the average delivery costs are known, then they can be added to the repair costs resulting in zero future costs. Now you can use your repairing experiences, and you just have to average over the repair costs to know whether repairing pays off. — Why is return decomposition so efficient? — Because of pattern recognition. For zero future costs, you have to estimate the expected brand-related delivery costs, which are e.g. packing costs. These brand-related costs are superimposed by brand-independent general delivery costs for shipment (e.g. time spent for delivery). Assume that general delivery costs are indicated by patterns, e.g. weather conditions, which delay delivery. Using a training set of completed deliveries, supervised learning can identify these patterns and attribute costs to them. This is return decomposition. In this way, only brand-related delivery costs remain and, therefore, can be estimated more efficiently than by MC.

**Related Work.** Our new learning algorithm is gradually changing the reward redistribution during learning, which is known as shaping [44, 47]. In contrast to RUDDER, potential-based shaping like reward shaping [27], look-ahead advice, and look-back advice [50] use a fixed reward redistribution. Moreover, since these methods keep the original reward, the resulting reward redistribution is not optimal, as described in the next section, and learning can still be exponentially slow. A monotonic positive reward transformation [28] changes the reward distribution but is not assured to keep optimal policies. Sibling Rivalry [48] overcomes local optima of distance-to-goal-based reward shaping by changing the original reward, while still finding an optimal policy for the original reward. Disentangled rewards [15] keep optimal policies but are neither environment nor policy specific, therefore cannot have zero expected rewards. Successor features decouple environment and policy from rewards, but changing the reward changes the optimal policies [7, 6]. Temporal Value Transport (TVT) uses an attentional memory mechanism to learn a value function that serves as fictitious reward but optimal policies are not guaranteed to be kept [21]. All these methods do not ensure zero expected future rewards that would speed up learning. Like RUDDER, previous methods have used supervised methods for reinforcement learning [34, 8, 38]. Separate backward models can be learned in a supervised manner to trace back from known goal states [14] or from high reward states [16]. "Hindsight Credit Assignment" (HCA) [17] and "Upside-Down Reinforcement Learning" [39, 45] use supervised learning via cross-entropy to select a backward model that predicts the action to take (output) to achieve a desired return (input 1) from current state (input 2) in a certain number of steps (input 3). For HCA the desired return can be replaced by a desired future state and the number of steps can be relaxed to being achieved until episode end. Instead of backward models,

"backpropagation through a model" [26, 31, 32, 49, 37, 4, 5] uses forward models, which predict the return and generate update signals for a policy by backward analysis via sensitivity analysis. While RUDDER also uses a forward model, it (i) uses contribution analysis instead of sensitivity analysis for backward analysis, and (ii) uses the whole state-action sequence to predict the return.

## 2  Reward Redistribution and Novel Learning Algorithms

Reward redistribution is our main new concept to achieve expected future rewards equal to zero. We start by introducing MDPs, return-equivalent sequence-Markov decision processes (SDPs), and reward redistribution. Furthermore, optimal reward redistribution is defined and novel learning algorithms based on reward redistribution are introduced.

**MDP Definitions and Return-Equivalent Sequence-Markov Decision Processes (SDPs).**  A finite Markov decision process (MDP) $\mathcal{P}$ is 5-tuple $\mathcal{P} = (\mathcal{S}, \mathcal{A}, \mathcal{R}, p, \gamma)$ of finite sets $\mathcal{S}$ of states $s$ (random variable $S_t$ at time $t$), $\mathcal{A}$ of actions $a$ (random variable $A_t$), and $\mathcal{R}$ of rewards $r$ (random variable $R_{t+1}$). Furthermore, $\mathcal{P}$ has transition-reward distributions $p(S_{t+1} = s', R_{t+1} = r \mid S_t = s, A_t = a)$ conditioned on state-actions, and a discount factor $\gamma \in [0, 1]$. The marginals are $p(r \mid s, a) = \sum_{s'} p(s', r \mid s, a)$ and $p(s' \mid s, a) = \sum_r p(s', r \mid s, a)$. The expected reward is $r(s, a) = \sum_r r p(r \mid s, a)$. The return $G_t$ is $G_t = \sum_{k=0}^{\infty} \gamma^k R_{t+k+1}$. For finite horizon MDPs with sequence length $T$ and $\gamma = 1$ the return is $G_t = \sum_{k=0}^{T-t} R_{t+k+1}$. A Markov policy is given as action distribution $\pi(A_t = a \mid S_t = s)$ conditioned on states. We often equip an MDP $\mathcal{P}$ with a policy $\pi$ without explicitly mentioning it. The action-value function $q^\pi(s, a)$ for policy $\pi$ is $q^\pi(s, a) = \mathrm{E}_\pi [G_t \mid S_t = s, A_t = a]$. The goal of learning is to maximize the expected return at time $t = 0$, that is $v_0^\pi = \mathrm{E}_\pi [G_0]$. The optimal policy $\pi^*$ is $\pi^* = \operatorname{argmax}_\pi [v_0^\pi]$. A *sequence-Markov decision process* (SDP) is defined as a decision process which is equipped with a Markov policy and has Markov transition probabilities but a reward that is not required to be Markov. Two SDPs $\tilde{\mathcal{P}}$ and $\mathcal{P}$ are *return-equivalent* if (i) they differ only in their reward distributions and (ii) they have the same expected return at $t = 0$ for each policy $\pi$: $\tilde{v}_0^\pi = v_0^\pi$. They are *strictly return-equivalent* if they have the same expected return for every episode and for each policy $\pi$. Strictly return-equivalent SDPs are return-equivalent. Return-equivalent SDPs have the same optimal policies. For more details see Supplements S2.2.

**Reward Redistribution.**  Strictly return-equivalent SDPs $\tilde{\mathcal{P}}$ and $\mathcal{P}$ can be constructed by reward redistributions. A *reward redistribution* given an SDP $\tilde{\mathcal{P}}$ is a procedure that redistributes for each sequence $s_0, a_0, \ldots, s_T, a_T$ the realization of the sequence-associated return variable $\tilde{G}_0 = \sum_{t=0}^{T} \tilde{R}_{t+1}$ or its expectation along the sequence. Later we will introduce a reward redistribution that depends on the SDP $\tilde{\mathcal{P}}$. The reward redistribution creates a new SDP $\mathcal{P}$ with the redistributed reward $R_{t+1}$ at time $(t+1)$ and the return variable $G_0 = \sum_{t=0}^{T} R_{t+1}$. A reward redistribution is second order Markov if the redistributed reward $R_{t+1}$ depends only on $(s_{t-1}, a_{t-1}, s_t, a_t)$. If the SDP $\mathcal{P}$ is obtained from the SDP $\tilde{\mathcal{P}}$ by reward redistribution, then $\tilde{\mathcal{P}}$ and $\mathcal{P}$ are strictly return-equivalent. The next theorem states that the optimal policies are still the same for $\tilde{\mathcal{P}}$ and $\mathcal{P}$ (proof after Supplements Theorem S2).

**Theorem 1.** *Both the SDP $\tilde{\mathcal{P}}$ with delayed reward $\tilde{R}_{t+1}$ and the SDP $\mathcal{P}$ with redistributed reward $R_{t+1}$ have the same optimal policies.*

**Optimal Reward Redistribution with Expected Future Rewards Equal to Zero.**  We move on to the main goal of this paper: to derive an SDP via reward redistribution that has expected future rewards equal to zero and, therefore, no delayed rewards. At time $(t-1)$ the immediate reward is $R_t$ with expectation $r(s_{t-1}, a_{t-1})$. We define the expected future rewards $\kappa(m, t-1)$ at time $(t-1)$ as the expected sum of future rewards from $R_{t+1}$ to $R_{t+1+m}$.

**Definition 1.** *For $1 \leqslant t \leqslant T$ and $0 \leqslant m \leqslant T - t$, the expected sum of delayed rewards at time $(t-1)$ in the interval $[t+1, t+m+1]$ is defined as $\kappa(m, t-1) = \mathrm{E}_\pi \left[ \sum_{\tau=0}^{m} R_{t+1+\tau} \mid s_{t-1}, a_{t-1} \right].$*

For every time point $t$, the expected future rewards $\kappa(T - t - 1, t)$ given $(s_t, a_t)$ is the expected sum of future rewards until sequence end, that is, in the interval $[t+2, T+1]$. For MDPs, the Bellman equation for Q-values becomes $q^\pi(s_t, a_t) = r(s_t, a_t) + \kappa(T - t - 1, t)$. We aim to derive an MDP

with $\kappa(T-t-1,t) = 0$, which yields $q^\pi(s_t, a_t) = r(s_t, a_t)$. In this case, learning the $Q$-values simplifies to estimating the expected immediate reward $r(s_t, a_t) = \mathrm{E}\left[R_{t+1} \mid s_t, a_t\right]$. Hence, the reinforcement learning task reduces to computing the mean, e.g. the arithmetic mean, for each state-action pair $(s_t, a_t)$. A reward redistribution is defined to be *optimal*, if $\kappa(T-t-1,t) = 0$ for $0 \leqslant t \leqslant T-1$. In general, an optimal reward redistribution violates the Markov assumptions and the Bellman equation does not hold (proof after Supplements Theorem S3). Therefore, we will consider SDPs in the following. The next theorem states that a delayed reward MDP $\tilde{\mathcal{P}}$ with a particular policy $\pi$ can be transformed into a return-equivalent SDP $\mathcal{P}$ with an optimal reward redistribution.

**Theorem 2.** *We assume a delayed reward MDP $\tilde{\mathcal{P}}$, where the accumulated reward is given at sequence end. A new SDP $\mathcal{P}$ is obtained by a second order Markov reward redistribution, which ensures that $\mathcal{P}$ is return-equivalent to $\tilde{\mathcal{P}}$. For a specific $\pi$, the following two statements are equivalent:*
*(I)  $\kappa(T-t-1,t) = 0$, i.e. the reward redistribution is optimal,*

*(II)  $\mathrm{E}\left[R_{t+1} \mid s_{t-1}, a_{t-1}, s_t, a_t\right] = \tilde{q}^\pi(s_t, a_t) - \tilde{q}^\pi(s_{t-1}, a_{t-1})$.* $\qquad\qquad$ (1)

*An optimal reward redistribution fulfills for $1 \leqslant t \leqslant T$ and $0 \leqslant m \leqslant T-t$: $\kappa(m, t-1) = 0$.*

The proof can be found after Supplements Theorem S4. The equation $\kappa(T-t-1,t) = 0$ implies that the new SDP $\mathcal{P}$ has no delayed rewards, that is, $\mathrm{E}_\pi\left[R_{t+1+\tau} \mid s_{t-1}, a_{t-1}\right] = 0$, for $0 \leqslant \tau \leqslant T-t-1$ (Supplements Corollary S1). The SDP $\mathcal{P}$ has no delayed rewards since no state-action pair can increase or decrease the expectation of a future reward. Equation (1) shows that for an optimal reward redistribution the expected reward has to be the difference of consecutive $Q$-values of the original delayed reward. The optimal reward redistribution is second order Markov since the expectation of $R_{t+1}$ at time $(t+1)$ depends on $(s_{t-1}, a_{t-1}, s_t, a_t)$.

The next theorem states the major advantage of an optimal reward redistribution: $\tilde{q}^\pi(s_t, a_t)$ can be estimated with an offset that depends only on $s_t$ by estimating the expected immediate redistributed reward. Thus, $Q$-value estimation becomes trivial and the computation of the advantage function of the MDP $\tilde{\mathcal{P}}$ is simplified.

**Theorem 3.** *If the reward redistribution is optimal, then the $Q$-values of the SDP $\mathcal{P}$ are given by*

$$q^\pi(s_t, a_t) = r(s_t, a_t) = \tilde{q}^\pi(s_t, a_t) - \mathrm{E}_{s_{t-1}, a_{t-1}}\left[\tilde{q}^\pi(s_{t-1}, a_{t-1}) \mid s_t\right] = \tilde{q}^\pi(s_t, a_t) - \psi^\pi(s_t).$$
$\qquad\qquad$ (2)

*The SDP $\mathcal{P}$ and the original MDP $\tilde{\mathcal{P}}$ have the same advantage function.*
*Using a behavior policy $\breve{\pi}$ the expected immediate reward is*

$$\mathrm{E}_{\breve{\pi}}\left[R_{t+1} \mid s_t, a_t\right] = \tilde{q}^\pi(s_t, a_t) - \psi^{\pi, \breve{\pi}}(s_t).$$
$\qquad\qquad$ (3)

The proof can be found after Supplements Theorem S5. If the reward redistribution is not optimal, then $\kappa(T-t-1,t)$ measures the deviation of the $Q$-value from $r(s_t, a_t)$. This theorem justifies several learning methods based on reward redistribution presented in the next paragraph.

**Novel Learning Algorithms Based on Reward Redistributions.** We assume $\gamma = 1$ and a finite horizon or an absorbing state original MDP $\tilde{\mathcal{P}}$ with delayed rewards. For this setting we introduce new reinforcement learning algorithms. They are gradually changing the reward redistribution during learning and are based on the estimations in Theorem 3. These algorithms are also valid for non-optimal reward redistributions, since the optimal policies are kept (Theorem 1). Convergence of RUDDER learning can under standard assumptions be proven by the stochastic approximation for two time-scale update rules [12, 22]. Learning consists of two updates: a reward redistribution network update and a $Q$-value update. Convergence proofs to an optimal policy are difficult, since locally stable attractors may not correspond to optimal policies.

According to Theorem 1, reward redistribution keeps the optimal policies. Therefore, even non-optimal reward redistributions ensure correct learning. However, an optimal reward redistribution speeds up learning considerably. Reward redistribution can be combined with methods that use $Q$-value ranks or advantage functions. We consider (A) $Q$-value estimation, (B) policy gradients, and (C) $Q$-learning. Type (A) methods estimate $Q$-values and are divided into variants (i), (ii), and (iii). Variant (i) assumes an optimal reward redistribution and estimates $\tilde{q}^\pi(s_t, a_t)$ with an offset depending only on $s_t$. The estimates are based on Theorem 3 either by on-policy direct $Q$-value estimation according to Eq. (2) or by off-policy immediate reward estimation according to Eq. (3). Variant (ii) methods assume a non-optimal reward redistribution and correct Eq. (2) by estimating $\kappa$.

Variant (iii) methods use eligibility traces for the redistributed reward. RUDDER learning can be based on policies like "greedy in the limit with infinite exploration" (GLIE) or "restricted rank-based randomized" (RRR) [43]. GLIE policies change toward greediness with respect to the $Q$-values during learning. For more details on these learning approaches see Supplements S2.7.1.

Type (B) methods replace in the expected updates $\mathrm{E}_\pi \left[ \nabla_\theta \log \pi(a \mid s; \boldsymbol{\theta}) q^\pi(s, a) \right]$ of policy gradients the value $q^\pi(s, a)$ by an estimate of $r(s, a)$ or by a sample of the redistributed reward. The offset $\psi^\pi(s)$ in Eq. (2) or $\psi^{\pi, \breve{\pi}}(s)$ in Eq. (3) reduces the variance as baseline normalization does. These methods can be extended to Trust Region Policy Optimization (TRPO) [40] as used in Proximal Policy Optimization (PPO) [42]. The type (C) method is $Q$-learning with the redistributed reward. Here, $Q$-learning is justified if immediate and future reward are drawn together, as typically done.

# 3 Constructing Reward Redistributions by Return Decomposition

We now propose methods to construct reward redistributions. Learning with non-optimal reward redistributions *does work* since the optimal policies do not change according to Theorem 1. However, reward redistributions that are optimal considerably speed up learning, since future expected rewards introduce biases in TD methods and high variances in MC methods. The expected optimal redistributed reward is the difference of $Q$-values according to Eq. (1). The more a reward redistribution deviates from these differences, the larger are the absolute $\kappa$-values and, in turn, the less optimal the reward redistribution gets. Consequently, to construct a reward redistribution which is close to optimal we aim at identifying the largest $Q$-value differences.

**Reinforcement Learning as Pattern Recognition.** We want to transform the reinforcement learning problem into a pattern recognition task to exploit deep learning approaches. The sum of the $Q$-value differences gives the difference between expected return at sequence begin and the expected return at sequence end (telescope sum). Thus, $Q$-value differences allow to predict the expected return of the whole state-action sequence. Identifying and redistributing the reward to the largest $Q$-value differences reduces the prediction error most. $Q$-value differences are assumed to be associated with patterns in state-action transitions. The largest $Q$-value differences are expected to be found more frequently in sequences with very large or very low return. The resulting task is to predict the expected return from the whole sequence and identify which state-action transitions have contributed the most to the prediction. This pattern recognition task serves to construct a reward redistribution, where the redistributed reward corresponds to the different contributions. The next paragraph shows how the return is decomposed and redistributed along the state-action sequence.

**Return Decomposition.** The *return decomposition* idea is that a function $g$ predicts the expectation of the return for a given state-action sequence (return for the whole sequence). The function $g$ is neither a value nor an action-value function since it predicts the expected return when the whole sequence is given. With the help of $g$ either the predicted value or the realization of the return is redistributed over the sequence. A state-action pair receives as redistributed reward its contribution to the prediction, which is determined by contribution analysis. We use contribution analysis since sensitivity analysis has serious drawbacks, e.g. local minima, instabilities, exploding or vanishing gradients, and proper exploration [19, 36]. The biggest drawback is that the relevance of actions is missed since sensitivity analysis does not consider the contribution of actions to the output, but only their effect on the output when slightly perturbing them. Contribution analysis determines how much a state-action pair contributes to the final prediction. We can use any contribution analysis method, but we specifically consider three methods: (A) differences of return predictions, (B) integrated gradients (IG) [46], and (C) layer-wise relevance propagation (LRP) [3]. For (A), $g$ must try to predict the sequence-wide return at every time step. The redistributed reward is given by the difference of consecutive predictions. The function $g$ can be decomposed into past, immediate, and future contributions to the return. Consecutive predictions share the same past and the same future contributions except for two immediate state-action pairs. Thus, in the difference of consecutive predictions contributions cancel except for the two immediate state-action pairs. Even for imprecise predictions of future contributions to the return, contribution analysis is more precise, since prediction errors cancel out. Methods (B) and (C) rely on information later in the sequence for determining the contribution and thereby may introduce a non-Markov reward. The reward can be viewed to be probabilistic but is prone to have high variance. Therefore, we prefer method (A).

**Explaining Away Problem.** We still have to tackle the problem that reward causing actions might not receive redistributed rewards since they are explained away by later states. To describe the problem, assume an MDP $\tilde{\mathcal{P}}$ with the only reward at sequence end. To ensure the Markov property, states in $\tilde{\mathcal{P}}$ have to store the reward contributions of previous state-actions; e.g. $s_T$ has to store all previous contributions such that the expectation $\tilde{r}(s_T, a_T)$ is Markov. The explaining away problem is that later states are used for return prediction, while reward causing earlier actions are missed. To avoid explaining away, we define a difference function $\Delta(s_{t-1}, a_{t-1}, s_t, a_t)$ between a state-action pair $(s_t, a_t)$ and its predecessor $(s_{t-1}, a_{t-1})$. That $\Delta$ is a function of $(s_t, a_t, s_{t-1}, a_{t-1})$ is justified by Eq. (1), which ensures that such $\Delta$s allow an optimal reward redistribution. The sequence of differences is $\Delta_{0:T} := \big(\Delta(s_{-1}, a_{-1}, s_0, a_0), \ldots, \Delta(s_{T-1}, a_{T-1}, s_T, a_T)\big)$. The components $\Delta$ are assumed to be statistically independent from each other, therefore $\Delta$ cannot store reward contributions of previous $\Delta$. The function $g$ should predict the return by $g(\Delta_{0:T}) = \tilde{r}(s_T, a_T)$ and can be decomposed into $g(\Delta_{0:T}) = \sum_{t=0}^{T} h_t$. The contributions are $h_t = h(\Delta(s_{t-1}, a_{t-1}, s_t, a_t))$ for $0 \leqslant t \leqslant T$. For the redistributed rewards $R_{t+1}$, we ensure $\mathrm{E}\left[R_{t+1} \mid s_{t-1}, a_{t-1}, s_t, a_t\right] = h_t$. The reward $\tilde{R}_{T+1}$ of $\tilde{\mathcal{P}}$ is probabilistic and the function $g$ might not be perfect, therefore neither $g(\Delta_{0:T}) = \tilde{r}_{T+1}$ for the return realization $\tilde{r}_{T+1}$ nor $g(\Delta_{0:T}) = \tilde{r}(s_T, a_T)$ for the expected return holds. Therefore, we need to introduce the compensation $\tilde{r}_{T+1} - \sum_{\tau=0}^{T} h(\Delta(s_{\tau-1}, a_{\tau-1}, s_\tau, a_\tau))$ as an extra reward $R_{T+2}$ at time $T + 2$ to ensure strictly return-equivalent SDPs. If $g$ was perfect, then it would predict the expected return which could be redistributed. The new redistributed rewards $R_{t+1}$ are based on the return decomposition, since they must have the contributions $h_t$ as mean:

$$\mathrm{E}\left[R_1 \mid s_0, a_0\right] \;=\; h_0\,, \quad R_{T+2} \;=\; \tilde{R}_{T+1} - \sum_{t=0}^{T} h_t\,, \tag{4}$$

$$\mathrm{E}\left[R_{t+1} \mid s_{t-1}, a_{t-1}, s_t, a_t\right] \;=\; h_t\,, \;\; 0 < t \leqslant T\,, \tag{5}$$

where the realization $\tilde{r}_{T+1}$ is replaced by its random variable $\tilde{R}_{T+1}$. If the prediction of $g$ is perfect, then we can redistribute the expected return via the prediction. Theorem 2 holds also for the correction $R_{T+2}$ (see Supplements Theorem S6). A $g$ with zero prediction errors results in an optimal reward redistribution. Small prediction errors lead to reward redistributions close to an optimal one.

**RUDDER: Return Decomposition using LSTM.** RUDDER uses a Long Short-Term Memory (LSTM) network for return decomposition and the resulting reward redistribution. RUDDER consists of three phases. **(I) Safe exploration.** Exploration sequences should generate LSTM training samples with delayed rewards by avoiding low $Q$-values during a particular time interval. Low $Q$-values hint at states where the agent gets stuck. Parameters comprise starting time, length, and $Q$-value threshold. **(II) Lessons replay buffer for training the LSTM.** If RUDDER's safe exploration discovers an episode with unseen delayed rewards, it is secured in a lessons replay buffer [24]. Unexpected rewards are indicated by a large prediction error of the LSTM. For LSTM training, episodes with larger errors are sampled more often from the buffer, similar to prioritized experience replay [35]. **(III) LSTM and return decomposition.** An LSTM learns to predict sequence-wide return at every time step and, thereafter, return decomposition uses differences of return predictions (contribution analysis method (A)) to construct a reward redistribution. For more details see [1].

**Feedforward Neural Networks (FFNs) vs. LSTMs.** In contrast to LSTMs, FNNs are not suited for processing sequences. Nevertheless, FNNs can learn a action-value function, which enables contribution analysis by differences of predictions. However, this leads to serious problems by spurious contributions that hinder learning. For example, any contributions would be incorrect if the true expectation of the return did not change. Therefore, prediction errors might falsely cause contributions leading to spurious rewards. FNNs are prone to such prediction errors since they have to predict the expected return again and again from each different state-action pair and cannot use stored information. In contrast, the LSTM is less prone to produce spurious rewards: (i) The LSTM will only learn to store information if a state-action pair has a strong evidence for a change in the expected return. In this way, key events can be stored. If information is stored, then internal states and, therefore, also the predictions change, otherwise the predictions stay unchanged. Hence, storing events receives a contribution and a corresponding reward, while by default nothing is stored and no contribution is given. (ii) The LSTM tends to have smaller prediction errors since it can reuse past information for predicting the expected return. (iii) Prediction errors of LSTMs are much more likely

to cancel via prediction differences than those of FNNs. Since consecutive predictions of LSTMs rely on the same internal states, they usually have highly correlated errors.

**Human Expert Episodes.** They are an alternative to exploration and can serve to fill the lessons replay buffer. Learning can be sped up considerably when the LSTM identifies human key actions. RUDDER will reward human key actions even for episodes with low return since other actions that thwart high returns receive negative reward. Using human demonstrations in reinforcement learning led to a huge improvement on some Atari games like Montezuma's Revenge [29, 2].

**Limitations.** RUDDER might not be effective for tasks without delayed rewards, since LSTM learning takes extra time and and struggles with extremely long sequences. Moreover, reward redistribution may introduce disturbing spurious reward signals.

# 4 Experiments

RUDDER is evaluated on three artificial tasks with delayed rewards. These tasks are designed to show problems of TD, MC, and potential-based reward shaping. RUDDER overcomes these problems. Next, we demonstrate that RUDDER also works for more complex tasks with delayed rewards. Therefore, we compare RUDDER with a Proximal Policy Optimization (PPO) baseline on 52 Atari games. All experiments use finite time horizon or absorbing states MDPs with $\gamma = 1$ and reward at episode end. For more information see Supplements S4.1.2.

**Artificial Tasks (I)–(III).** Task (I) shows that TD methods have problems with vanishing information for delayed rewards. The goal is to learn that a delayed reward is larger than a distracting immediate reward. Therefore, the correct expected future reward must be assigned to many state-action pairs. Task (II) is a variation of the introductory pocket watch example with delayed rewards. It shows that MC methods have problems with the high variance of future unrelated rewards. The expected future reward that is caused by the first action has to be estimated. Large future rewards that are not associated with the first action impede MC estimations. Task (III) shows that potential-based reward shaping methods have problems with delayed rewards. For this task, only the first two actions are relevant, to which the delayed reward has to be propagated back.

The tasks have different delays, are tabular ($Q$-table), and use an $\epsilon$-greedy policy with $\epsilon = 0.2$. We compare RUDDER, MC, and TD($\lambda$) on all tasks, and Monte Carlo Tree Search (MCTS) on task (I). Additionally, on task (III), SARSA($\lambda$) and reward shaping are compared. We use $\lambda = 0.9$ as suggested previously [47]. Reward shaping is either the original method, the look-forward advice, or the look-back advice all with three different potential functions. RUDDER uses an LSTM net without output and forget gates, no lessons buffer, and no safe exploration. Contribution analysis is performed with differences of return predictions. A $Q$-table is learned by an exponential moving average of the redistributed reward (RUDDER's $Q$-value estimation) or by $Q$-learning. The performance is measured by the learning time to achieve 90% of the maximal expected return. A Wilcoxon signed-rank indicates a significant performance difference between RUDDER and other methods.

**(I) Grid World** shows problems of TD methods with delayed rewards. The task illustrates a time bomb that explodes at episode end. The agent has to defuse the bomb and then run away as far as possible since defusing fails with a certain probability. Alternatively, the agent can immediately run away, which, however, leads to less reward on average. The Grid World is a $31 \times 31$ grid with *bomb* at coordinate $[30, 15]$ and *start* at $[30 - d, 15]$, where $d$ is the delay of the task. The agent can move *up*, *down*, *left*, and *right* as long as it stays on the grid. At the end of the episode, after $\lfloor 1.5d \rfloor$ steps, the agent receives a reward of 1000 with probability of 0.5, if it has visited *bomb*. At each time step, the agent receives an immediate reward of $c \cdot t \cdot h$, where $c$ depends on the chosen action, $t$ is the current time step, and $h$ is the Hamming distance to *bomb*. Each move toward the *bomb*, is immediately penalized with $c = -0.09$. Each move away from the *bomb*, is immediately rewarded with $c = 0.1$. The agent must learn the $Q$-values precisely to recognize that directly running away is not optimal. Figure 1(I) shows the learning times to solve the task vs. the delay of the reward averaged over 100 trials. For all delays, RUDDER is significantly faster than all other methods with $p$-values $< 10^{-12}$. Speed-ups vs. MC and MCTS, suggest to be exponential with delay time. RUDDER is exponentially faster with increasing delay than $Q(\lambda)$, supporting Supplements Theorem S8. **RUDDER significantly outperforms all other methods.**

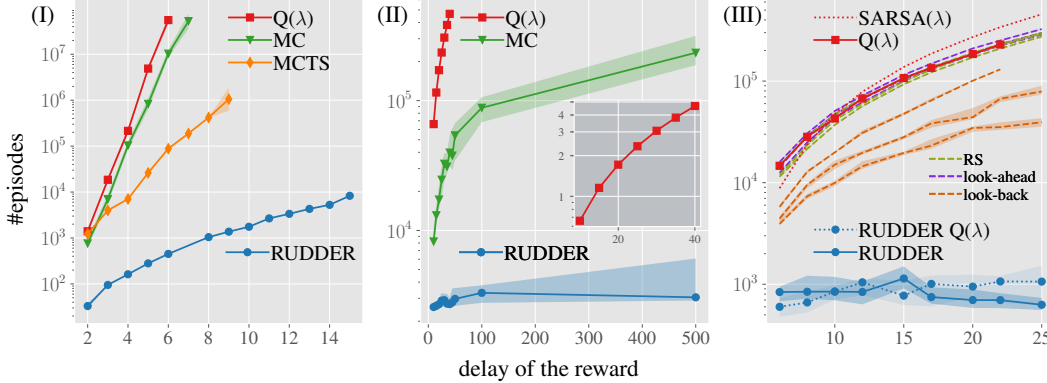

Figure 1: Comparison of RUDDER and other methods on artificial tasks with respect to the learning time in episodes (median of 100 trials) vs. the delay of the reward. The shadow bands indicate the $40\%$ and $60\%$ quantiles. In (II), the y-axis of the inlet is scaled by $10^5$. In (III), reward shaping (RS), look-ahead advice (look-ahead), and look-back advice (look-back) use three different potential functions. In (III), the dashed blue line represents RUDDER with $Q(\lambda)$, in contrast to RUDDER with $Q$-estimation. In all tasks, RUDDER significantly outperforms all other methods.

**(II) The Choice** shows problems of MC methods with delayed rewards. This task has probabilistic state transitions, which can be represented as a tree with states as nodes. The agent traverses the tree from the root (initial state) to the leafs (final states). At the root, the agent has to choose between the left and the right subtree, where one subtree has a higher expected reward. Thereafter, it traverses the tree randomly according to the transition probabilities. Each visited node adds its fixed share to the final reward. The delayed reward is given as accumulated shares at a leaf. The task is solved when the agent always chooses the subtree with higher expected reward. Figure 1(II) shows the learning times to solve the task vs. the delay of the reward averaged over 100 trials. For all delays, RUDDER is significantly faster than all other methods with $p$-values $< 10^{-8}$. The speed-up vs. MC, suggests to be exponential with delay time. RUDDER is exponentially faster with increasing delay than $Q(\lambda)$, supporting Supplements Theorem S8. **RUDDER significantly outperforms all other methods.**

**(III) Trace-Back** shows problems of potential-based reward shaping methods with delayed rewards. We investigate how fast information about delayed rewards is propagated back by RUDDER, $Q(\lambda)$, SARSA($\lambda$), and potential-based reward shaping. MC is skipped since it does not transfer back information. The agent can move in a $15 \times 15$ grid to the 4 adjacent positions as long as it remains on the grid. Starting at $(7, 7)$, the number of moves per episode is $T = 20$. The optimal policy moves the agent up in $t = 1$ and right in $t = 2$, which gives immediate reward of $-50$ at $t = 2$, and a delayed reward of 150 at the end $t = 20 = T$. Therefore, the optimal return is 100. For any other policy, the agent receives only an immediate reward of 50 at $t = 2$. For $t \leqslant 2$, state transitions are deterministic, while for $t > 2$ they are uniformly distributed and independent of the actions. Thus, the return does not depend on actions at $t > 2$. We compare RUDDER, original reward shaping, look-ahead advice, and look-back advice. As suggested by the authors, we use SARSA instead of $Q$-learning for look-back advice. We use three different potential functions for reward shaping, which are all based on the reward redistribution (see Supplements). At $t = 2$, there is a distraction since the immediate reward is $-50$ for the optimal and 50 for other actions. RUDDER is significantly faster than all other methods with $p$-values $< 10^{-17}$. Figure 1(III) shows the learning times averaged over 100 trials. **RUDDER is exponentially faster than all other methods and significantly outperforms them.**

**Atari Games.** RUDDER is evaluated with respect to its learning time and achieved scores on Atari games of the Arcade Learning Environment (ALE) [9] and OpenAI Gym [13]. RUDDER is used on top of the TRPO-based [40] policy gradient method PPO that uses GAE [41]. Our PPO baseline differs from the original PPO baseline [42] in two aspects. (i) Instead of using the sign function of the rewards, rewards are scaled by their current maximum. In this way, the ratio between different rewards remains unchanged and the characteristics of games with large delayed rewards can be recognized. (ii) The safe exploration strategy of RUDDER is used. The entropy coefficient is replaced by Proportional Control [11, 10]. A coarse hyperparameter optimization is performed for the PPO baseline. For all 52 Atari games, RUDDER uses the same architectures, losses, and hyperparameters,

|         | RUDDER | baseline | delay | delay-event |
|---------|--------|----------|-------|-------------|
| Bowling | 192    | 56       | 200   | strike pins |
| Solaris | 1,827  | 616      | 122   | navigate map |
| Venture | 1,350  | 820      | 150   | find treasure |
| Seaquest | 4,770 | 1,616    | 272   | collect divers |

Table 1: Average scores over 3 random seeds with 10 trials each for delayed reward Atari games. "delay": frames between reward and first related action. RUDDER considerably improves the PPO baseline on delayed reward games.

which were optimized for the baseline. The only difference to the PPO baseline is that the policy network predicts the value function of the redistributed reward to utilize reward redistribution for PPO. Contribution analysis uses an LSTM network with differences of return predictions. Here $\Delta$ is the pixel-wise difference of two consecutive frames augmented with the current frame. LSTM training and reward redistribution are restricted to sequence chunks of 500 frames.

Policies are trained with no-op starting condition for 200M game frames using every 4th frame. Training episodes end with losing a life or at maximal 108K frames. All scores are averaged over 3 different random seeds for the network and the ALE initialization. We assess the performance by the learning time and the achieved scores. First, we compare RUDDER to the baseline by average scores per game throughout training, to assess learning speed [42]. For 32 (20) games RUDDER (baseline) learns on average faster. Next, we compare the average scores of the last 10 training games. For 29 (23) games RUDDER (baseline) has higher average scores. In the majority of games RUDDER, improves the scores of the PPO baseline. To compare RUDDER and the baseline on Atari games that are characterized by delayed rewards, we select the games Bowling, Solaris, Venture, and Seaquest. In these games, high scores are achieved by learning the delayed rewards, while learning the immediate rewards and extensive exploration (like for Montezuma's revenge) is less important. The results are presented in Table 1. For more details and further results see Supplements S4.2. Figure 2 displays how RUDDER redistributes rewards to key events in Bowling. **At delayed reward Atari games, RUDDER considerably increases the scores compared to the PPO baseline.**

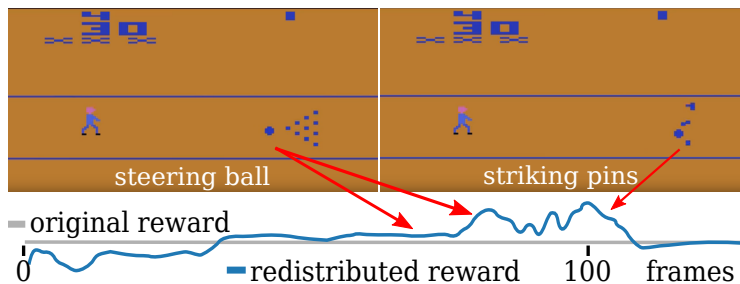

Figure 2: RUDDER redistributes rewards to key events in the Atari game Bowling. Originally, rewards are delayed and only given at episode end. The first 120 out of 200 frames of the episode are shown. RUDDER identifies key actions that steer the ball to hit all pins.

**Conclusion.** We have introduced RUDDER, a novel reinforcement learning algorithm based on the new concepts of reward redistribution and return decomposition via contribution analysis. On artificial tasks, RUDDER significantly outperforms TD($\lambda$), MC, MCTS, and reward shaping methods. On Atari games, RUDDER on average improves a PPO baseline, with the most prominent improvement on long delayed reward games.

**Acknowledgments**

This work was supported by NVIDIA Corporation, Merck KGaA, Audi.JKU Deep Learning Center, Audi Electronic Venture GmbH, Janssen Pharmaceutica (madeSMART), TGW Logistics Group, ZF Friedrichshafen AG, UCB S.A., FFG grant 871302, LIT grant DeepToxGen, and AI-SNN.

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
