[Supplementary Material]

# SUPPLEMENTS to the manuscript
## "RUDDER: Return Decomposition for Delayed Rewards"

### Abstract

We present supplementary material for the paper "RUDDER: Return Decomposition for Delayed Rewards". We provide proofs for the theorems and statements in the paper. We give more details on the new concepts of return decomposition, reward redistribution, and optimal reward redistribution. The experiments are described in more detail and completed with additional experiments. A bias-variance analysis of temporal difference and Monte Carlo learning is given. The exponentially slow correction of the bias of TD in the number of delay steps is proved. That for MC a delayed reward can affect exponentially many variances of other estimation is proved. The reproducibility checklist is included at the end.

# Contents

## S1 Definition of Finite Markov Decision Processes

We consider a finite Markov decision process (MDP) $\mathcal{P}$, which is a 5-tuple $\mathcal{P} = (\mathcal{S}, \mathcal{A}, \mathcal{R}, p, \gamma)$:

- $\mathcal{S}$ is a finite set of states; $S_t$ is the random variable for states at time $t$ with value $s \in \mathcal{S}$. $S_t$ has a discrete probability distribution.

- $\mathcal{A}$ is a finite set of actions (sometimes state-dependent $\mathcal{A}(s)$); $A_t$ is the random variable for actions at time $t$ with value $a \in \mathcal{A}$. $A_t$ has a discrete probability distribution.

- $\mathcal{R}$ is a finite set of rewards; $R_{t+1}$ is the random variable for rewards at time $(t+1)$ with value $r \in \mathcal{R}$. $R_t$ has a discrete probability distribution.

- $p(S_{t+1} = s', R_{t+1} = r \mid S_t = s, A_t = a)$ are the transition and reward distributions over states and rewards, respectively, conditioned on state-actions,

- $\gamma \in [0, 1]$ is a discount factor for the reward.

The Markov policy $\pi$ is a distribution over actions given the state: $\pi(A_t = a \mid S_t = s)$. We often equip an MDP $\mathcal{P}$ with a policy $\pi$ without explicitly mentioning it. At time $t$, the random variables give the states, actions, and rewards of the MDP, while low-case letters give possible values. At each time $t$, the environment is in some state $s_t \in \mathcal{S}$. The policy $\pi$ takes an action $a_t \in \mathcal{A}$, which causes a transition of the environment to state $s_{t+1}$ and a reward $r_{t+1}$ for the policy. Therefore, the MDP creates a sequence

$$(S_0, A_0, R_1, S_1, A_1, R_2, S_2, A_2, R_3, \ldots) \ . \tag{S1}$$

The marginal probabilities for

$$p(s', r \mid s, a) \;=\; \Pr\left[S_{t+1} = s', R_{t+1} = r \mid S_t = s, A_t = a\right] \tag{S2}$$

are:

$$p(r \mid s, a) \;=\; \Pr\left[R_{t+1} = r \mid S_t = s, A_t = a\right] \;=\; \sum_{s'} p(s', r \mid s, a) \ , \tag{S3}$$

$$p(s' \mid s, a) \;=\; \Pr\left[S_{t+1} = s' \mid S_t = s, A_t = a\right] \;=\; \sum_{r} p(s', r \mid s, a) \ . \tag{S4}$$

We use a sum convention: $\sum_{a,b}$ goes over all possible values of $a$ and $b$, that is, all combinations which fulfill the constraints on $a$ and $b$. If $b$ is a function of $a$ (fully determined by $a$), then $\sum_{a,b} = \sum_a$.
We denote expectations:

- $\mathrm{E}_\pi$ is the expectation where the random variable is an MDP sequence of states, actions, and rewards generated with policy $\pi$.

- $\mathrm{E}_s$ is the expectation where the random variable is $S_t$ with values $s \in \mathcal{S}$.

- $\mathrm{E}_a$ is the expectation where the random variable is $A_t$ with values $a \in \mathcal{A}$.

- $\mathrm{E}_r$ is the expectation where the random variable is $R_{t+1}$ with values $r \in \mathcal{R}$.

- $\mathrm{E}_{s,a,r,s',a'}$ is the expectation where the random variables are $S_{t+1}$ with values $s' \in \mathcal{S}$, $S_t$ with values $s \in \mathcal{S}$, $A_t$ with values $a \in \mathcal{A}$, $A_{t+1}$ with values $a' \in \mathcal{A}$, and $R_{t+1}$ with values $r \in \mathcal{R}$. If more or fewer random variables are used, the notation is consistently adapted.

The return $G_t$ is the accumulated reward starting from $t + 1$:

$$G_t \;=\; \sum_{k=0}^{\infty} \gamma^k \, R_{t+k+1} \ . \tag{S5}$$

The discount factor $\gamma$ determines how much immediate rewards are favored over more delayed rewards. For $\gamma = 0$ the return (the objective) is determined as the largest expected immediate reward, while for $\gamma = 1$ the return is determined by the expected sum of future rewards if the sum exists.

**State-Value and Action-Value Function.** The state-value function $v^\pi(s)$ for policy $\pi$ and state $s$ is defined as

$$v^\pi(s) \;=\; \mathrm{E}_\pi\left[G_t \mid S_t = s\right] \;=\; \mathrm{E}_\pi\left[\sum_{k=0}^{\infty} \gamma^k \, R_{t+k+1} \mid S_t = s\right] \ . \tag{S6}$$

Starting at $t = 0$:

$$v_0^\pi = \mathrm{E}_\pi \left[ \sum_{t=0}^\infty \gamma^t R_{t+1} \right] = \mathrm{E}_\pi \left[ G_0 \right] , \tag{S7}$$

the optimal state-value function $v_*$ and policy $\pi_*$ are

$$v_*(s) = \max_\pi v^\pi(s) , \tag{S8}$$

$$\pi_* = \arg\max_\pi v^\pi(s) \text{ for all } s . \tag{S9}$$

The action-value function $q^\pi(s, a)$ for policy $\pi$ is the expected return when starting from $S_t = s$, taking action $A_t = a$, and following policy $\pi$:

$$q^\pi(s, a) = \mathrm{E}_\pi \left[ G_t \mid S_t = s, A_t = a \right] = \mathrm{E}_\pi \left[ \sum_{k=0}^\infty \gamma^k R_{t+k+1} \mid S_t = s, A_t = a \right] . \tag{S10}$$

The optimal action-value function $q_*$ and policy $\pi_*$ are

$$q_*(s, a) = \max_\pi q^\pi(s, a) , \tag{S11}$$

$$\pi_* = \arg\max_\pi q^\pi(s, a) \text{ for all } (s, a) . \tag{S12}$$

The optimal action-value function $q_*$ can be expressed via the optimal value function $v_*$:

$$q_*(s, a) = \mathrm{E} \left[ R_{t+1} + \gamma v_*(S_{t+1}) \mid S_t = s, A_t = a \right] . \tag{S13}$$

The optimal state-value function $v_*$ can be expressed via the optimal action-value function $q_*$ using the optimal policy $\pi_*$:

$$v_*(s) = \max_a q^{\pi_*}(s, a) = \max_a \mathrm{E}_{\pi_*} \left[ G_t \mid S_t = s, A_t = a \right] = \tag{S14}$$

$$\max_a \mathrm{E}_{\pi_*} \left[ R_{t+1} + \gamma G_{t+1} \mid S_t = s, A_t = a \right] =$$

$$\max_a \mathrm{E} \left[ R_{t+1} + \gamma v_*(S_{t+1}) \mid S_t = s, A_t = a \right] .$$

**Finite time horizon and no discount.** We consider a **finite** time horizon, that is, we consider only episodes of length $T$, but may receive reward $R_{T+1}$ at episode end at time $T + 1$. The finite time horizon MDP creates a sequence

$$(S_0, A_0, R_1, S_1, A_1, R_2, S_2, A_2, R_3, \ldots, S_{T-1}, A_{T-1}, R_T, S_T, A_T, R_{T+1}) . \tag{S15}$$

Furthermore, we do not discount future rewards, that is, we set $\gamma = 1$. The return $G_t$ from time $t$ to $T$ is the sum of rewards:

$$G_t = \sum_{k=0}^{T-t} R_{t+k+1} . \tag{S16}$$

The state-value function $v$ for policy $\pi$ is

$$v^\pi(s) = \mathrm{E}_\pi \left[ G_t \mid S_t = s \right] = \mathrm{E}_\pi \left[ \sum_{k=0}^{T-t} R_{t+k+1} \mid S_t = s \right] \tag{S17}$$

and the action-value function $q$ for policy $\pi$ is

$$q^\pi(s, a) = \mathrm{E}_\pi \left[ G_t \mid S_t = s, A_t = a \right] = \mathrm{E}_\pi \left[ \sum_{k=0}^{T-t} R_{t+k+1} \mid S_t = s, A_t = a \right] \tag{S18}$$

$$= \mathrm{E}_\pi \left[ R_{t+1} + G_{t+1} \mid S_t = s, A_t = a \right]$$

$$= \sum_{s',r} p(s', r \mid s, a) \left[ r + \sum_{a'} \pi(a' \mid s') q^\pi(s', a') \right] .$$

From the Bellman equation Eq. (S18), we obtain:

$$\sum_{s'} p(s' \mid s, a) \sum_{a'} \pi(a' \mid s') \, q^{\pi}(s', a') \;=\; q^{\pi}(s, a) \;-\; \sum_{r} r \, p(r \mid s, a) \,, \qquad \text{(S19)}$$

$$\mathrm{E}_{s', a'} \left[ q^{\pi}(s', a') \mid s, a \right] \;=\; q^{\pi}(s, a) \;-\; r(s, a) \,. \qquad \text{(S20)}$$

The expected return at time $t = 0$ for policy $\pi$ is

$$v_0^{\pi} \;=\; \mathrm{E}_{\pi} \left[ G_0 \right] \;=\; \mathrm{E}_{\pi} \left[ \sum_{t=0}^{T} R_{t+1} \right] \,, \qquad \text{(S21)}$$

$$\pi^* \;=\; \underset{\pi}{\mathrm{argmax}} \; v_0^{\pi} \,.$$

The agent may start in a particular starting state $S_0$ which is a random variable. Often $S_0$ has only one value $s_0$.

**Learning.** The **goal** of learning is to find the policy $\pi^*$ that maximizes the expected future discounted reward (the return) if starting at $t = 0$. Thus, the optimal policy $\pi^*$ is

$$\pi^* \;=\; \underset{\pi}{\mathrm{argmax}} \; v_0^{\pi} \,. \qquad \text{(S22)}$$

We consider two learning approaches for $Q$-values: Monte Carlo and temporal difference.

**Monte Carlo (MC).** To estimate $q^{\pi}(s, a)$, MC computes the arithmetic mean of all observed returns $(G_t \mid S_t = s, A_t = a)$ in the data. When using Monte Carlo for learning a policy we use an exponentially weighted arithmetic mean since the policy steadily changes.
For the $i$th update Monte Carlo tries to minimize $\frac{1}{2} M(s_t, a_t)^2$ with the residual $M(s_t, a_t)$

$$M(s_t, a_t) \;=\; (q^{\pi})^i(s_t, a_t) \;-\; \sum_{\tau=0}^{T-t-1} \gamma^{\tau} r_{t+1+\tau} \,, \qquad \text{(S23)}$$

such that the update of the action-value $q$ at state-action $(s_t, a_t)$ is

$$(q^{\pi})^{i+1}(s_t, a_t) \;=\; (q^{\pi})^i(s_t, a_t) \;-\; \alpha \, M(s_t, a_t) \,. \qquad \text{(S24)}$$

This update is called *constant-$\alpha$ MC* [78].

**Temporal difference (TD) methods.** TD updates are based on the Bellman equation. If $r(s, a)$ and $\mathrm{E}_{s', a'} \left[ \hat{q}^{\pi}(s', a') \mid s, a \right]$ have been estimated, the $Q$-values can be updated according to the Bellman equation:

$$(\hat{q}^{\pi})^{\mathrm{new}}(s, a) \;=\; r(s, a) \;+\; \gamma \, \mathrm{E}_{s', a'} \left[ \hat{q}^{\pi}(s', a') \mid s, a \right] \,. \qquad \text{(S25)}$$

The update is applying the Bellman operator with estimates $\mathrm{E}_{s', a'} \left[ \hat{q}^{\pi}(s', a') \mid s, a \right]$ and $r(s, a)$ to $\hat{q}^{\pi}$ to obtain $(\hat{q}^{\pi})^{\mathrm{new}}$. The new estimate $(\hat{q}^{\pi})^{\mathrm{new}}$ is closer to the fixed point $q^{\pi}$ of the Bellman operator, since the Bellman operator is a contraction
Since the estimates $\mathrm{E}_{s', a'} \left[ \hat{q}^{\pi}(s', a') \mid s, a \right]$ and $r(s, a)$ are not known, TD methods try to minimize $\frac{1}{2} B(s, a)^2$ with the Bellman residual $B(s, a)$:

$$B(s, a) \;=\; \hat{q}^{\pi}(s, a) \;-\; r(s, a) \;-\; \gamma \, \mathrm{E}_{s', a'} \left[ \hat{q}^{\pi}(s', a') \right] \,. \qquad \text{(S26)}$$

TD methods use an estimate $\hat{B}(s, a)$ of $B(s, a)$ and a learning rate $\alpha$ to make an update

$$\hat{q}^{\pi}(s, a)^{\mathrm{new}} \;\leftarrow\; \hat{q}^{\pi}(s, a) \;-\; \alpha \, \hat{B}(s, a) \,. \qquad \text{(S27)}$$

For all TD methods $r(s, a)$ is estimated by $R_{t+1}$ and $s'$ by $S_{t+1}$, while $\hat{q}^{\pi}(s', a')$ does not change with the current sample, that is, it is fixed for the estimate. However, the sample determines which $(s', a')$ is chosen. The TD methods differ in how they select $a'$. **SARSA** [63] selects $a'$ by sampling from the policy:

$$\mathrm{E}_{s', a'} \left[ \hat{q}^{\pi}(s', a') \right] \;\approx\; \hat{q}^{\pi}(S_{t+1}, A_{t+1})$$

and **expected SARSA** [30] averages over selections

$$\mathrm{E}_{s', a'} \left[ \hat{q}^{\pi}(s', a') \right] \;\approx\; \sum_{a} \pi(a \mid S_{t+1}) \, \hat{q}^{\pi}(S_{t+1}, a).$$

It is possible to estimate $r(s, a)$ separately via an unbiased minimal variance estimator like the arithmetic mean and then perform TD updates with the Bellman error using the estimated $r(s, a)$ [61]. **Q-learning** [87] is an off-policy TD algorithm which is proved to converge [88, 12]. The proofs were later generalized [29, 82]. $Q$-learning uses

$$\mathrm{E}_{s', a'}\left[\hat{q}^{\pi}(s', a')\right] \approx \max_{a} \hat{q}(S_{t+1}, a) . \tag{S28}$$

The action-value function $q$, which is learned by $Q$-learning, approximates $q_*$ independently of the policy that is followed. More precisely, with $Q$-learning $q$ converges with probability 1 to the optimal $q_*$. However, the policy still determines which state-action pairs are encountered during learning. The convergence only requires that all action-state pairs are visited and updated infinitely often.

## S2    Reward Redistribution, Return-Equivalent SDPs, Novel Learning Algorithms, and Return Decomposition

### S2.1    State Enriched MDPs

For MDPs with a delayed reward the states have to code the reward. However, for an immediate reward the states can be made more compact by removing the reward information. For example, states with memory of a delayed reward can be mapped to states without memory. Therefore, in order to compare MDPs, we introduce the concept of homomorphic MDPs. We first need to define a partition of a set induced by a function. Let $B$ be a partition of a set $X$. For any $x \in X$, we denote $[x]_B$ the block of $B$ to which $x$ belongs. Any function $f$ from a set $X$ to a set $Y$ induces a partition (or equivalence relation) on $X$, with $[x]_f = [x']_f$ if and only if $f(x) = f(x')$. We now can define homomorphic MDPs.

**Definition S1** (Ravindran and Barto [57, 58]). *An MDP homomorphism $h$ from an MDP $\mathcal{P} = (\mathcal{S}, \mathcal{A}, \mathcal{R}, p, \gamma)$ to an MDP $\tilde{\mathcal{P}} = (\tilde{\mathcal{S}}, \tilde{\mathcal{A}}, \tilde{\mathcal{R}}, \tilde{p}, \tilde{\gamma})$ is a a tuple of surjections $(f, g_1, g_2, \ldots, g_n)$ ($n$ is number of states), with $h(s, a) = (f(s), g_s(a))$, where $f : \mathcal{S} \to \tilde{\mathcal{S}}$ and $g_s : \mathcal{A}_s \to \tilde{\mathcal{A}}_{f(s)}$ for $s \in \mathcal{S}$ ($\mathcal{A}_s$ are the admissible actions in state $s$ and $\tilde{\mathcal{A}}_{f(s)}$ are the admissible actions in state $\tilde{s}$). Furthermore, for all $s, s' \in \mathcal{S}, a \in \mathcal{A}_s$:*

$$\tilde{p}(f(s') \mid f(s), g_s(a)) = \sum_{s'' \in [s']_f} p(s'' \mid s, a) , \tag{S29}$$

$$\tilde{p}(\tilde{r} \mid f(s), g_s(a)) = p(r \mid s, a) . \tag{S30}$$

*We use $[s]_f = [s']_f$ if and only if $f(s) = f(s')$.*

We call $\tilde{\mathcal{P}}$ the *homomorphic image* of $\mathcal{P}$ under $h$. For homomorphic images the optimal $Q$-values and the optimal policies are the same.

**Lemma S1** (Ravindran and Barto [57]). *If $\tilde{\mathcal{P}}$ is a homomorphic image of $\mathcal{P}$, then the optimal $Q$-values are the same and a policy that is optimal in $\tilde{\mathcal{P}}$ can be transformed to an optimal policy in $\mathcal{P}$ by normalizing the number of actions $a$ that are mapped to the same action $\tilde{a}$.*

Consequently, the original MDP can be solved by solving a homomorphic image.
Similar results have been obtained by Givan et al. using stochastically bisimilar MDPs: "Any stochastic bisimulation used for aggregation preserves the optimal value and action sequence properties as well as the optimal policies of the model" [18]. Theorem 7 and Corollary 9.1 in Givan et al. show the facts of Lemma S1. Li et al. give an overview over state abstraction and state aggregation for Markov decision processes, which covers homomorphic MDPs [37].
A Markov decision process $\tilde{\mathcal{P}}$ is state-enriched compared to an MDP $\mathcal{P}$ if $\tilde{\mathcal{P}}$ has the same states, actions, transition probabilities, and reward probabilities as $\mathcal{P}$ but with additional information in its states. We define state-enrichment as follows:

**Definition S2.** *A Markov decision process $\tilde{\mathcal{P}}$ is* state-enriched *compared to a Markov decision process $\mathcal{P}$ if $\mathcal{P}$ is a homomorphic image of $\tilde{\mathcal{P}}$, where $g_{\tilde{s}}$ is the identity and $f(\tilde{s}) = s$ is not bijective.*

Being not bijective means that there exist $\tilde{s}'$ and $\tilde{s}''$ with $f(\tilde{s}') = f(\tilde{s}'')$, that is, $\tilde{\mathcal{S}}$ has more elements than $\mathcal{S}$. In particular, state-enrichment does not change the optimal policies nor the $Q$-values in the sense of Lemma S1.

**Proposition S1.** *If an MDP $\tilde{\mathcal{P}}$ is* state-enriched *compared to an MDP $\mathcal{P}$, then both MDPs have the same optimal $Q$-values and the same optimal policies.*

*Proof.* According to the definition $\mathcal{P}$ is a homomorphic image of $\tilde{\mathcal{P}}$. The statements of Proposition S1 follow directly from Lemma S1. □

Optimal policies of the state-enriched MDP $\tilde{\mathcal{P}}$ can be transformed to optimal policies of the original MDP $\mathcal{P}$ and, vice versa, each optimal policy of the original MDP $\mathcal{P}$ corresponds to at least one optimal policy of the state-enriched MDP $\tilde{\mathcal{P}}$.

### S2.2 Return-Equivalent Sequence-Markov Decision Processes (SDPs)

Our goal is to compare Markov decision processes (MDPs) with delayed rewards to decision processes (DPs) without delayed rewards. The DPs without delayed rewards can but need not to be Markov in the rewards. Toward this end, we consider two DPs $\tilde{\mathcal{P}}$ and $\mathcal{P}$ which differ only in their (non-Markov) reward distributions. However for each policy $\pi$ the DPs $\tilde{\mathcal{P}}$ and $\mathcal{P}$ have the same expected return at $t = 0$, that is, $\tilde{v}_0^\pi = v_0^\pi$, or they have the same expected return for every episode.

#### S2.2.1 Sequence-Markov Decision Processes (SDPs)

We first define decision processes that are Markov except for the reward, which is not required to be Markov.

**Definition S3.** *A sequence-Markov decision process (SDP) is defined as a finite decision process which is equipped with a Markov policy and has Markov transition probabilities but a reward distribution that is not required to be Markov.*

**Proposition S2.** *Markov decision processes are sequence-Markov decision processes.*

*Proof.* MDPs have Markov transition probabilities and are equipped with Markov policies. □

**Definition S4.** *We call two sequence-Markov decision processes $\mathcal{P}$ and $\tilde{\mathcal{P}}$ that have the same Markov transition probabilities and are equipped with the same Markov policy* sequence-equivalent.

**Lemma S2.** *Two sequence-Markov decision processes that are sequence-equivalent have the same probability to generate state-action sequences $(s_0, a_0, \ldots, s_t, a_t)$, $0 \leqslant t \leqslant T$.*

*Proof.* Sequence generation only depends on transition probabilities and policy. Therefore the probability of generating a particular sequences is the same for both SDPs. □

Next we define the state-value and action-value function for sequence-Markov decision processes (SDPs). In contrast to MDPs, the state-value and action-value functions for SDPs also depend on the past since the return distributions are not Markov.
We have to redefine the expectation $\mathrm{E}_\pi$:

$$\mathrm{E}_\pi\left[.\mid.\right] = \mathrm{E}_{s_0,a_0,\ldots,s_T,a_T}\left[.\mid.\right]. \tag{S31}$$

The expectation does not include state and actions that enter the condition.
The state-value function $v$ for policy $\pi$ is

$$v^\pi(s) = \mathrm{E}_\pi\left[G_t \mid S_t = s\right] = \mathrm{E}_\pi\left[\sum_{k=0}^{T-t} R_{t+k+1} \mid S_t = s\right] \tag{S32}$$

$$= \mathrm{E}_{s_0,a_0,\ldots,s_{t-1},a_{t-1},a_t,s_{t+1},a_{t+1},\ldots,s_T,a_T}\left[\sum_{k=0}^{T-t} R_{t+k+1} \mid S_t = s\right]$$

and the action-value function $q$ for policy $\pi$ is

$$q^\pi(s,a) = \mathrm{E}_\pi\left[G_t \mid S_t = s, A_t = a\right] = \mathrm{E}_\pi\left[\sum_{k=0}^{T-t} R_{t+k+1} \mid S_t = s, A_t = a\right] \tag{S33}$$

$$= \mathrm{E}_{s_0,a_0,\ldots,s_{t-1},a_{t-1},s_{t+1},a_{t+1},\ldots,s_T,a_T}\left[\sum_{k=0}^{T-t} R_{t+k+1} \mid S_t = s, A_t = a\right].$$

We extend the definitions by including the past in the state-value and action-value functions.

The state-value function $v$ for policy $\pi$ depending on $k$ past state-actions is

$$
\begin{aligned}
v^\pi & (s_t, a_{t-1}, s_{t-1}, \ldots, a_{t-k}, s_{t-k}) \qquad \text{(S34)}\\
&= \mathrm{E}_\pi \left[ G_t \mid S_t = s_t, A_{t-1} = a_{t-1}, S_{t-1} = s_{t-1}, \ldots, A_{t-k} = a_{t-k}, S_{t-k} = s_{t-k} \right]\\
&= \mathrm{E}_\pi \left[ \sum_{k=0}^{T-t} R_{t+k+1} \mid S_t = s_t, A_{t-1} = a_{t-1}, S_{t-1} = s_{t-1}, \ldots, A_{t-k} = a_{t-k}, S_{t-k} = s_{t-k} \right]\\
&= \mathrm{E}_{s_0, a_0, \ldots, s_{t-k-1}, a_{t-k-1}, a_t, s_{t+1}, a_{t+1}, \ldots, s_T, a_T} \left[ \sum_{k=0}^{T-t} R_{t+k+1} \mid \right.\\
& \qquad \left. S_t = s_t, A_{t-1} = a_{t-1}, S_{t-1} = s_{t-1}, \ldots, A_{t-k} = a_{t-k}, S_{t-k} = s_{t-k} \right]
\end{aligned}
$$

and the action-value function $q$ for policy $\pi$ depending on $k$ past state-actions is

$$
\begin{aligned}
q^\pi & (a_t, s_t, a_{t-1}, s_{t-1}, \ldots, a_{t-k}, s_{t-k}) \qquad \text{(S35)}\\
&= \mathrm{E}_\pi \left[ G_t \mid A_t = a_t, S_t = s_t, A_{t-1} = a_{t-1}, S_{t-1} = s_{t-1}, \ldots, A_{t-k} = a_{t-k}, S_{t-k} = s_{t-k} \right]\\
&= \mathrm{E}_\pi \left[ \sum_{k=0}^{T-t} R_{t+k+1} \mid A_t = a_t, S_t = s_t, A_{t-1} = a_{t-1}, S_{t-1} = s_{t-1}, \ldots, A_{t-k} = a_{t-k}, S_{t-k} = s_{t-k} \right]\\
&= \mathrm{E}_{s_0, a_0, \ldots, s_{t-k-1}, a_{t-k-1}, s_{t+1}, a_{t+1}, \ldots, s_T, a_T} \left[ \sum_{k=0}^{T-t} R_{t+k+1} \mid \right.\\
& \qquad \left. A_t = a_t, S_t = s_t, A_{t-1} = a_{t-1}, S_{t-1} = s_{t-1}, \ldots, A_{t-k} = a_{t-k}, S_{t-k} = s_{t-k} \right] .
\end{aligned}
$$

The state-value function $v$ for policy $\pi$ depending on the complete past is

$$
\begin{aligned}
v^\pi (s_t, \ldots, a_0, s_0) &= \mathrm{E}_\pi \left[ G_t \mid S_t = s_t, \ldots, A_0 = a_0, S_0 = s_0 \right] \qquad \text{(S36)}\\
&= \mathrm{E}_\pi \left[ \sum_{k=0}^{T-t} R_{t+k+1} \mid S_t = s_t, \ldots, A_0 = a_0, S_0 = s_0 \right]\\
&= \mathrm{E}_{a_t, \ldots, s_T, a_T} \left[ \sum_{k=0}^{T-t} R_{t+k+1} \mid S_t = s_t, \ldots, A_0 = a_0, S_0 = s_0 \right]
\end{aligned}
$$

and the action-value function $q$ for policy $\pi$ depending on the complete past is

$$
\begin{aligned}
q^\pi & (a_t, s_t, \ldots, a_0, s_0) \qquad \text{(S37)}\\
&= \mathrm{E}_\pi \left[ G_t \mid A_t = a_t, S_t = s_t, \ldots, A_0 = a_0, S_0 = s_0 \right]\\
&= \mathrm{E}_\pi \left[ \sum_{k=0}^{T-t} R_{t+k+1} \mid A_t = a_t, S_t = s_t, \ldots, A_0 = a_0, S_0 = s_0 \right]\\
&= \mathrm{E}_{s_{t+1}, a_{t+1}, \ldots, s_T, a_T} \left[ \sum_{k=0}^{T-t} R_{t+k+1} \mid A_t = a_t, S_t = s_t, \ldots, A_0 = a_0, S_0 = s_0 \right] .
\end{aligned}
$$

**Markov decision processes (MDPs):** the state-value function $v$ and the action-value function $q$ remain the same. The rewards do not depend on the past, therefore the past can be integrated out in the definitions.

The Bellman equation does not hold any longer if the reward depends on the past. For example

$$
\begin{aligned}
q^\pi(s, a) &= \mathrm{E}_\pi \left[ G_t \mid S_t = s, A_t = a \right] = \mathrm{E}_\pi \left[ \sum_{k=0}^{T-t} R_{t+k+1} \mid S_t = s, A_t = a \right] \qquad \text{(S38)}\\
&= \mathrm{E}_\pi \left[ R_{t+1} \mid S_t = s, A_t = a \right] + \mathrm{E}_\pi \left[ \sum_{k=0}^{T-t-1} R_{t+k+2} \mid S_t = s, A_t = a \right] .
\end{aligned}
$$

Since $\mathrm{E}_\pi$ does not average over $a_t, s_t$, the term $\mathrm{E}_\pi \left[ \sum_{k=0}^{T-t-1} R_{t+k+2} \mid S_t = s, A_t = a \right]$ cannot be expressed by $q^\pi(s_{t+1}, a_{t+1})$, which requires an average over $a_t, s_t$. Thus, the Bellman equation does not hold as it is a recursive equation where $q^\pi(s_t, a_t)$ is expressed by $q^\pi(s_{t+1}, a_{t+1})$.

### S2.2.2 Return-Equivalent SDPs

We define return-equivalent SDPs which can be shown to have the same optimal policies.

**Definition S5.** *Two sequence-Markov decision processes $\tilde{\mathcal{P}}$ and $\mathcal{P}$ are* return-equivalent *if they differ only in their reward but for each policy $\pi$ have the same expected return $\tilde{v}_0^\pi = v_0^\pi$. $\tilde{\mathcal{P}}$ and $\mathcal{P}$ are* strictly return-equivalent *if they have the same expected return for every episode and for each policy $\pi$:*

$$\mathrm{E}_\pi \left[ \tilde{G}_0 \mid s_0, a_0, \ldots, s_T, a_T \right] \;=\; \mathrm{E}_\pi \left[ G_0 \mid s_0, a_0, \ldots, s_T, a_T \right] . \tag{S39}$$

The definition of return-equivalence can be generalized to strictly monotonic functions $f$ for which $\tilde{v}_0^\pi = f(v_0^\pi)$. Since strictly monotonic functions do not change the ordering of the returns, maximal returns stay maximal after applying the function $f$.

Strictly return-equivalent SDPs are return-equivalent as the next proposition states.

**Proposition S3.** *Strictly return-equivalent sequence-Markov decision processes are return-equivalent.*

*Proof.* The expected return at $t = 0$ given a policy is the sum of the probability of generating a sequence times the expected reward for this sequence. Both expectations are the same for two strictly return-equivalent sequence-Markov decision processes. Therefore the expected return at time $t = 0$ is the same. $\qquad\square$

The next proposition states that return-equivalent SDPs have the same optimal policies.

**Proposition S4.** *Return-equivalent sequence-Markov decision processes have the same optimal policies.*

*Proof.* The optimal policy is defined as maximizing the expected return at time $t = 0$. For each policy the expected return at time $t = 0$ is the same for return-equivalent decision processes. Consequently, the optimal policies are the same. $\qquad\square$

Two strictly return-equivalent SDPs have the same expected return for each state-action sub-sequence $(s_0, a_0, \ldots, s_t, a_t)$, $0 \leqslant t \leqslant T$.

**Lemma S3.** *Two strictly return-equivalent SDPs $\tilde{\mathcal{P}}$ and $\mathcal{P}$ have the same expected return for each state-action sub-sequence $(s_0, a_0, \ldots, s_t, a_t)$, $0 \leqslant t \leqslant T$:*

$$\mathrm{E}_\pi \left[ \tilde{G}_0 \mid s_0, a_0, \ldots, s_t, a_t \right] \;=\; \mathrm{E}_\pi \left[ G_0 \mid s_0, a_0, \ldots, s_t, a_t \right] . \tag{S40}$$

*Proof.* Since the SDPs are strictly return-equivalent, we have

$$\mathrm{E}_\pi \left[ \tilde{G}_0 \mid s_0, a_0, \ldots, s_t, a_t \right] \tag{S41}$$

$$= \sum_{s_{t+1}, a_{t+1}, \ldots, s_T, a_T} p_\pi(s_{t+1}, a_{t+1}, \ldots, s_T, a_T \mid s_t, a_t) \, \mathrm{E}_\pi \left[ \tilde{G}_0 \mid s_0, a_0, \ldots, s_T, a_T \right]$$

$$= \sum_{s_{t+1}, a_{t+1}, \ldots, s_T, a_T} p_\pi(s_{t+1}, a_{t+1}, \ldots, s_T, a_T \mid s_t, a_t) \, \mathrm{E}_\pi \left[ G_0 \mid s_0, a_0, \ldots, s_T, a_T \right]$$

$$= \mathrm{E}_\pi \left[ G_0 \mid s_0, a_0, \ldots, s_t, a_t \right] .$$

We used the marginalization of the full probability and the Markov property of the state-action sequence. $\qquad\square$

We now give the analog definitions and results for MDPs which are SDPs.

**Definition S6.** *Two Markov decision processes $\tilde{\mathcal{P}}$ and $\mathcal{P}$ are* return-equivalent *if they differ only in $p(\tilde{r} \mid s, a)$ and $p(r \mid s, a)$ but have the same expected return $\tilde{v}_0^\pi = v_0^\pi$ for each policy $\pi$. $\tilde{\mathcal{P}}$ and $\mathcal{P}$ are* strictly return-equivalent *if they have the same expected return for every episode and for each policy $\pi$:*

$$\mathrm{E}_\pi \left[ \tilde{G}_0 \mid s_0, a_0, \ldots, s_T, a_T \right] \;=\; \mathrm{E}_\pi \left[ G_0 \mid s_0, a_0, \ldots, s_T, a_T \right] . \tag{S42}$$

Strictly return-equivalent MDPs are return-equivalent as the next proposition states.

**Proposition S5.** *Strictly return-equivalent decision processes are return-equivalent.*

*Proof.* Since MDPs are SDPs, the proposition follows from Proposition S3. □

**Proposition S6.** *Return-equivalent Markov decision processes have the same optimal policies.*

*Proof.* Since MDPs are SDPs, the proposition follows from Proposition S4. □

For strictly return-equivalent MDPs the expected return is the same if a state-action sub-sequence is given.

**Proposition S7.** *Strictly return-equivalent MDPs $\tilde{\mathcal{P}}$ and $\mathcal{P}$ have the same expected return for a given state-action sub-sequence $(s_0, a_0, \ldots, s_t, a_t)$, $0 \leqslant t \leqslant T$:*

$$\mathrm{E}_\pi \left[ \tilde{G}_0 \mid s_0, a_0, \ldots, s_t, a_t \right] \ = \ \mathrm{E}_\pi \left[ G_0 \mid s_0, a_0, \ldots, s_t, a_t \right] . \tag{S43}$$

*Proof.* Since MDPs are SDPs, the proposition follows from Lemma S3. □

### S2.3 Reward Redistribution for Strictly Return-Equivalent SDPs

Strictly return-equivalent SDPs $\tilde{\mathcal{P}}$ and $\mathcal{P}$ can be constructed by a reward redistribution.

#### S2.3.1 Reward Redistribution

We define reward redistributions for SDPs.

**Definition S7.** *A* reward redistribution *given an SDP $\tilde{\mathcal{P}}$ is a fixed procedure that redistributes for each state-action sequence $s_0, a_0, \ldots, s_T, a_T$ the realization of the associated return variable $\tilde{G}_0 = \sum_{t=0}^{T} \tilde{R}_{t+1}$ or its expectation $\mathrm{E} \left[ \tilde{G}_0 \mid s_0, a_0, \ldots, s_T, a_T \right]$ along the sequence. The redistribution creates a new SDP $\mathcal{P}$ with redistributed reward $R_{t+1}$ at time $(t+1)$ and return variable $G_0 = \sum_{t=0}^{T} R_{t+1}$. The redistribution procedure ensures for each sequence either $\tilde{G}_0 = G_0$ or*

$$\mathrm{E}_\pi \left[ \tilde{G}_0 \mid s_0, a_0, \ldots, s_T, a_T \right] \ = \ \mathrm{E}_\pi \left[ G_0 \mid s_0, a_0, \ldots, s_T, a_T \right] . \tag{S44}$$

Reward redistributions can be very general. A special case is if the return can be deduced from the past sequence, which makes the return causal.

**Definition S8.** *A reward redistribution is* causal *if for the redistributed reward $R_{t+1}$ the following holds:*

$$\mathrm{E} \left[ R_{t+1} \mid s_0, a_0, \ldots, s_T, a_T \right] \ = \ \mathrm{E} \left[ R_{t+1} \mid s_0, a_0, \ldots, s_t, a_t \right] . \tag{S45}$$

For our approach we only need reward redistributions that are second order Markov.

**Definition S9.** *A causal reward redistribution is* second order Markov *if*

$$\mathrm{E} \left[ R_{t+1} \mid s_0, a_0, \ldots, s_t, a_t \right] \ = \ \mathrm{E} \left[ R_{t+1} \mid s_{t-1}, a_{t-1}, s_t, a_t \right] . \tag{S46}$$

### S2.4 Reward Redistribution Constructs Strictly Return-Equivalent SDPs

**Theorem S1.** *If the SDP $\mathcal{P}$ is obtained by reward redistribution from the SDP $\tilde{\mathcal{P}}$, then $\tilde{\mathcal{P}}$ and $\mathcal{P}$ are strictly return-equivalent.*

*Proof.* For redistributing the reward we have for each state-action sequence $s_0, a_0, \ldots, s_T, a_T$ the same return $\tilde{G}_0 = G_0$, therefore

$$\mathrm{E}_\pi \left[ \tilde{G}_0 \mid s_0, a_0, \ldots, s_T, a_T \right] \ = \ \mathrm{E}_\pi \left[ G_0 \mid s_0, a_0, \ldots, s_T, a_T \right] . \tag{S47}$$

For redistributing the expected return the last equation holds by definition. The last equation is the definition of strictly return-equivalent SDPs. □

The next theorem states that the optimal policies are still the same when redistributing the reward.

**Theorem S2.** *If the SDP $\mathcal{P}$ is obtained by reward redistribution from the SDP $\tilde{\mathcal{P}}$, then both SDPs have the same optimal policies.*

*Proof.* According to Theorem S1, the SDP $\mathcal{P}$ is strictly return-equivalent to the SDP $\tilde{\mathcal{P}}$. According to Proposition S3 and Proposition S4 the SDP $\mathcal{P}$ and the SDP $\tilde{\mathcal{P}}$ have the same optimal policies. □

### S2.4.1 Special Cases of Strictly Return-Equivalent Decision Processes: Reward Shaping, Look-Ahead Advice, and Look-Back Advice

Redistributing the reward via reward shaping [49, 90], look-ahead advice, and look-back advice [91] is a special case of reward redistribution that leads to MDPs which are strictly return-equivalent to the original MDP. We show that reward shaping is a special case of reward redistributions that lead to MDPs which are strictly return-equivalent to the original MDP. First, we subtract from the potential the constant $c = (\Phi(s_0, a_0) - \gamma^T \Phi(s_T, a_T))/(1 - \gamma^T)$, which is the potential of the initial state minus the discounted potential in the last state divided by a fixed divisor. Consequently, the sum of additional rewards in reward shaping, look-ahead advice, or look-back advice from $1$ to $T$ is zero. The original sum of additional rewards is

$$\sum_{i=1}^{T} \gamma^{i-1} \left( \gamma \Phi(s_i, a_i) - \Phi(s_{i-1}, a_{i-1}) \right) = \gamma^T \Phi(s_T, a_T) - \Phi(s_0, a_0) . \tag{S48}$$

If we assume $\gamma^T \Phi(s_T, a_T) = 0$ and $\Phi(s_0, a_0) = 0$, then reward shaping does not change the return and the shaping reward is a reward redistribution leading to an MDP that is strictly return-equivalent to the original MDP. For $T \to \infty$ only $\Phi(s_0, a_0) = 0$ is required. The assumptions can always be fulfilled by adding a single new initial state and a single new final state to the original MDP. Without the assumptions $\gamma^T \Phi(s_T, a_T) = 0$ and $\Phi(s_0, a_0) = 0$, we subtract $c = (\Phi(s_0, a_0) - \gamma^T \Phi(s_T, a_T))/(1 - \gamma^T)$ from all potentials $\Phi$, and obtain

$$\sum_{i=1}^{T} \gamma^{i-1} \left( \gamma(\Phi(s_i, a_i) - c) - (\Phi(s_{i-1}, a_{i-1}) - c) \right) = 0 . \tag{S49}$$

Therefore, the potential-based shaping function (the additional reward) added to the original reward does not change the return, which means that the shaping reward is a reward redistribution that leads to an MDP that is strictly return-equivalent to the original MDP. Obviously, reward shaping is a special case of reward redistribution that leads to a strictly return-equivalent MDP. Reward shaping does not change the general learning behavior if a constant $c$ is subtracted from the potential function $\Phi$. The $Q$-function of the original reward shaping and the $Q$-function of the reward shaping, which has a constant $c$ subtracted from the potential function $\Phi$, differ by $c$ for every $Q$-value [49, 90]. For infinite horizon MDPs with $\gamma < 1$, the terms $\gamma^T$ and $\gamma^T \Phi(s_T, a_T)$ vanish, therefore it is sufficient to subtract $c = \Phi(s_0, a_0)$ from the potential function.

Since TD based reward shaping methods keep the original reward, they can still be exponentially slow for delayed rewards. Reward shaping methods like reward shaping, look-ahead advice, and look-back advice rely on the Markov property of the original reward, while an optimal reward redistribution is not Markov. In general, reward shaping does not lead to an optimal reward redistribution according to Section S2.6.1.

As discussed in Paragraph S2.9, the optimal reward redistribution does not comply to the Bellman equation. Also look-ahead advice does not comply to the Bellman equation. The return for the look-ahead advice reward $\tilde{R}_{t+1}$ is

$$G_t = \sum_{i=0}^{\infty} \tilde{R}_{t+i+1} \tag{S50}$$

with expectations for the reward $\tilde{R}_{t+1}$

$$\mathrm{E}_\pi \left[ \tilde{R}_{t+1} \mid s_{t+1}, a_{t+1}, s_t, a_t \right] = \tilde{r}(s_{t+1}, a_{t+1}, s_t, a_t) = \gamma \Phi(s_{t+1}, a_{t+1}) - \Phi(s_t, a_t) . \tag{S51}$$

The expected reward $\tilde{r}(s_{t+1}, a_{t+1}, s_t, a_t)$ depends on future states $s_{t+1}$ and, more importantly, on future actions $a_{t+1}$. It is a non-causal reward redistribution. Therefore look-ahead advice cannot be directly used for selecting the optimal action at time $t$. For look-back advice we have

$$\mathrm{E}_\pi \left[ \tilde{R}_{t+1} \mid s_t, a_t, s_{t-1}, a_{t-1} \right] = \tilde{r}(s_t, a_t, s_{t-1}, a_{t-1}) = \Phi(s_t, a_t) - \gamma^{-1} \Phi(s_{t-1}, a_{t-1}) . \tag{S52}$$

Therefore look-back advice introduces a second-order Markov reward like the optimal reward redistribution.

## S2.5    Transforming an Immediate Reward MDP to a Delayed Reward MDP

We assume to have a Markov decision process $\mathcal{P}$ with immediate reward. The MDP $\mathcal{P}$ is transformed into an MDP $\tilde{\mathcal{P}}$ with delayed reward, where the reward is given at sequence end. The reward-equivalent MDP $\tilde{\mathcal{P}}$ with delayed reward is state-enriched, which ensures that it is an MDP.
The state-enriched MDP $\tilde{\mathcal{P}}$ has

- reward:

$$\tilde{R}_t \; = \; \begin{cases} 0 \, , & \text{for } t \leqslant T \\ \sum_{k=0}^T R_{k+1} \, , & \text{for } t = T+1 \, . \end{cases} \tag{S53}$$

- state:

$$\tilde{s}_t \; = \; (s_t, \rho_t) \, , \tag{S54}$$

$$\rho_t \; = \; \sum_{k=0}^{t-1} r_{k+1} \, , \quad \text{with } R_{k+1} = r_{k+1} \, . \tag{S55}$$

Here we assume that $\rho$ can only take a finite number of values to assure that the enriched states $\tilde{s}$ are finite. If the original reward was continuous, then $\rho$ can represent the accumulated reward with any desired precision if the sequence length is $T$ and the original reward was bounded. We assume that $\rho$ is sufficiently precise to distinguish the optimal policies, which are deterministic, from sub-optimal deterministic policies. The random variable $R_{k+1}$ is distributed according to $p(r \mid s_k, a_k)$. We assume that the time $t$ is coded in $s$ in order to know when the episode ends and reward is no longer received, otherwise we introduce an additional state variable $\tau = t$ that codes the time.

**Proposition S8.** *If a Markov decision process $\mathcal{P}$ with immediate reward is transformed by above defined $\tilde{R}_t$ and $\tilde{s}_t$ to a Markov decision process $\tilde{\mathcal{P}}$ with delayed reward, where the reward is given at sequence end, then: (I) the optimal policies do not change, and (II) for $\tilde{\pi}(a \mid \tilde{s}) = \pi(a \mid s)$*

$$\tilde{q}^{\tilde{\pi}}(\tilde{s}, a) \; = \; q^{\pi}(s, a) \; + \; \sum_{k=0}^{t-1} r_{k+1} \, , \tag{S56}$$

*for $\tilde{S}_t = \tilde{s}$, $S_t = s$, and $A_t = a$.*

*Proof.* For (I) we first perform an state-enrichment of $\mathcal{P}$ by $\tilde{s}_t = (s_t, \rho_t)$ with $\rho_t = \sum_{k=0}^{t-1} r_{k+1}$ for $R_{k+1} = r_{k+1}$ leading to an intermediate MDP. We assume that the finite-valued $\rho$ is sufficiently precise to distinguish the optimal policies, which are deterministic, from sub-optimal deterministic policies. Proposition S1 ensures that neither the optimal $Q$-values nor the optimal policies change between the original MDP $\mathcal{P}$ and the intermediate MDP. Next, we redistribute the original reward $R_{t+1}$ according to the redistributed reward $\tilde{R}_t$. The new MDP $\tilde{\mathcal{P}}$ with state enrichment and reward redistribution is strictly return-equivalent to the intermediate MDP with state enrichment but the original reward. The new MDP $\tilde{\mathcal{P}}$ is Markov since the enriched state ensures that $\tilde{R}_{T+1}$ is Markov. Proposition S5 and Proposition S6 ensure that the optimal policies are the same.
For (II) we show a proof without Bellman equation and a proof using the Bellman equation.
**Equivalence without Bellman equation.** We have $\tilde{G}_0 = G_0$. The Markov property ensures that the future reward is independent of the already received reward:

$$\mathrm{E}_{\pi} \left[ \sum_{k=t}^T R_{k+1} \mid S_t = s, A_t = a, \rho = \sum_{k=0}^{t-1} r_{k+1} \right] \; = \; \mathrm{E}_{\pi} \left[ \sum_{k=t}^T R_{k+1} \mid S_t = s, A_t = a \right] \, . \tag{S57}$$

We assume $\tilde{\pi}(a \mid \tilde{s}) = \pi(a \mid s)$.

We obtain

$$\tilde{q}^{\tilde{\pi}}(\tilde{s}, a) = \mathrm{E}_{\tilde{\pi}}\left[\tilde{G}_0 \mid \tilde{S}_t = \tilde{s}, A_t = a\right] \tag{S58}$$

$$= \mathrm{E}_{\tilde{\pi}}\left[\sum_{k=0}^{T} R_{k+1} \mid S_t = s, \rho = \sum_{k=0}^{t-1} r_{k+1}, A_t = a\right]$$

$$= \mathrm{E}_{\tilde{\pi}}\left[\sum_{k=t}^{T} R_{k+1} \mid S_t = s, \rho = \sum_{k=0}^{t-1} r_{k+1}, A_t = a\right] + \sum_{k=0}^{t-1} r_{k+1}$$

$$= \mathrm{E}_{\pi}\left[\sum_{k=t}^{T} R_{k+1} \mid S_t = s, A_t = a\right] + \sum_{k=0}^{t-1} r_{k+1}$$

$$= q^{\pi}(s, a) + \sum_{k=0}^{t-1} r_{k+1} \ .$$

We used $\mathrm{E}_{\tilde{\pi}} = \mathrm{E}_{\pi}$, which is ensured since reward probabilities, transition probabilities, and the probability of choosing an action by the policy correspond to each other in both settings.

Since the optimal policies do not change for reward-equivalent and state-enriched processes, we have

$$\tilde{q}^*(\tilde{s}, a) = q^*(s, a) + \sum_{k=0}^{t-1} r_{k+1} \ . \tag{S59}$$

**Equivalence with Bellman equation.** With $q^{\pi}(s, a)$ as optimal action-value function for the original Markov decision process, we define a new Markov decision process with action-state function $\tilde{q}^{\tilde{\pi}}$. For $\tilde{S}_t = \tilde{s}$, $S_t = s$, and $A_t = a$ we have

$$\tilde{q}^{\tilde{\pi}}(\tilde{s}, a) := q^{\pi}(s, a) + \sum_{k=0}^{t-1} r_{k+1} \ , \tag{S60}$$

$$\tilde{\pi}(a \mid \tilde{s}) := \pi(a \mid s) \ . \tag{S61}$$

Since $\tilde{s}' = (s', \rho')$, $\rho' = r + \rho$, and $\tilde{r}$ is constant, the values $\tilde{S}_{t+1} = \tilde{s}'$ and $\tilde{R}_{t+1} = \tilde{r}$ can be computed from $R_{t+1} = r$, $\rho$, and $S_{t+1} = s'$. Therefore, we have

$$\tilde{p}(\tilde{s}', \tilde{r} \mid s, \rho, a) = \tilde{p}(s', \rho', \tilde{r} \mid s, \rho, a) = p(s', r \mid s, a) \ . \tag{S62}$$

For $t < T$, we have $\tilde{r} = 0$ and $\rho' = r + \rho$, where we set $r = r_{t+1}$:

$$\tilde{q}^{\tilde{\pi}}(\tilde{s}, a) = q^{\pi}(s, a) + \sum_{k=0}^{t-1} r_{k+1} \tag{S63}$$

$$= \sum_{s', r} p(s', r \mid s, a) \left[r + \sum_{a'} \pi(a' \mid s') \, q^{\pi}(s', a')\right] + \sum_{k=0}^{t-1} r_{k+1}$$

$$= \sum_{s', \rho'} \tilde{p}(s', \rho', \tilde{r} \mid s, \rho, a) \left[r + \sum_{a'} \pi(a' \mid s') \, q^{\pi}(s', a')\right] + \sum_{k=0}^{t-1} r_{k+1}$$

$$= \sum_{\tilde{s}', \tilde{r}} \tilde{p}(\tilde{s}', \tilde{r} \mid \tilde{s}, a) \left[r + \sum_{a'} \pi(a' \mid s') \, q^{\pi}(s', a') + \sum_{k=0}^{t-1} r_{k+1}\right]$$

$$= \sum_{\tilde{s}', \tilde{r}} \tilde{p}(\tilde{s}', \tilde{r} \mid \tilde{s}, a) \left[\tilde{r} + \sum_{a'} \pi(a' \mid s') \, q^{\pi}(s', a') + \sum_{k=0}^{t} r_{k+1}\right]$$

$$= \sum_{\tilde{s}', \tilde{r}} \tilde{p}(\tilde{s}', \tilde{r} \mid \tilde{s}, a) \left[\tilde{r} + \sum_{a'} \tilde{\pi}(a' \mid \tilde{s}') \, \tilde{q}^{\tilde{\pi}}(\tilde{s}', a')\right] \ .$$

For $t = T$ we have $\tilde{r} = \sum_{k=0}^{T} r_{k+1} = \rho'$ and $q^{\pi}(s', a') = 0$ as well as $\tilde{q}^{\tilde{\pi}}(\tilde{s}', a') = 0$. Both $q$ and $\tilde{q}$ must be zero for $t \geqslant T$ since after time $t = T + 1$ there is no more reward. We obtain for $t = T$ and

$r = r_{T+1}$:

$$\tilde{q}^{\tilde{\pi}}(\tilde{s}, a) = q^{\pi}(s, a) + \sum_{k=0}^{T-1} r_{k+1} \tag{S64}$$

$$= \sum_{s',r} p(s', r \mid s, a) \left[ r + \sum_{a'} \pi(a' \mid s') \, q^{\pi}(s', a') \right] + \sum_{k=0}^{T-1} r_{k+1}$$

$$= \sum_{s',\rho',r} \tilde{p}(s', \rho' \mid s, \rho, a) \left[ r + \sum_{a'} \pi(a' \mid s') \, q^{\pi}(s', a') \right] + \sum_{k=0}^{T-1} r_{k+1}$$

$$= \sum_{s',\rho',r} \tilde{p}(s', \rho' \mid s, \rho, a) \left[ \sum_{k=0}^{T} r_{k+1} + \sum_{a'} \pi(a' \mid s') \, q^{\pi}(s', a') \right]$$

$$= \sum_{\tilde{s}',\rho'} \tilde{p}(\tilde{s}' \mid \tilde{s}, a) \left[ \rho' + \sum_{a'} \pi(a' \mid s') \, q^{\pi}(s', a') \right]$$

$$= \sum_{\tilde{s}',\rho'} \tilde{p}(\tilde{s}' \mid \tilde{s}, a) \, [\rho' + 0]$$

$$= \sum_{\tilde{s}',\tilde{r}} \tilde{p}(\tilde{s}' \mid \tilde{s}, a) \left[ \tilde{r} + \sum_{a'} \tilde{\pi}(a' \mid \tilde{s}') \, \tilde{q}^{\tilde{\pi}}(\tilde{s}', a') \right] .$$

Since $\tilde{q}^{\tilde{\pi}}(\tilde{s}, a)$ fulfills the Bellman equation, it is the action-value function for $\tilde{\pi}$.

□

## S2.6 Transforming an Delayed Reward MDP to an Immediate Reward SDP

Next we consider the opposite direction, where the delayed reward MDP $\tilde{\mathcal{P}}$ is given and we want to find an immediate reward SDP $\mathcal{P}$ that is return-equivalent to $\tilde{\mathcal{P}}$. We assume an episodic reward for $\tilde{\mathcal{P}}$, that is, reward is only given at sequence end. The realization of final reward, that is the realization of the return, $\tilde{r}_{T+1}$ is redistributed to previous time steps. Instead of redistributing the realization $\tilde{r}_{T+1}$ of the random variable $\tilde{R}_{T+1}$, also its expectation $\tilde{r}(s_T, a_T) = \mathrm{E}\left[ \tilde{R}_{T+1} \mid s_T, a_T \right]$ can be redistributed since $Q$-value estimation considers only the mean. We used the Markov property

$$\mathrm{E}_{\pi}\left[ \tilde{G}_0 \mid s_0, a_0, \ldots, s_T, a_T \right] = \mathrm{E}_{\pi}\left[ \sum_{t=0}^{T} \tilde{R}_{t+1} \mid s_0, a_0, \ldots, s_T, a_T \right] \tag{S65}$$

$$= \mathrm{E}\left[ \tilde{R}_{T+1} \mid s_0, a_0, \ldots, s_T, a_T \right]$$

$$= \mathrm{E}\left[ \tilde{R}_{T+1} \mid s_T, a_T \right] .$$

Redistributing the expectation reduces the variance of estimators since the variance of the random variable is already factored out.

We assume a delayed reward MDP $\tilde{\mathcal{P}}$ with reward

$$\tilde{R}_t = \begin{cases} 0, & \text{for } t \leqslant T \\ \tilde{R}_{T+1}, & \text{for } t = T+1 , \end{cases} \tag{S66}$$

where $\tilde{R}_t = 0$ means that the random variable $\tilde{R}_t$ is always zero. The expected reward at the last time step is

$$\tilde{r}(s_T, a_T) = \mathrm{E}\left[ \tilde{R}_{T+1} \mid s_T, a_T \right] , \tag{S67}$$

which is also the expected return. Given a state-action sequence $(s_0, a_0, \ldots, s_T, a_T)$, we want to redistribute either the realization $\tilde{r}_{T+1}$ of the random variable $\tilde{R}_{T+1}$ or its expectation $\tilde{r}(s_T, a_T)$,

### S2.6.1 Optimal Reward Redistribution

The main goal in this paper is to derive an SDP via reward redistribution that has zero expected future rewards. Consequently the SDP has no delayed rewards. To measure the amount of delayed rewards, we define the expected sum of delayed rewards $\kappa(m, t-1)$.

**Definition S10.** *For $1 \leqslant t \leqslant T$ and $0 \leqslant m \leqslant T - t$, the expected sum of delayed rewards at time $(t-1)$ in the interval $[t+1, t+m+1]$ is defined as*

$$\kappa(m, t-1) = \mathrm{E}_\pi \left[ \sum_{\tau=0}^m R_{t+1+\tau} \mid s_{t-1}, a_{t-1} \right] . \tag{S68}$$

The Bellman equation for $Q$-values becomes

$$q^\pi(s_t, a_t) = r(s_t, a_t) + \kappa(T - t - 1, t) , \tag{S69}$$

where $\kappa(T - t - 1, t)$ is the expected sum of future rewards until sequence end given $(s_t, a_t)$, that is, in the interval $[t+2, T+1]$. We aim to derive an MDP with $\kappa(T - t - 1, t) = 0$, which gives $q^\pi(s_t, a_t) = r(s_t, a_t)$. In this case, learning the $Q$-values reduces to estimating the average immediate reward $r(s_t, a_t) = \mathrm{E}\left[R_{t+1} \mid s_t, a_t\right]$. Hence, the reinforcement learning task reduces to computing the mean, e.g. the arithmetic mean, for each state-action pair $(s_t, a_t)$. Next, we define an optimal reward redistribution.

**Definition S11.** *A reward redistribution is optimal, if $\kappa(T - t - 1, t) = 0$ for $0 \leqslant t \leqslant T - 1$.*

Next theorem states that in general an MDP with optimal reward redistribution does not exist, which is the reason why we will consider SDPs in the following.

**Theorem S3.** *In general, an optimal reward redistribution violates the assumption that the reward distribution is Markov, therefore the Bellman equation does not hold.*

*Proof.* We assume an MDP $\tilde{\mathcal{P}}$ with $\tilde{r}(s_T, a_T) \neq 0$ and which has policies that lead to different expected returns at time $t = 0$. If all reward is given at time $t = 0$, all policies have the same expected return at time $t = 0$. This violates our assumption, therefore not all reward can be given at $t = 0$. In vector and matrix notation the Bellman equation is

$$\boldsymbol{q}_t^\pi = \boldsymbol{r}_t + \boldsymbol{P}_{t \to t+1} \, \boldsymbol{q}_{t+1}^\pi , \tag{S70}$$

where $\boldsymbol{P}_{t \to t+1}$ is the row-stochastic matrix with $p(s_{t+1} \mid s_t, a_t)\pi(a_{t+1} \mid s_{t+1})$ at positions $((s_t, a_t), (s_{t+1}, a_{t+1}))$. An optimal reward redistribution requires the expected future rewards to be zero:

$$\boldsymbol{P}_{t \to t+1} \, \boldsymbol{q}_{t+1}^\pi = \boldsymbol{0} \tag{S71}$$

and, since optimality requires $\boldsymbol{q}_{t+1}^\pi = \boldsymbol{r}_{t+1}$, we have

$$\boldsymbol{P}_{t \to t+1} \, \boldsymbol{r}_{t+1} = \boldsymbol{0} , \tag{S72}$$

where $\boldsymbol{r}_{t+1}$ is the vector with components $\tilde{r}(s_{t+1}, a_{t+1})$. Since (i) the MDPs are return-equivalent, (ii) $\tilde{r}(s_T, a_T) \neq 0$, and (iii) not all reward is given at $t = 0$, an $(t+1)$ exists with $\boldsymbol{r}_{t+1} \neq \boldsymbol{0}$. We can construct an MDP $\tilde{\mathcal{P}}$ which has (a) at least as many state-action pairs $(s_t, a_t)$ as pairs $(s_{t+1}, a_{t+1})$ and (b) the transition matrix $\boldsymbol{P}_{t \to t+1}$ has full rank. $\boldsymbol{P}_{t \to t+1}\boldsymbol{r}_{t+1} = \boldsymbol{0}$ is now a contradiction to $\boldsymbol{r}_{t+1} \neq 0$ and $\boldsymbol{P}_{t \to t+1}$ has full rank. Consequently, simultaneously ensuring Markov properties and ensuring zero future return is in general not possible. $\quad\square$

For a particular $\pi$, the next theorem states that an optimal reward redistribution, that is $\kappa = 0$, is equivalent to a redistributed reward which expectation is the difference of consecutive $Q$-values of the original delayed reward. The theorem states that an optimal reward redistribution exists but we have to assume an SDP $\mathcal{P}$ that has a second order Markov reward redistribution.

**Theorem S4.** *We assume a delayed reward MDP $\tilde{\mathcal{P}}$, where the accumulated reward is given at sequence end. An new SDP $\mathcal{P}$ is obtained by a second order Markov reward redistribution, which ensures that $\mathcal{P}$ is return-equivalent to $\tilde{\mathcal{P}}$. For a specific $\pi$, the following two statements are equivalent: (I)  $\kappa(T - t - 1, t) = 0$, i.e. the reward redistribution is optimal,*

*(II)* $\mathrm{E}\left[R_{t+1} \mid s_{t-1}, a_{t-1}, s_t, a_t\right] = \tilde{q}^\pi(s_t, a_t) - \tilde{q}^\pi(s_{t-1}, a_{t-1}) .$ $\tag{S73}$

*Furthermore, an optimal reward redistribution fulfills for $1 \leqslant t \leqslant T$ and $0 \leqslant m \leqslant T - t$:*

$$\kappa(m, t-1) = 0 . \tag{S74}$$

*Proof.* PART (I): we assume that the reward redistribution is optimal, that is,

$$\kappa(T - t - 1, t) = 0 . \tag{S75}$$

The redistributed reward $R_{t+1}$ is second order Markov. We abbreviate the expected $R_{t+1}$ by $h_t$:

$$\mathrm{E}\left[R_{t+1} \mid s_{t-1}, a_{t-1}, s_t, a_t\right] = h_t . \tag{S76}$$

The assumptions of Lemma S3 hold for for the delayed reward MDP $\tilde{\mathcal{P}}$ and the redistributed reward SDP $\mathcal{P}$. Therefore for a given state-action sub-sequence $(s_0, a_0, \ldots, s_t, a_t), 0 \leqslant t \leqslant T$:

$$\mathrm{E}_\pi\left[\tilde{G}_0 \mid s_0, a_0, \ldots, s_t, a_t\right] = \mathrm{E}_\pi\left[G_0 \mid s_0, a_0, \ldots, s_t, a_t\right] \tag{S77}$$

with $G_0 = \sum_{\tau=0}^T R_{\tau+1}$ and $\tilde{G}_0 = \tilde{R}_{T+1}$. The Markov property of the MDP $\tilde{\mathcal{P}}$ ensures that the future reward from $t + 1$ on is independent of the past sub-sequence $s_0, a_0, \ldots, s_{t-1}, a_{t-1}$:

$$\mathrm{E}_\pi\left[\sum_{\tau=0}^{T-t} \tilde{R}_{t+1+\tau} \mid s_t, a_t\right] = \mathrm{E}_\pi\left[\sum_{\tau=0}^{T-t} \tilde{R}_{t+1+\tau} \mid s_0, a_0, \ldots, s_t, a_t\right] . \tag{S78}$$

The second order Markov property of the SDP $\mathcal{P}$ ensures that the future reward from $t + 2$ on is independent of the past sub-sequence $s_0, a_0, \ldots, s_{t-1}, a_{t-1}$:

$$\mathrm{E}_\pi\left[\sum_{\tau=0}^{T-t-1} R_{t+2+\tau} \mid s_t, a_t\right] = \mathrm{E}_\pi\left[\sum_{\tau=0}^{T-t-1} R_{t+2+\tau} \mid s_0, a_0, \ldots, s_t, a_t\right] . \tag{S79}$$

Using these properties we obtain

$$
\begin{aligned}
\tilde{q}^\pi(s_t, a_t) &= \mathrm{E}_\pi\left[\sum_{\tau=0}^{T-t} \tilde{R}_{t+1+\tau} \mid s_t, a_t\right] \\
&= \mathrm{E}_\pi\left[\sum_{\tau=0}^{T-t} \tilde{R}_{t+1+\tau} \mid s_0, a_0, \ldots, s_t, a_t\right] \\
&= \mathrm{E}_\pi\left[\tilde{R}_{T+1} \mid s_0, a_0, \ldots, s_t, a_t\right] \\
&= \mathrm{E}_\pi\left[\sum_{\tau=0}^T \tilde{R}_{\tau+1} \mid s_0, a_0, \ldots, s_t, a_t\right] \\
&= \mathrm{E}_\pi\left[\tilde{G}_0 \mid s_0, a_0, \ldots, s_t, a_t\right] \\
&= \mathrm{E}_\pi\left[G_0 \mid s_0, a_0, \ldots, s_t, a_t\right] \\
&= \mathrm{E}_\pi\left[\sum_{\tau=0}^T R_{\tau+1} \mid s_0, a_0, \ldots, s_t, a_t\right] \\
&= \mathrm{E}_\pi\left[\sum_{\tau=0}^{T-t-1} R_{t+2+\tau} \mid s_0, a_0, \ldots, s_t, a_t\right] + \sum_{\tau=0}^t h_\tau \\
&= \mathrm{E}_\pi\left[\sum_{\tau=0}^{T-t-1} R_{t+2+\tau} \mid s_t, a_t\right] + \sum_{\tau=0}^t h_\tau \\
&= \kappa(T - t - 1, t) + \sum_{\tau=0}^t h_\tau \\
&= \sum_{\tau=0}^t h_\tau .
\end{aligned}
\tag{S80}
$$

We used

$$\kappa(T - t - 1, t) = \mathrm{E}_\pi\left[\sum_{\tau=0}^{T-t-1} R_{t+2+\tau} \mid s_t, a_t\right] = 0 . \tag{S81}$$

It follows that

$$\mathrm{E}\left[R_{t+1} \mid s_{t-1}, a_{t-1}, s_t, a_t\right] = h_t \tag{S82}$$
$$= \tilde{q}^\pi(s_t, a_t) - \tilde{q}^\pi(s_{t-1}, a_{t-1}) \,.$$

PART (II): we assume that

$$\mathrm{E}\left[R_{t+1} \mid s_{t-1}, a_{t-1}, s_t, a_t\right] = h_t \tag{S83}$$
$$= \tilde{q}^\pi(s_t, a_t) - \tilde{q}^\pi(s_{t-1}, a_{t-1}) \,.$$

The expectations $\mathrm{E}_\pi\left[. \mid s_{t-1}, a_{t-1}\right]$ like $\mathrm{E}_\pi\left[\tilde{R}_{T+1} \mid s_{t-1}, a_{t-1}\right]$ are expectations over all episodes starting in $(s_{t-1}, a_{t-1})$ and ending in some $(s_T, a_T)$.
First, we consider $m = 0$ and $1 \leqslant t \leqslant T$, therefore $\kappa(0, t-1) = \mathrm{E}_\pi\left[R_{t+1} \mid s_{t-1}, a_{t-1}\right]$. Since $\tilde{r}(s_{t-1}, a_{t-1}) = 0$ for $1 \leqslant t \leqslant T$, we have

$$\tilde{q}^\pi(s_{t-1}, a_{t-1}) = \tilde{r}(s_{t-1}, a_{t-1}) + \sum_{s_t, a_t} p(s_t, a_t \mid s_{t-1}, a_{t-1}) \, \tilde{q}^\pi(s_t, a_t) \tag{S84}$$

$$= \sum_{s_t, a_t} p(s_t, a_t \mid s_{t-1}, a_{t-1}) \, \tilde{q}^\pi(s_t, a_t) \,.$$

Using this equation we obtain for $1 \leqslant t \leqslant T$:

$$\kappa(0, t-1) = \mathrm{E}_{s_t, a_t, R_{t+1}}\left[R_{t+1} \mid s_{t-1}, a_{t-1}\right] \tag{S85}$$
$$= \mathrm{E}_{s_t, a_t}\left[\tilde{q}^\pi(s_t, a_t) - \tilde{q}^\pi(s_{t-1}, a_{t-1}) \mid s_{t-1}, a_{t-1}\right]$$
$$= \sum_{s_t, a_t} p(s_t, a_t \mid s_{t-1}, a_{t-1}) \, (\tilde{q}^\pi(s_t, a_t) - \tilde{q}^\pi(s_{t-1}, a_{t-1}))$$
$$= \tilde{q}^\pi(s_{t-1}, a_{t-1}) - \sum_{s_t, a_t} p(s_t, a_t \mid s_{t-1}, a_{t-1}) \, \tilde{q}^\pi(s_{t-1}, a_{t-1})$$
$$= \tilde{q}^\pi(s_{t-1}, a_{t-1}) - \tilde{q}^\pi(s_{t-1}, a_{t-1}) = 0 \,.$$

Next, we consider the expectation of $\sum_{\tau=0}^m R_{t+1+\tau}$ for $1 \leqslant t \leqslant T$ and $1 \leqslant m \leqslant T - t$ (for $m > 0$)

$$\kappa(m, t-1) = \mathrm{E}_\pi\left[\sum_{\tau=0}^m R_{t+1+\tau} \mid s_{t-1}, a_{t-1}\right] \tag{S86}$$
$$= \mathrm{E}_\pi\left[\sum_{\tau=0}^m (\tilde{q}^\pi(s_{\tau+t}, a_{\tau+t}) - \tilde{q}^\pi(s_{\tau+t-1}, a_{\tau+t-1})) \mid s_{t-1}, a_{t-1}\right]$$
$$= \mathrm{E}_\pi\left[\tilde{q}^\pi(s_{t+m}, a_{t+m}) - \tilde{q}^\pi(s_{t-1}, a_{t-1}) \mid s_{t-1}, a_{t-1}\right]$$
$$= \mathrm{E}_\pi\left[\mathrm{E}_\pi\left[\sum_{\tau=t+m}^T \tilde{R}_{\tau+1} \mid s_{t+m}, a_{t+m}\right] \mid s_{t-1}, a_{t-1}\right]$$
$$- \mathrm{E}_\pi\left[\mathrm{E}_\pi\left[\sum_{\tau=t-1}^T \tilde{R}_{\tau+1} \mid s_{t-1}, a_{t-1}\right] \mid s_{t-1}, a_{t-1}\right]$$
$$= \mathrm{E}_\pi\left[\tilde{R}_{T+1} \mid s_{t-1}, a_{t-1}\right] - \mathrm{E}_\pi\left[\tilde{R}_{T+1} \mid s_{t-1}, a_{t-1}\right]$$
$$= 0 \,.$$

We used that $\tilde{R}_{t+1} = 0$ for $t < T$.
For $t = \tau + 1$ and $m = T - t = T - \tau - 1$ we have

$$\kappa(T - \tau - 1, \tau) = 0 \,, \tag{S87}$$

which characterizes an optimal reward redistribution.

$\square$

Thus, an SDP with an optimal reward redistribution has a expected future rewards that are zero. Equation $\kappa(T - t - 1, t) = 0$ means that the new SDP $\mathcal{P}$ has no delayed rewards as shown in next corollary.

**Corollary S1.** *An SDP with an optimal reward redistribution fulfills for $0 \leqslant \tau \leqslant T - t - 1$*

$$\mathrm{E}_\pi \left[ R_{t+1+\tau} \mid s_{t-1}, a_{t-1} \right] \;=\; 0 \,. \tag{S88}$$

*The SDP has no delayed rewards since no state-action pair can increase or decrease the expectation of a future reward.*

*Proof.* For $\tau = 0$ we use $\kappa(m, t - 1) = 0$ from Theorem S4 with $m = 0$:

$$\mathrm{E}_\pi \left[ R_{t+1} \mid s_{t-1}, a_{t-1} \right] \;=\; \kappa(0, t - 1) \;=\; 0 \,. \tag{S89}$$

For $\tau > 0$, we also use $\kappa(m, t - 1) = 0$ from Theorem S4:

$$\mathrm{E}_\pi \left[ R_{t+1+\tau} \mid s_{t-1}, a_{t-1} \right] \;=\; \mathrm{E}_\pi \left[ \sum_{k=0}^{\tau} R_{t+1+k} \;-\; \sum_{k=0}^{\tau-1} R_{t+1+k} \mid s_{t-1}, a_{t-1} \right] \tag{S90}$$

$$= \; \mathrm{E}_\pi \left[ \sum_{k=0}^{\tau} R_{t+1+k} \mid s_{t-1}, a_{t-1} \right] \;-\; \mathrm{E}_\pi \left[ \sum_{k=0}^{\tau-1} R_{t+1+k} \mid s_{t-1}, a_{t-1} \right]$$

$$= \; \kappa(\tau, t - 1) \;-\; \kappa(\tau - 1, t - 1) \;=\; 0 \;-\; 0 \;=\; 0 \,.$$

$\square$

A related approach is to ensure zero return by reward shaping if the exact value function is known [67].

The next theorem states the major advantage of an optimal reward redistribution: $\tilde{q}^\pi(s_t, a_t)$ can be estimated with an offset that depends only on $s_t$ by estimating the expected immediate redistributed reward. Thus, $Q$-value estimation becomes trivial and the computation of the advantage function of the MDP $\tilde{\mathcal{P}}$ is simplified.

**Theorem S5.** *If the reward redistribution is optimal, then the Q-values of the SDP $\mathcal{P}$ are given by*

$$q^\pi(s_t, a_t) \;=\; r(s_t, a_t) \;=\; \tilde{q}^\pi(s_t, a_t) \;-\; \mathrm{E}_{s_{t-1}, a_{t-1}} \left[ \tilde{q}^\pi(s_{t-1}, a_{t-1}) \mid s_t \right] \tag{S91}$$

$$= \; \tilde{q}^\pi(s_t, a_t) \;-\; \psi^\pi(s_t) \,.$$

*The SDP $\mathcal{P}$ and the original MDP $\tilde{\mathcal{P}}$ have the same advantage function. Using a behavior policy $\breve{\pi}$ the expected immediate reward is*

$$\mathrm{E}_{\breve{\pi}} \left[ R_{t+1} \mid s_t, a_t \right] \;=\; \tilde{q}^\pi(s_t, a_t) \;-\; \psi^{\pi, \breve{\pi}}(s_t) \,. \tag{S92}$$

*Proof.* The expected reward $r(s_t, a_t)$ is computed for $0 \leqslant t \leqslant T$, where $s_{-1}, a_{-1}$ are states and actions, which are introduced for formal reasons at the beginning of an episode. The expected reward $r(s_t, a_t)$ is with $\tilde{q}^\pi(s_{-1}, a_{-1}) = 0$:

$$r(s_t, a_t) \;=\; \mathrm{E}_{r_{t+1}} \left[ R_{t+1} \mid s_t, a_t \right] \;=\; \mathrm{E}_{s_{t-1}, a_{t-1}} \left[ \tilde{q}^\pi(s_t, a_t) \;-\; \tilde{q}^\pi(s_{t-1}, a_{t-1}) \mid s_t, a_t \right] \tag{S93}$$

$$= \; \tilde{q}^\pi(s_t, a_t) \;-\; \mathrm{E}_{s_{t-1}, a_{t-1}} \left[ \tilde{q}^\pi(s_{t-1}, a_{t-1}) \mid s_t, a_t \right] \,.$$

The expectations $\mathrm{E}_\pi \left[ . \mid s_t, a_t \right]$ like $\mathrm{E}_\pi \left[ \tilde{R}_{T+1} \mid s_t, a_t \right]$ are expectations over all episodes starting in $(s_t, a_t)$ and ending in some $(s_T, a_T)$.

The $Q$-values for the SDP $\mathcal{P}$ are defined for $0 \leqslant t \leqslant T$ as:

$$q^\pi(s_t, a_t) \;=\; \mathrm{E}_\pi \left[ \sum_{\tau=0}^{T-t} R_{t+1+\tau} \mid s_t, a_t \right] \tag{S94}$$

$$= \; \mathrm{E}_\pi \left[ \tilde{q}^\pi(s_T, a_T) \;-\; \tilde{q}^\pi(s_{t-1}, a_{t-1}) \mid s_t, a_t \right]$$

$$= \; \mathrm{E}_\pi \left[ \tilde{q}^\pi(s_T, a_T) \mid s_t, a_t \right] \;-\; \mathrm{E}_\pi \left[ \tilde{q}^\pi(s_{t-1}, a_{t-1}) \mid s_t, a_t \right]$$

$$= \; \tilde{q}^\pi(s_t, a_t) \;-\; \mathrm{E}_{s_{t-1}, a_{t-1}} \left[ \tilde{q}^\pi(s_{t-1}, a_{t-1}) \mid s_t, a_t \right]$$

$$= \; r(s_t, a_t) \,.$$

The second equality uses

$$\sum_{\tau=0}^{T-t} R_{t+1+\tau} = \sum_{\tau=0}^{T-t} \tilde{q}^\pi(s_{t+\tau}, a_{t+\tau}) - \tilde{q}^\pi(s_{t+\tau-1}, a_{t+\tau-1}) \tag{S95}$$

$$= \tilde{q}^\pi(s_T, a_T) - \tilde{q}^\pi(s_{t-1}, a_{t-1}) \ .$$

The posterior $p(s_{t-1}, a_{t-1} \mid s_t, a_t)$ is

$$p(s_{t-1}, a_{t-1} \mid s_t, a_t) = \frac{p(s_t, a_t \mid s_{t-1}, a_{t-1}) \, p(s_{t-1}, a_{t-1})}{p(s_t, a_t)} \tag{S96}$$

$$= \frac{p(s_t \mid s_{t-1}, a_{t-1}) \, p(s_{t-1}, a_{t-1})}{p(s_t)} = p(s_{t-1}, a_{t-1} \mid s_t) \ ,$$

where we used $p(s_t, a_t \mid s_{t-1}, a_{t-1}) = \pi(a_t \mid s_t) p(s_t \mid s_{t-1}, a_{t-1})$ and $p(s_t, a_t) = \pi(a_t \mid s_t) p(s_t)$. The posterior does no longer contain $a_t$. We can express the mean of previous $Q$-values by the posterior $p(s_{t-1}, a_{t-1} \mid s_t, a_t)$:

$$\mathrm{E}_{s_{t-1}, a_{t-1}} \left[ \tilde{q}^\pi(s_{t-1}, a_{t-1}) \mid s_t, a_t \right] = \sum_{s_{t-1}, a_{t-1}} p(s_{t-1}, a_{t-1} \mid s_t, a_t) \, \tilde{q}^\pi(s_{t-1}, a_{t-1}) \tag{S97}$$

$$= \sum_{s_{t-1}, a_{t-1}} p(s_{t-1}, a_{t-1} \mid s_t) \, \tilde{q}^\pi(s_{t-1}, a_{t-1}) = \mathrm{E}_{s_{t-1}, a_{t-1}} \left[ \tilde{q}^\pi(s_{t-1}, a_{t-1}) \mid s_t \right] = \psi^\pi(s_t) \ ,$$

with

$$\psi^\pi(s_t) = \mathrm{E}_{s_{t-1}, a_{t-1}} \left[ \tilde{q}^\pi(s_{t-1}, a_{t-1}) \mid s_t \right] \ . \tag{S98}$$

The SDP $\mathcal{P}$ and the MDP $\tilde{\mathcal{P}}$ have the same advantage function, since the value functions are the expected $Q$-values across the actions and follow the equation $v^\pi(s_t) = \tilde{v}^\pi(s_t) + \psi^\pi(s_t)$. Therefore $\psi^\pi(s_t)$ cancels in the advantage function of the SDP $\mathcal{P}$.

Using a behavior policy $\breve{\pi}$ the expected immediate reward is

$$\mathrm{E}_{\breve{\pi}} \left[ R_{t+1} \mid s_t, a_t \right] = \mathrm{E}_{r_{t+1}, \breve{\pi}} \left[ R_{t+1} \mid s_t, a_t \right] = \mathrm{E}_{s_{t-1}, a_{t-1}, \breve{\pi}} \left[ \tilde{q}^\pi(s_t, a_t) - \tilde{q}^\pi(s_{t-1}, a_{t-1}) \mid s_t, a_t \right] \tag{S99}$$

$$= \tilde{q}^\pi(s_t, a_t) - \mathrm{E}_{s_{t-1}, a_{t-1}, \breve{\pi}} \left[ \tilde{q}^\pi(s_{t-1}, a_{t-1}) \mid s_t, a_t \right] \ .$$

The posterior $p_{\breve{\pi}}(s_{t-1}, a_{t-1} \mid s_t, a_t)$ is

$$p_{\breve{\pi}}(s_{t-1}, a_{t-1} \mid s_t, a_t) = \frac{p_{\breve{\pi}}(s_t, a_t \mid s_{t-1}, a_{t-1}) \, p_{\breve{\pi}}(s_{t-1}, a_{t-1})}{p_{\breve{\pi}}(s_t, a_t)} \tag{S100}$$

$$= \frac{p(s_t \mid s_{t-1}, a_{t-1}) \, p_{\breve{\pi}}(s_{t-1}, a_{t-1})}{p_{\breve{\pi}}(s_t)} = p_{\breve{\pi}}(s_{t-1}, a_{t-1} \mid s_t) \ ,$$

where we used $p_{\breve{\pi}}(s_t, a_t \mid s_{t-1}, a_{t-1}) = \breve{\pi}(a_t \mid s_t) p(s_t \mid s_{t-1}, a_{t-1})$ and $p_{\breve{\pi}}(s_t, a_t) = \breve{\pi}(a_t \mid s_t) p_{\breve{\pi}}(s_t)$. The posterior does no longer contain $a_t$. We can express the mean of previous $Q$-values by the posterior $p_{\breve{\pi}}(s_{t-1}, a_{t-1} \mid s_t, a_t)$:

$$\mathrm{E}_{s_{t-1}, a_{t-1}, \breve{\pi}} \left[ \tilde{q}^\pi(s_{t-1}, a_{t-1}) \mid s_t, a_t \right] = \sum_{s_{t-1}, a_{t-1}} p_{\breve{\pi}}(s_{t-1}, a_{t-1} \mid s_t, a_t) \, \tilde{q}^\pi(s_{t-1}, a_{t-1}) \tag{S101}$$

$$= \sum_{s_{t-1}, a_{t-1}} p_{\breve{\pi}}(s_{t-1}, a_{t-1} \mid s_t) \, \tilde{q}^\pi(s_{t-1}, a_{t-1}) = \mathrm{E}_{s_{t-1}, a_{t-1}, \breve{\pi}} \left[ \tilde{q}^\pi(s_{t-1}, a_{t-1}) \mid s_t \right] = \psi^{\pi, \breve{\pi}}(s_t) \ ,$$

with

$$\psi^{\pi, \breve{\pi}}(s_t) = \mathrm{E}_{s_{t-1}, a_{t-1}, \breve{\pi}} \left[ \tilde{q}^\pi(s_{t-1}, a_{t-1}) \mid s_t \right] \ . \tag{S102}$$

Therefore we have

$$\mathrm{E}_{\breve{\pi}} \left[ R_{t+1} \mid s_t, a_t \right] = \tilde{q}^\pi(s_t, a_t) - \psi^{\pi, \breve{\pi}}(s_t) \ . \tag{S103}$$

$\square$

## S2.7 Novel Learning Algorithms based on Reward Redistributions

We assume $\gamma = 1$ and a finite horizon or absorbing state original MDP $\tilde{\mathcal{P}}$ with delayed reward. According to Theorem S5, $\tilde{q}^\pi(s_t, a_t)$ can be estimated with an offset that depends only on $s_t$ by estimating the expected immediate redistributed reward. Thus, $Q$-value estimation becomes trivial and the computation of the advantage function of the MDP $\tilde{\mathcal{P}}$ is simplified. All reinforcement learning methods like policy gradients that use $\arg\max_{a_t} \tilde{q}^\pi(s_t, a_t)$ or the advantage function $\tilde{q}^\pi(s_t, a_t) - \mathrm{E}_{a_t}\tilde{q}^\pi(s_t, a_t)$ of the original MDP $\tilde{\mathcal{P}}$ can be used. These methods either rely on Theorem S5 and either estimate $q^\pi(s_t, a_t)$ according to Eq. (S91) or the expected immediate reward according to Eq. (S92). Both approaches estimate $\tilde{q}^\pi(s_t, a_t)$ with an offset that depends only on $s_t$ (either $\psi^\pi(s_t)$ or $\psi^{\pi,\breve{\pi}}(s_t)$). Behavior policies like "greedy in the limit with infinite exploration" (GLIE) or "restricted rank-based randomized" (RRR) allow to prove convergence of SARSA [71]. These policies can be used with reward redistribution. GLIE policies can be realized by a softmax with exploration coefficient on the $Q$-values, therefore $\psi^\pi(s_t)$ or $\psi^{\pi,\breve{\pi}}(s_t)$ cancels. RRR policies select actions probabilistically according to the ranks of their $Q$-values, where the greedy action has highest probability. Therefore $\psi(s_t)$ or $\psi^{\pi,\breve{\pi}}(s_t)$ is not required. For function approximation, convergence of the $Q$-value estimation together with reward redistribution and GLIE or RRR policies can under standard assumptions be proven by the stochastic approximation theory for two time-scale update rules [9, 31]. Proofs for convergence to an optimal policy are in general difficult, since locally stable attractors may not correspond to optimal policies.

Reward redistribution can be used for

- (A) $Q$-value estimation,
- (B) policy gradients, and
- (C) $Q$-learning.

### S2.7.1 Q-Value Estimation

Like SARSA, RUDDER learning continually predicts $Q$-values to improve the policy. Type (A) methods estimate $Q$-values and are divided into variants (i), (ii), and (iii). Variant (i) assumes an optimal reward redistribution and estimates $\tilde{q}^\pi(s_t, a_t)$ with an offset depending only on $s_t$. The estimates are based on Theorem S5 either by on-policy direct $Q$-value estimation according to Eq. (S91) or by off-policy immediate reward estimation according to Eq. (S92). Variant (ii) methods assume a non-optimal reward redistribution and correct Eq. (S91) by estimating $\kappa$. Variant (iii) methods use eligibility traces for the redistributed reward.

**Variant (i): Estimation of $\tilde{q}^\pi(s_t, a_t)$ with an offset assuming optimality.** Theorem S5 justifies the estimation of $\tilde{q}^\pi(s_t, a_t)$ with an offset by on-policy direct $Q$-value estimation via Eq. (S91) or by off-policy immediate reward estimation via Eq. (S92). RUDDER learning can be based on policies like "greedy in the limit with infinite exploration" (GLIE) or "restricted rank-based randomized" (RRR) [71]. GLIE policies change toward greediness with respect to the $Q$-values during learning.

**Variant (ii): TD-learning of $\kappa$ and correction of the redistributed reward.** For non-optimal reward redistributions $\kappa(T - t - 1, t)$ can be estimated to correct the $Q$-values. **TD-learning of $\kappa$.** The expected sum of delayed rewards $\kappa(T - t - 1, t)$ can be formulated as

$$
\begin{aligned}
\kappa(T - t - 1, t) &= \mathrm{E}_\pi\left[\sum_{\tau=0}^{T-t-1} R_{t+2+\tau} \mid s_t, a_t\right] \qquad\qquad\qquad\qquad\text{(S104)} \\
&= \mathrm{E}_\pi\left[R_{t+2} + \sum_{\tau=0}^{T-(t+1)-1} R_{(t+1)+2+\tau} \mid s_t, a_t\right] \\
&= \mathrm{E}_{s_{t+1}, a_{t+1}, r_{t+2}}\left[R_{t+2} + \mathrm{E}_\pi\left[\sum_{\tau=0}^{T-(t+1)-1} R_{(t+1)+2+\tau} \mid s_{t+1}, a_{t+1}\right] \mid s_t, a_t\right] \\
&= \mathrm{E}_{s_{t+1}, a_{t+1}, r_{t+2}}\left[R_{t+2} + \kappa(T - t - 2, t + 1) \mid s_t, a_t\right].
\end{aligned}
$$

Therefore, $\kappa(T - t - 1, t)$ can be estimated by $R_{t+2}$ and $\kappa(T - t - 2, t + 1)$, if the last two are drawn together, i.e. considered as pairs. Otherwise the expectations of $R_{t+2}$ and $\kappa(T - t - 2, t + 1)$ given $(s_t, a_t)$ must be estimated. We can use TD-learning if the immediate reward and the sum of delayed

rewards are drawn as pairs, that is, simultaneously. The TD-error $\delta_\kappa$ becomes

$$\delta_\kappa(T - t - 1, t) \;=\; R_{t+2} \;+\; \kappa(T - t - 2, t + 1) \;-\; \kappa(T - t - 1, t)\,. \qquad \text{(S105)}$$

We now define eligibility traces for $\kappa$. Let the $n$-step return samples of $\kappa$ for $1 \leqslant n \leqslant T - t$ be

$$\kappa^{(1)}(T - t - 1, t) \;=\; R_{t+2} \;+\; \kappa(T - t - 2, t + 1) \qquad \text{(S106)}$$

$$\kappa^{(2)}(T - t - 1, t) \;=\; R_{t+2} \;+\; R_{t+3} \;+\; \kappa(T - t - 3, t + 2)$$

$$\cdots$$

$$\kappa^{(n)}(T - t, t) \;=\; R_{t+2} \;+\; R_{t+3} \;+\; \ldots \;+\; R_{t+n+1} \;+\; \kappa(T - t - n - 1, t + n)\,.$$

The $\lambda$-return for $\kappa$ is

$$\kappa^{(\lambda)}(T - t - 1, t) \;=\; (1 - \lambda) \sum_{n=1}^{T-t-1} \lambda^{n-1}\,\kappa^{(n)}(T - t - 1, t) \;+\; \lambda^{T-t-1}\,\kappa^{(T-t)}(T - t - 1, t)\,.$$

$$\text{(S107)}$$

We obtain

$$\begin{aligned}
\kappa^{(\lambda)}(T - t - 1, t) \;=\;\; & R_{t+2} \;+\; \kappa(T - t - 2, t + 1) && \text{(S108)} \\
& +\; \lambda\,\left(R_{t+3} \;+\; \kappa(T - t - 3, t + 2) \;-\; \kappa(T - t - 2, t + 1)\right) \\
& +\; \lambda^2\,\left(R_{t+4} \;+\; \kappa(T - t - 4, t + 3) \;-\; \kappa(T - t - 3, t + 2)\right) \\
& \cdots \\
& +\; \lambda^{T-1-t}\,\left(R_{T+1} \;+\; \kappa(0, T - 1) \;-\; \kappa(1, T - 2)\right)\,.
\end{aligned}$$

We can reformulate this as

$$\kappa^{(\lambda)}(T - t - 1, t) \;=\; \kappa(T - t - 1, t) \;+\; \sum_{n=0}^{T-t-1} \lambda^n\,\delta_\kappa(T - t - n - 1, t + n)\,. \qquad \text{(S109)}$$

The $\kappa$ error $\Delta_\kappa$ is

$$\Delta_\kappa(T - t - 1, t) \;=\; \kappa^{(\lambda)}(T - t - 1, t) \;-\; \kappa(T - t - 1, t) \;=\; \sum_{n=0}^{T-t-1} \lambda^n\,\delta_\kappa(T - t - n - 1, t + n)\,.$$

$$\text{(S110)}$$

The derivative of

$$1/2\,\Delta_\kappa(T - t - 1, t)^2 \;=\; 1/2\,\left(\kappa^{(\lambda)}(T - t - 1, t) \;-\; \kappa(T - t - 1, t; \boldsymbol{w})\right)^2 \qquad \text{(S111)}$$

with respect to $\boldsymbol{w}$ is

$$-\left(\kappa^{(\lambda)}(T - t - 1, t) \;-\; \kappa(T - t - 1, t; \boldsymbol{w})\right)\,\nabla_w\kappa(T - t - 1, t; \boldsymbol{w}) \qquad \text{(S112)}$$

$$=\; -\sum_{n=0}^{T-t-1} \lambda^n\,\delta_\kappa(T - t - n - 1, t + n)\,\nabla_w\kappa(T - t - 1, t; \boldsymbol{w})\,.$$

The full gradient of the sum of $\kappa$ errors is

$$1/2\,\nabla_w \sum_{t=0}^{T-1} \Delta_\kappa(T - t - 1, t)^2 \qquad \text{(S113)}$$

$$=\; -\sum_{t=0}^{T-1} \sum_{n=0}^{T-t-1} \lambda^n\,\delta_\kappa(T - t - n - 1, t + n)\,\nabla_w\kappa(T - t - 1, t; \boldsymbol{w})$$

$$=\; -\sum_{t=0}^{T-1} \sum_{\tau=t}^{T-1} \lambda^{\tau-t}\,\delta_\kappa(T - \tau - 1, \tau)\,\nabla_w\kappa(T - t - 1, t; \boldsymbol{w})$$

$$=\; -\sum_{\tau=0}^{T-1} \delta_\kappa(T - \tau - 1, \tau) \sum_{t=0}^{\tau} \lambda^{\tau-t}\,\nabla_w\kappa(T - t - 1, t; \boldsymbol{w})\,.$$

We set $n = \tau - t$, so that $n = 0$ becomes $\tau = t$ and $n = T - t - 1$ becomes $\tau = T - 1$. The recursion

$$f(t) = \lambda f(t-1) + a_t, \qquad f(0) = 0 \tag{S114}$$

can be written as

$$f(T) = \sum_{t=1}^{T} \lambda^{T-t} a_t. \tag{S115}$$

Therefore, we can use following update rule for minimizing $\sum_{t=0}^{T-1} \Delta_\kappa(T, t)^2$ with respect to $\boldsymbol{w}$ with $1 \leqslant \tau \leqslant T - 1$:

$$\boldsymbol{z}_{-1} = 0 \tag{S116}$$
$$\boldsymbol{z}_\tau = \lambda \boldsymbol{z}_{\tau-1} + \nabla_w \kappa(T - \tau, \tau; \boldsymbol{w}) \tag{S117}$$
$$\delta_\kappa(T - \tau, \tau) = R_{\tau+2} + \kappa(T - \tau - 1, \tau + 1; \boldsymbol{w}) - \kappa(T - \tau, \tau; \boldsymbol{w}) \tag{S118}$$
$$\boldsymbol{w}^{\text{new}} = \boldsymbol{w} + \alpha \, \delta_\kappa(T - \tau, \tau) \, \boldsymbol{z}_\tau. \tag{S119}$$

**Correction of the reward redistribution.** For correcting the redistributed reward, we apply a method similar to reward shaping or look-back advice. This method ensures that the corrected redistributed reward leads to an SDP that is has the same return per sequence as the SDP $\mathcal{P}$. The reward correction is

$$F(s_t, a_t, s_{t-1}, a_{t-1}) = \kappa(m, t) - \kappa(m, t - 1), \tag{S120}$$

we define the corrected redistributed reward as

$$R_{t+1}^{\text{c}} = R_{t+1} + F(s_t, a_t, s_{t-1}, a_{t-1}) = R_{t+1} + \kappa(m, t) - \kappa(m, t - 1). \tag{S121}$$

We assume that $\kappa(m, -1) = \kappa(m, T + 1) = 0$, therefore

$$\sum_{t=0}^{T+1} F(s_t, a_t, s_{t-1}, a_{t-1}) = \sum_{t=0}^{T+1} \kappa(m, t) - \kappa(m, t - 1) = \kappa(m, T + 1) - \kappa(m, -1) = 0. \tag{S122}$$

Consequently, the corrected redistributed reward $R_{t+1}^{\text{c}}$ does not change the expected return for a sequence, therefore, the resulting SDP has the same optimal policies as the SDP without correction. For a predictive reward of $\rho$ at time $t = k$, which can be predicted from time $t = l < k$ to time $t = k - 1$, we have:

$$\kappa(m, t) = \begin{cases} 0, & \text{for } t < l, \\ \rho, & \text{for } l \leqslant t < k, \\ 0, & \text{for } t \geqslant k. \end{cases} \tag{S123}$$

The reward correction is

$$F(s_t, a_t, s_{t-1}, a_{t-1}) = \begin{cases} 0, & \text{for } t < l, \\ \rho, & \text{for } t = l, \\ 0, & \text{for } l < t < k, \\ -\rho, & \text{for } t = k, \\ 0, & \text{for } t > k. \end{cases} \tag{S124}$$

**Using $\kappa$ as auxiliary task in predicting the return for return decomposition.** A $\kappa$ prediction can serve as additional output of the function $g$ that predicts the return and is the basis of the return decomposition. Even a partly prediction of $\kappa$ means that the reward can be distributed further back. If $g$ can partly predict $\kappa$, then $g$ has all information to predict the return earlier in the sequence. If the return is predicted earlier, then the reward will be distributed further back. Consequently, the reward redistribution comes closer to an optimal reward redistribution. However, at the same time, $\kappa$ can no longer be predicted. The function $g$ must find another $\kappa$ that can be predicted. If no such $\kappa$ is found, then optimal reward redistribution is indicated.

**Variant (iii): Eligibility traces assuming optimality.** We can use eligibility traces to further distribute the reward back. For an optimal reward redistribution, we have $\mathrm{E}_{s_{t+1}}[V(s_{t+1})] = 0$. The new returns $\mathcal{R}_t$ are given by the recursion

$$\mathcal{R}_t = r_{t+1} + \lambda \, \mathcal{R}_{t+1} \,, \tag{S125}$$

$$\mathcal{R}_{T+2} = 0 \,. \tag{S126}$$

The expected policy gradient updates with the new returns $\mathcal{R}$ are $\mathrm{E}_\pi\left[\nabla_\theta \log \pi(a_t \mid s_t; \boldsymbol{\theta}) \mathcal{R}_t\right]$. To avoid an estimation of the value function $V(s_{t+1})$, we assume optimality, which might not be valid. However, the error should be small if the return decomposition works well. Instead of estimating a value function, we can use a correction as it is shown in next paragraph.

### S2.7.2 Policy Gradients

Type (B) methods are policy gradients. In the expected updates $\mathrm{E}_\pi\left[\nabla_\theta \log \pi(a \mid s; \boldsymbol{\theta}) q^\pi(s, a)\right]$ of policy gradients, the value $q^\pi(s, a)$ is replaced by an estimate of $r(s, a)$ or by samples of the redistributed reward. Convergence to optimal policies is guaranteed even with the offset $\psi^\pi(s)$ in Eq. (S91) similar to baseline normalization for policy gradients. With baseline normalization, the baseline $b(s) = \mathrm{E}_a[r(s, a)] = \sum_a \pi(a \mid s) r(s, a)$ is subtracted from $r(s, a)$, which gives the policy gradient $\mathrm{E}_\pi\left[\nabla_\theta \log \pi(a \mid s; \boldsymbol{\theta})(r(s, a) - b(s))\right]$. With eligibility traces using $\lambda \in [0, 1]$ for $G_t^\lambda$ [78], we have the new returns $\mathcal{G}_t = r_t + \lambda \mathcal{G}_{t+1}$ with $\mathcal{G}_{T+2} = 0$. The expected updates with the new returns $\mathcal{G}$ are $\mathrm{E}_\pi\left[\nabla_\theta \log \pi(a_t \mid s_t; \boldsymbol{\theta}) \mathcal{G}_t\right]$.

### S2.7.3 Q-Learning

The type (C) method is $Q$-learning with the redistributed reward. Here, $Q$-learning is justified if immediate and future reward are drawn together, as typically done. Also other temporal difference methods are justified when immediate and future reward are drawn together.

### S2.8 Return Decomposition to construct a Reward Redistribution

We now propose methods to construct reward redistributions which ideally would be optimal. Learning with non-optimal reward redistributions *does work* since the optimal policies do not change according to Theorem S2. However reward redistributions that are optimal considerably speed up learning, since future expected rewards introduce biases in TD-methods and the high variance in MC-methods. The expected optimal redistributed reward is according to Eq. (S73) the difference of $Q$-values. The more a reward redistribution deviates from these differences, the larger are the absolute $\kappa$-values and, in turn, the less optimal is the reward redistribution. Consequently we aim at identifying the largest $Q$-value differences to construct a reward redistribution which is close to optimal. Assume a grid world where you have to take a key to later open a door to a treasure room. Taking the key increases the chances to receive the treasure and, therefore, is associated with a large positive $Q$-value difference. Smaller positive $Q$-value difference are steps toward the key location.

**Reinforcement Learning as Pattern Recognition.** We want to transform the reinforcement learning problem into a pattern recognition problem to employ deep learning approaches. The sum of the $Q$-value differences gives the difference between expected return at sequence begin and the expected return at sequence end (telescope sum). Thus, $Q$-value differences allow to predict the expected return of the whole state-action sequence. Identifying the largest $Q$-value differences reduce the prediction error most. $Q$-value differences are assumed to be associated with patterns in state-action transitions like taking the key in our example. The largest $Q$-value differences are expected to be found more frequently in sequences with very large or very low return. The resulting task is to predict the expected return from the whole sequence and identify which state-action transitions contributed most to the prediction. This pattern recognition task is utilized to construct a reward redistribution, where redistributed reward corresponds to the contribution.

### S2.8.1 Return Decomposition Idea

The *return decomposition idea* is to predict the realization of the return or its expectation by a function $g$ from the state-action sequence

$$(s, a)_{0:T} := (s_0, a_0, s_1, a_1, \ldots, s_T, a_T) \,. \tag{S127}$$

The return is the accumulated reward along the whole sequence $(s, a)_{0:T}$. The function $g$ depends on the policy $\pi$ that is used to generate the state-action sequences. Subsequently, the prediction or the realization of the return is distributed over the sequence with the help of $g$. One important

advantage of a deterministic function $g$ is that it predicts with proper loss functions and if being perfect the expected return. Therefore, it removes the sampling variance of returns. In particular the variance of probabilistic rewards is averaged out. Even an imperfect function $g$ removes the variance as it is deterministic. As described later, the sampling variance may be reintroduced when strictly return-equivalent SDPs are ensured. We want to determine for each sequence element its contribution to the prediction of the function $g$. Contribution analysis computes the contribution of each state-action pair to the prediction, that is, the information of each state-action pair about the prediction. In principle, we can use any contribution analysis method. However, we prefer three methods: (A) Differences in predictions. If we can ensure that $g$ predicts the sequence-wide return at every time step. The difference of two consecutive predictions is a measure of the contribution of the current state-action pair to the return prediction. The difference of consecutive predictions is the redistributed reward. (B) Integrated gradients (IG) [76]. (C) Layer-wise relevance propagation (LRP) [1]. The methods (B) and (C) use information later in the sequence for determining the contribution of the current state-action pair. Therefore, they introduce a non-Markov reward. However, the non-Markov reward can be viewed as probabilistic reward. Since probabilistic reward increases the variance, we prefer method (A).

**Explaining Away Problem.** We still have to tackle the problem that reward causing actions do not receive redistributed rewards since they are explained away by later states. To describe the problem, assume an MDP $\tilde{\mathcal{P}}$ with the only reward at sequence end. To ensure the Markov property, states in $\tilde{\mathcal{P}}$ have to store the reward contributions of previous state-actions; e.g. $s_T$ has to store all previous contributions such that the expectation $\tilde{r}(s_T, a_T)$ is Markov. The explaining away problem is that later states are used for return prediction, while reward causing earlier actions are missed. To avoid explaining away, between the state-action pair $(s_t, a_t)$ and its predecessor $(s_{t-1}, a_{t-1})$, where $(s_{-1}, a_{-1})$ are introduced for starting an episode. The sequence of differences is defined as

$$\Delta_{0:T} := \big(\Delta(s_{-1}, a_{-1}, s_0, a_0), \dots, \Delta(s_{T-1}, a_{T-1}, s_T, a_T)\big). \tag{S128}$$

We assume that the differences $\Delta$ are mutually independent [28]:

$$p\left(\Delta(s_{t-1}, a_{t-1}, s_t, a_t) \mid \Delta(s_{-1}, a_{-1}, s_0, a_0), \dots, \Delta(s_{t-2}, a_{t-2}, s_{t-1}, a_{t-1}), \right. \tag{S129}$$
$$\left. \Delta(s_t, a_t, s_{t+1}, a_{t+1}) \dots, \Delta(s_{T-1}, a_{T-1}, s_T, a_T)\right) = p\left(\Delta(s_{t-1}, a_{t-1}, s_t, a_t)\right).$$

The function $g$ predicts the realization of the sequence-wide return or its expectation from the sequence $\Delta_{0:T}$:

$$g\big(\Delta_{0:T}\big) = \mathrm{E}\left[\tilde{R}_{T+1} \mid s_T, a_T\right] = \tilde{r}_{T+1}. \tag{S130}$$

**Return decomposition** deconstructs $g$ into contributions $h_t = h(\Delta(s_{t-1}, a_{t-1}, s_t, a_t)$ at time $t$:

$$g\big(\Delta_{0:T}\big) = \sum_{t=0}^{T} h(\Delta(s_{t-1}, a_{t-1}, s_t, a_t)) = \tilde{r}_{T+1}. \tag{S131}$$

If we can assume that $g$ can predict the return at every time step:

$$g\big(\Delta_{0:t}\big) = \mathrm{E}_\pi\left[\tilde{R}_{T+1} \mid s_t, a_t\right], \tag{S132}$$

then we use the contribution analysis method "differences of return predictions", where the contributions are defined as:

$$h_0 = h(\Delta(s_{-1}, a_{-1}, s_0, a_0)) := g\big(\Delta_{0:0}\big) \tag{S133}$$
$$h_t = h(\Delta(s_{t-1}, a_{t-1}, s_t, a_t)) := g\big(\Delta_{0:t}\big) - g\big(\Delta_{0:(t-1)}\big). \tag{S134}$$

We assume that the sequence-wide return cannot be predicted from the last state. The reason is that either immediate rewards are given only at sequence end without storing them in the states or information is removed from the states. Therefore, a relevant event for predicting the final reward must be identified by the function $g$. The prediction errors at the end of the episode become, in general, smaller since the future is less random. Therefore, prediction errors later in the episode are up-weighted while early predictions ensure that information is captured in $h_t$ for being used later. The prediction at time $T$ has the largest weight and relies on information from the past.

If $g$ does predict the return at every time step, contribution analysis decomposes $g$. For decomposing a linear $g$ one can use the Taylor decomposition (a linear approximation) of $g$ with respect to the $h$ [1, 45]. A non-linear $g$ can be decomposed by layerwise relevance propagation (LRP) [1, 46] or integrated gradients (IG) [76].

### S2.8.2 Reward Redistribution based on Return Decomposition

We assume a return decomposition

$$g\big(\Delta_{0:T}\big) \;=\; \sum_{t=0}^{T} h_t \,, \tag{S135}$$

with

$$h_0 \;=\; h(\Delta(s_{-1}, a_{-1}, s_0, a_0)) \,, \tag{S136}$$
$$h_t \;=\; h(\Delta(s_{t-1}, a_{t-1}, s_t, a_t)) \quad \text{for } 0 < t \leqslant T \,. \tag{S137}$$

We use these contributions for redistributing the reward. The reward redistribution is given by the random variable $R_{t+1}$ for the reward at time $t+1$. These new redistributed rewards $R_{t+1}$ must have the contributions $h_t$ as mean:

$$\mathrm{E}\left[R_{t+1} \mid s_{t-1}, a_{t-1}, s_t, a_t\right] \;=\; h_t \tag{S138}$$

The reward $\tilde{R}_{T+1}$ of $\tilde{\mathcal{P}}$ is probabilistic and the function $g$ might not be perfect, therefore neither $g(\Delta_{0:T}) = \tilde{r}_{T+1}$ for the return realization $\tilde{r}_{T+1}$ nor $g(\Delta_{0:T}) = \tilde{r}(s_T, a_T)$ for the expected return holds. To assure strictly return-equivalent SDPs, we have to compensate for both a probabilistic reward $\tilde{R}_{T+1}$ and an imperfect function $g$. The compensation is given by

$$\tilde{r}_{T+1} \;-\; \sum_{\tau=0}^{T} h_t \,. \tag{S139}$$

We compensate with an extra reward $R_{T+2}$ at time $T+2$ which is immediately given after $R_{T+1}$ at time $T+1$ after the state-action pair $(s_T, a_T)$. The new redistributed reward $R_{t+1}$ is

$$\mathrm{E}\left[R_1 \mid s_0, a_0\right] \;=\; h_0 \,, \tag{S140}$$
$$\mathrm{E}\left[R_{t+1} \mid s_{t-1}, a_{t-1}, s_t, a_t\right] \;=\; h_t \quad \text{for } 0 < t \leqslant T \,, \tag{S141}$$
$$R_{T+2} \;=\; \tilde{R}_{T+1} \;-\; \sum_{t=0}^{T} h_t \,, \tag{S142}$$

where the realization $\tilde{r}_{T+1}$ is replaced by its random variable $\tilde{R}_{T+1}$. If the prediction of $g$ is perfect, then we can set $R_{T+2} = 0$ and redistribute the expected return which is the predicted return. $R_{T+2}$ compensates for both a probabilistic reward $\tilde{R}_{T+1}$ and an imperfect function $g$. Consequently all variance of sampling the return is moved to $R_{T+2}$. Only the imperfect function $g$ must be corrected while the variance does not matter. However, we cannot distinguish, e.g. in early learning phases, between errors of $g$ and random reward. **A perfect $g$ results in an optimal reward redistribution.** Next theorem shows that Theorem S4 holds also for the correction $R_{T+2}$.

**Theorem S6.** *The optimality conditions hold also for reward redistributions with corrections:*

$$\kappa(T - t + 1, t - 1) \;=\; 0 \,. \tag{S143}$$

*Proof.* The expectation of $\kappa(T - t + 1, t - 1) = \sum_{\tau=0}^{T-t+1} R_{t+1+\tau}$, that is $\kappa(m, t-1)$ with $m = T - t + 1$.

$$\mathrm{E}_\pi\left[\sum_{\tau=0}^{T-t+1} R_{t+1+\tau} \mid s_{t-1}, a_{t-1}\right] \tag{S144}$$

$$= \mathrm{E}_\pi\left[\tilde{R}_{T+1} \;-\; \tilde{q}^\pi(s_T, a_T) \;+\; \sum_{\tau=0}^{T-t} (\tilde{q}^\pi(s_{\tau+t}, a_{\tau+t}) \;-\; \tilde{q}^\pi(s_{\tau+t-1}, a_{\tau+t-1})) \mid s_{t-1}, a_{t-1}\right]$$

$$= \mathrm{E}_\pi\left[\tilde{R}_{T+1} \;-\; \tilde{q}^\pi(s_{t-1}, a_{t-1}) \mid s_{t-1}, a_{t-1}\right]$$

$$= \mathrm{E}_\pi\left[\tilde{R}_{T+1} \mid s_{t-1}, a_{t-1}\right] \;-\; \mathrm{E}_\pi\left[\mathrm{E}_\pi\left[\sum_{\tau=t-1}^{T} \tilde{R}_{\tau+1} \mid s_{t-1}, a_{t-1}\right] \mid s_{t-1}, a_{t-1}\right]$$

$$= \mathrm{E}_\pi\left[\tilde{R}_{T+1} \mid s_{t-1}, a_{t-1}\right] \;-\; \mathrm{E}_\pi\left[\tilde{R}_{T+1} \mid s_{t-1}, a_{t-1}\right]$$

$$= 0 \,.$$

If we substitute $t - 1$ by $t$ ($t$ one step further and $m$ one step smaller) it follows

$$\kappa(T - t, t) = 0 .\tag{S145}$$

Next, we consider the case $t = T + 1$, that is $\kappa(0, T)$, which is the expected correction. We will use following equality for the expected delayed reward at sequence end:

$$\tilde{q}^\pi(s_T, a_T) = \mathrm{E}_{\tilde{R}_{T+1}}\left[\tilde{R}_{T+1} \mid s_T, a_T\right] = \tilde{r}_{T+1}(s_T, a_T) ,\tag{S146}$$

since $\tilde{q}^\pi(s_{T+1}, a_{T+1}) = 0$. For $t = T + 1$ we obtain

$$\begin{aligned}\mathrm{E}_{R_{T+2}}\left[R_{T+2} \mid s_T, a_T\right] &= \mathrm{E}_{\tilde{R}_{T+1}}\left[\tilde{R}_{T+1} - \tilde{q}^\pi(s_T, a_T) \mid s_T, a_T\right]\\ &= \tilde{r}_{T+1}(s_T, a_T) - \tilde{r}_{T+1}(s_T, a_T) = 0 .\end{aligned}\tag{S147}$$

$\square$

In the experiments we also use a uniform compensation where each reward has the same contribution to the compensation:

$$R_1 = h_0 + \frac{1}{T+1}\left(\tilde{R}_{T+1} - \sum_{\tau=0}^{T} h(\Delta(s_{\tau-1}, a_{\tau-1}, s_\tau, a_\tau))\right)\tag{S148}$$

$$R_{t+1} = h_t + \frac{1}{T+1}\left(\tilde{R}_{T+1} - \sum_{\tau=0}^{T} h(\Delta(s_{\tau-1}, a_{\tau-1}, s_\tau, a_\tau))\right) .\tag{S149}$$

Consequently all variance of sampling the return is uniformly distributed across the sequence. Also the error of $g$ is uniformly distributed across the sequence.

An optimal reward redistribution implies

$$g(\Delta_{0:t}) = \sum_{\tau=0}^{t} h(\Delta(s_{\tau-1}, a_{\tau-1}, s_\tau, a_\tau)) = \tilde{q}^\pi(s_t, a_t)\tag{S150}$$

since the expected reward is

$$\begin{aligned}\mathrm{E}\left[R_{t+1} \mid s_{t-1}, a_{t-1}, s_t, a_t\right] &= h(\Delta(s_{t-1}, a_{t-1}, s_t, a_t))\\ &= \tilde{q}^\pi(s_t, a_t) - \tilde{q}^\pi(s_{t-1}, a_{t-1})\end{aligned}\tag{S151}$$

according to Eq. (S73) in Theorem S4 and

$$\begin{aligned}h_0 &= h(\Delta(s_{-1}, a_{-1}, s_0, a_0))\\ &= g(\Delta_{0:0}) = \tilde{q}^\pi(s_0, a_0) .\end{aligned}\tag{S152}$$

## S2.9 Remarks on Return Decomposition

### S2.9.1 Return Decomposition for Binary Reward

A special case is a reward that indicates success or failure by giving a reward of 1 or 0, respectively. The return is equal to the final reward $R$, which is a Bernoulli variable. For each state $s$ or each state-action pair $(s, a)$ the expected return can be considered as a Bernoulli variable with success probability $p_R(s)$ or $p_R(s, a)$. The value function is $v^\pi(s) = \mathrm{E}_\pi(G \mid s) = p_R(s)$ and the action-value is $q^\pi(s) = \mathrm{E}_\pi(G \mid s, a) = p_R(s, a)$ which is in both cases the expectation of success. In this case, the optimal reward redistribution tracks the success probability

$$R_1 = h_0 = h(\Delta(s_{-1}, a_{-1}, s_0, a_0)) = \tilde{q}^\pi(s_0, a_0) = p_R(s_0, a_0)\tag{S153}$$

$$\begin{aligned}R_{t+1} = h_t &= h(\Delta(s_{t-1}, a_{t-1}, s_t, a_t)) = \tilde{q}^\pi(s_t, a_t) - \tilde{q}^\pi(s_{t-1}, a_{t-1})\\ &= p_R(s_t, a_t) - p_R(s_{t-1}, a_{t-1}) \ \text{ for } 0 < t \leqslant T\end{aligned}\tag{S154}$$

$$R_{T+2} = \tilde{R}_{T+1} - \tilde{r}_{T+1} = R - p_R(s_T, a_T) .\tag{S155}$$

The redistributed reward is the change in the success probability. A good action increases the success probability and obtains a positive reward while a bad action reduces the success probability and obtains a negative reward.

### S2.9.2 Optimal Reward Redistribution reduces the MDP to a Stochastic Contextual Bandit Problem

The new SDP $\mathcal{P}$ has a redistributed reward with random variable $R_t$ at time $t$ distributed according to $p(r \mid s_t, a_t)$. Theorem S5 states

$$q^\pi(s_t, a_t) \; = \; r(s_t, a_t) \,. \tag{S156}$$

This equation looks like a contextual bandit problem, where $r(s_t, a_t)$ is an estimate of the mean reward for action $a_t$ for state or context $s_t$. Contextual bandits [36, p. 208] are characterized by a conditionally $\sigma$-subgaussian noise (Def. 5.1 [36, p. 68]). We define the zero mean noise variable $\eta$ by

$$\eta_t \; = \; \eta(s_t, a_t) \; = \; R_t \, - \, r(s_t, a_t) \,, \tag{S157}$$

where we assume that $\eta_t$ is a conditionally $\sigma$-subgaussian noise variable. Therefore, $\eta$ is distributed according to $p(r - r(s_t, a_t) \mid s_t, a_t)$ and fulfills

$$\mathrm{E}\left[\eta(s_t, a_t)\right] \; = \; 0 \,, \tag{S158}$$

$$\mathrm{E}\left[\exp(\lambda \eta(s_t, a_t)\right] \; \leqslant \; \exp(\lambda^2 \sigma^2/2) \,. \tag{S159}$$

Subgaussian random variables have tails that decay almost as fast as a Gaussian. If the reward $r$ is bounded by $|r| < B$, then $\eta$ is bounded by $|\eta| < B$ and, therefore, a $B$-subgaussian. For binary rewards it is of interest that a Bernoulli variable is 0.5-subgaussian [36, p. 71]. In summary, an optimal reward redistribution reduces the MDP to a stochastic contextual bandit problem.

### S2.9.3 Relation to "Backpropagation through a Model"

The relation of reward redistribution if applied to policy gradients and "Backpropagation through a Model" is discussed here. For a delayed reward that is only received at the end of an episode, we decompose the return $\tilde{r}_{T+1}$ into

$$g(\Delta_{0:T}) \; = \; \tilde{r}_{T+1} \; = \; \sum_{t=0}^{T} h(\Delta(s_{t-1}, a_{t-1}, s_t, a_t)) \,. \tag{S160}$$

The policy gradient for an optimal reward redistribution is

$$\mathrm{E}_\pi\left[\nabla_\theta \log \pi(a_t \mid s_t; \boldsymbol{\theta}) \, h(\Delta(s_{t-1}, a_{t-1}, s_t, a_t))\right] \,. \tag{S161}$$

Summing up the gradient for one episode, the gradient becomes

$$\mathrm{E}_\pi\left[\sum_{t=0}^{T} \nabla_\theta \log \pi(a_t \mid s_t; \boldsymbol{\theta}) \, h(\Delta(s_{t-1}, a_{t-1}, s_t, a_t))\right] \tag{S162}$$
$$= \; \mathrm{E}_\pi\left[\boldsymbol{J}_\theta(\log \pi(\boldsymbol{a} \mid \boldsymbol{s}; \boldsymbol{\theta})) \, \boldsymbol{h}(\Delta(\boldsymbol{s}', \boldsymbol{a}', \boldsymbol{s}, \boldsymbol{a}))\right] \,,$$

where $\boldsymbol{a}' = (a_{-1}, a_0, a_1, \ldots, a_{T-1})$ and $\boldsymbol{a} = (a_0, a_1, \ldots, a_T)$ are the sequences of actions, $\boldsymbol{s}' = (s_{-1}, s_0, s_1, \ldots, s_{T-1})$ and $\boldsymbol{s} = (s_0, s_1, \ldots, s_T)$ are the sequences of states, $\boldsymbol{J}_\theta(\log \pi)$ is the Jacobian of the log-probability of the state sequence with respect to the parameter vector $\boldsymbol{\theta}$, and $\boldsymbol{h}(\Delta(\boldsymbol{s}', \boldsymbol{a}', \boldsymbol{s}, \boldsymbol{a}))$ is the vector with entries $h(\Delta(s_{t-1}, a_{t-1}, s_t, a_t))$.

An alternative approach via sensitivity analysis is "Backpropagation through a Model", where $g(\Delta_{0:T})$ is maximized, that is, the return is maximized. Continuous actions are directly fed into $g$ while probabilistic actions are sampled before entering $g$. Analog to gradients used for Restricted Boltzmann Machines, for probabilistic actions the log-likelihood of the actions is used to construct a gradient. The likelihood can also be formulated as the cross-entropy between the sampled actions and the action probability. The gradient for "Backpropagation through a Model" is

$$\mathrm{E}_\pi\left[\boldsymbol{J}_\theta(\log \pi(\boldsymbol{a} \mid \boldsymbol{s}; \boldsymbol{\theta})) \, \nabla_a g(\Delta_{0:T})\right] \,, \tag{S163}$$

where $\nabla_a g(\Delta_{0:T})$ is the gradient of $g$ with respect to the action sequence $\boldsymbol{a}$.

If for "Backpropagation through a Model" the model gradient with respect to actions is replaced by the vector of contributions of actions in the model, then we obtain redistribution applied to policy gradients.

## S3    Bias-Variance Analysis of MDP Q-Value Estimators

Bias-variance investigations have been done for $Q$-learning. Grünewälder & Obermayer [20] investigated the bias of temporal difference learning (TD), Monte Carlo estimators (MC), and least-squares temporal difference learning (LSTD). Mannor et al. [40] and O'Donoghue et al. [50] derived bias and variance expressions for updating $Q$-values.

The true, but unknown, action-value function $q^\pi$ is the expected future return. We assume to have the data $D$, which is a set of state-action sequences with return, that is a set of episodes with return. Using data $D$, $q^\pi$ is estimated by $\hat{q}^\pi = \hat{q}^\pi(D)$, which is an estimate with bias and variance. For bias and variance we have to compute the expectation $\mathrm{E}_D\,[.]$ over the data $D$. The mean squared error (MSE) of an estimator $\hat{q}^\pi(s,a)$ is

$$\mathbf{mse}\,\hat{q}^\pi(s,a) \;=\; \mathrm{E}_D\left[\left(\hat{q}^\pi(s,a) \;-\; q^\pi(s,a)\right)^2\right]\,. \tag{S164}$$

The bias of an estimator $\hat{q}^\pi(s,a)$ is

$$\mathbf{bias}\,\hat{q}^\pi(s,a) \;=\; \mathrm{E}_D\left[\hat{q}^\pi(s,a)\right] \;-\; q^\pi(s,a)\,. \tag{S165}$$

The variance of an estimator $\hat{q}^\pi(s,a)$ is

$$\mathbf{var}\,\hat{q}^\pi(s,a) \;=\; \mathrm{E}_D\left[\left(\hat{q}^\pi(s,a) \;-\; \mathrm{E}_D\left[\hat{q}^\pi(s,a)\right]\right)^2\right]\,. \tag{S166}$$

The bias-variance decomposition of the MSE of an estimator $\hat{q}^\pi(s,a)$ is

$$\mathbf{mse}\,\hat{q}^\pi(s,a) \;=\; \mathbf{var}\,\hat{q}^\pi(s,a) \;+\; \left(\mathbf{bias}\,\hat{q}^\pi(s,a)\right)^2\,. \tag{S167}$$

The bias-variance decomposition of the MSE of an estimator $\hat{q}^\pi$ as a vector is

$$\mathbf{mse}\,\hat{q}^\pi \;=\; \mathrm{E}_D\left[\sum_{s,a}\left(\hat{q}^\pi(s,a) \;-\; q^\pi(s,a)\right)^2\right] \;=\; \mathrm{E}_D\left[\|\hat{q}^\pi \;-\; q^\pi\|^2\right]\,, \tag{S168}$$

$$\mathbf{bias}\,\hat{q}^\pi \;=\; \mathrm{E}_D\left[\hat{q}^\pi\right] \;-\; q^\pi\,, \tag{S169}$$

$$\mathbf{var}\,\hat{q}^\pi \;=\; \mathrm{E}_D\left[\sum_{s,a}\left(\hat{q}^\pi(s,a) \;-\; \mathrm{E}_D\left[\hat{q}^\pi(s,a)\right]\right)^2\right] \;=\; \mathrm{TrVar}_D\left[\hat{q}^\pi\right]\,, \tag{S170}$$

$$\mathbf{mse}\,\hat{q}^\pi \;=\; \mathbf{var}\,\hat{q}^\pi \;+\; \left(\mathbf{bias}\,\hat{q}^\pi\right)^T\mathbf{bias}\,\hat{q}^\pi\,. \tag{S171}$$

### S3.1    Bias-Variance for MC and TD Estimates of the Expected Return

**Monte Carlo (MC)** computes the arithmetic mean $\hat{q}^\pi(s,a)$ of $G_t$ for $(s_t = s, a_t = a)$ over the episodes given by the data.

For **temporal difference (TD)** methods, like SARSA, with learning rate $\alpha$ the updated estimate of $q^\pi(s_t,a_t)$ is:

$$\begin{aligned}(\hat{q}^\pi)^{\mathrm{new}}(s_t,a_t) \;&=\; \hat{q}^\pi(s_t,a_t) \;-\; \alpha\left(\hat{q}^\pi(s_t,a_t) \;-\; R_{t+1} \;-\; \gamma\,\hat{q}^\pi(s_{t+1},a_{t+1})\right)\\ &=\; (1 \;-\; \alpha)\,\hat{q}^\pi(s_t,a_t) \;+\; \alpha\left(R_{t+1} \;+\; \gamma\,\hat{q}^\pi(s_{t+1},a_{t+1})\right)\,.\end{aligned} \tag{S172}$$

Similar updates are used for expected SARSA and $Q$-learning, where only $a_{t+1}$ is chosen differently. Therefore, for the estimation of $\hat{q}^\pi(s_t,a_t)$, SARSA and $Q$-learning perform an exponentially weighted arithmetic mean of $(R_{t+1}+\gamma\hat{q}^\pi(s_{t+1},a_{t+1}))$. If for the updates $\hat{q}^\pi(s_{t+1},a_{t+1})$ is fixed on some data, then SARSA and $Q$-learning perform an exponentially weighted arithmetic mean of the immediate reward $R_{t+1}$ plus averaging over which $\hat{q}^\pi(s_{t+1},a_{t+1})$ (which $(s_{t+1},a_{t+1})$) is chosen. In summary, TD methods like SARSA and $Q$-learning are biased via $\hat{q}^\pi(s_{t+1},a_{t+1})$ and perform an exponentially weighted arithmetic mean of the immediate reward $R_{t+1}$ and the next (fixed) $\hat{q}^\pi(s_{t+1},a_{t+1})$.

**Bias-Variance for Estimators of the Mean.** Both Monte Carlo and TD methods, like SARSA and $Q$-learning, respectively, estimate $q^\pi(s,a) = \mathrm{E}\left[G_t \mid s,a\right]$, which is the expected future return. The expectations are estimated by either an arithmetic mean over samples with Monte Carlo or an exponentially weighted arithmetic mean over samples with TD methods. Therefore, we are interested in computing the bias and variance of these estimators of the expectation. In particular, we consider the arithmetic mean and the exponentially weighted arithmetic mean.

We assume $n$ samples for a state-action pair $(s,a)$. However, the expected number of samples depends on the probabilistic number of visits of $(s,a)$ per episode.

**Arithmetic mean.** For $n$ samples $\{X_1, \ldots, X_n\}$ from a distribution with mean $\mu$ and variance $\sigma^2$, the arithmetic mean, its bias and and its variance are:

$$\hat{\mu}_n = \frac{1}{n} \sum_{i=1}^{n} X_i, \quad \textbf{bias}(\hat{\mu}_n) = 0, \quad \textbf{var}(\hat{\mu}_n) = \frac{\sigma^2}{n}. \tag{S173}$$

The estimation variance of the arithmetic mean is determined by $\sigma^2$, the variance of the distribution the samples are drawn from.

**Exponentially weighted arithmetic mean.** For $n$ samples $\{X_1, \ldots, X_n\}$ from a distribution with mean $\mu$ and variance $\sigma$, the variance of the exponential mean with initial value $\mu_0$ is

$$\hat{\mu}_0 = \mu_0, \quad \hat{\mu}_k = (1 - \alpha)\,\hat{\mu}_{k-1} + \alpha\,X_k, \tag{S174}$$

which gives

$$\hat{\mu}_n = \alpha \sum_{i=1}^{n} (1 - \alpha)^{n-i} X_i + (1 - \alpha)^n \mu_0. \tag{S175}$$

This is a weighted arithmetic mean with exponentially decreasing weights, since the coefficients sum up to one:

$$\alpha \sum_{i=1}^{n} (1 - \alpha)^{n-i} + (1 - \alpha)^n = \alpha \frac{1 - (1 - \alpha)^n}{1 - (1 - \alpha)} + (1 - \alpha)^n \tag{S176}$$
$$= 1 - (1 - \alpha)^n + (1 - \alpha)^n = 1.$$

The estimator $\hat{\mu}_n$ is biased, since:

$$\textbf{bias}(\hat{\mu}_n) = \mathrm{E}\left[\hat{\mu}_n\right] - \mu = \mathrm{E}\left[\alpha \sum_{i=1}^{n} (1 - \alpha)^{n-i} X_i\right] + (1 - \alpha)^n \mu_0 - \mu \tag{S177}$$
$$= \alpha \sum_{i=1}^{n} (1 - \alpha)^{n-i} \mathrm{E}\left[X_i\right] + (1 - \alpha)^n \mu_0 - \mu$$
$$= \mu\,\alpha \sum_{i=0}^{n-1} (1 - \alpha)^i + (1 - \alpha)^n \mu_0 - \mu$$
$$= \mu\,(1 - (1 - \alpha)^n) + (1 - \alpha)^n \mu_0 - \mu = (1 - \alpha)^n\,(\mu_0 - \mu).$$

Asymptotically ($n \to \infty$) the estimate is unbiased. The variance is

$$\textbf{var}(\hat{\mu}_n) \;=\; \mathrm{E}\left[\hat{\mu}_n^2\right] \;-\; \mathrm{E}^2\left[\hat{\mu}_n\right] \tag{S178}$$

$$= \mathrm{E}\left[\alpha^2 \sum_{i=1}^{n}\sum_{j=1}^{n}(1-\alpha)^{n-i}\,X_i\,(1-\alpha)^{n-j}\,X_j\right]$$

$$+ \mathrm{E}\left[2\,(1-\alpha)^n\,\mu_0\,\alpha\,\sum_{i=1}^{n}(1-\alpha)^{n-i}\,X_i\right] \;+\; (1-\alpha)^{2n}\,\mu_0^2$$

$$- \left((1-\alpha)^n\,(\mu_0-\mu)+\mu\right)^2$$

$$= \alpha^2\,\mathrm{E}\left[\sum_{i=1}^{n}(1-\alpha)^{2(n-i)}\,X_i^2 + \sum_{i=1}^{n}\sum_{j=1,j\neq i}^{n}(1-\alpha)^{n-i}\,X_i\,(1-\alpha)^{n-j}\,X_j\right]$$

$$+ 2\,(1-\alpha)^n\,\mu_0\,\mu\,\alpha\,\sum_{i=1}^{n}(1-\alpha)^{n-i} \;+\; (1-\alpha)^{2n}\,\mu_0^2$$

$$- \left((1-\alpha)^n\,\mu_0 + (1-(1-\alpha)^n)\,\mu\right)^2$$

$$= \alpha^2\left(\sum_{i=1}^{n}(1-\alpha)^{2(n-i)}\left(\sigma^2+\mu^2\right) + \sum_{i=1}^{n}\sum_{j=1,j\neq i}^{n}(1-\alpha)^{n-i}\,(1-\alpha)^{n-j}\,\mu^2\right)$$

$$+ 2\,(1-\alpha)^n\,\mu_0\,\mu\,(1-(1-\alpha)^n) + (1-\alpha)^{2n}\,\mu_0^2$$

$$- (1-\alpha)^{2n}\,\mu_0^2 - 2\,(1-\alpha)^n\,\mu_0\,(1-(1-\alpha)^n)\,\mu - (1-(1-\alpha)^n)^2\,\mu^2$$

$$= \sigma^2\,\alpha^2\sum_{i=0}^{n-1}\left((1-\alpha)^2\right)^i + \mu^2\,\alpha^2\left(\sum_{i=0}^{n-1}(1-\alpha)^i\right)^2 - (1-(1-\alpha)^n)^2\,\mu^2$$

$$= \sigma^2\,\alpha^2\,\frac{1-(1-\alpha)^{2n}}{1-(1-\alpha)^2} \;=\; \sigma^2\,\frac{\alpha\,(1-(1-\alpha)^{2n})}{2-\alpha}\;.$$

Also the estimation variance of the exponentially weighted arithmetic mean is proportional to $\sigma^2$, which is the variance of the distribution the samples are drawn from.

The deviation of random variable $X$ from its mean $\mu$ can be analyzed with Chebyshev's inequality. Chebyshev's inequality [8, 81] states that for a random variable $X$ with expected value $\mu$ and variance $\tilde{\sigma}^2$ and for any real number $\epsilon > 0$:

$$\Pr\left[|X-\mu| \geqslant \epsilon\,\tilde{\sigma}\right] \leqslant \frac{1}{\epsilon^2} \tag{S179}$$

or, equivalently,

$$\Pr\left[|X-\mu| \geqslant \epsilon\right] \leqslant \frac{\tilde{\sigma}^2}{\epsilon^2}\;. \tag{S180}$$

For $n$ samples $\{X_1,\ldots,X_n\}$ from a distribution with expectation $\mu$ and variance $\sigma$ we compute the arithmetic mean $\frac{1}{n}\sum_{i=1}^{n}X_i$. If $X$ is the arithmetic mean, then $\tilde{\sigma}^2 = \sigma^2/n$ and we obtain

$$\Pr\left[\left|\frac{1}{n}\sum_{i=1}^{n}X_i - \mu\right| \geqslant \epsilon\right] \leqslant \frac{\sigma^2}{n\,\epsilon^2}\;. \tag{S181}$$

Following Grünewälder and Obermayer [20], Bernstein's inequality can be used to describe the deviation of the arithmetic mean (unbiased estimator of $\mu$) from the expectation $\mu$ (see Theorem 6 of Gábor Lugosi's lecture notes [39]):

$$\Pr\left[\left|\frac{1}{n}\sum_{i=1}^{n}X_i - \mu\right| \geqslant \epsilon\right] \leqslant 2\,\exp\left(-\frac{\epsilon^2\,n}{2\,\sigma^2 + \frac{2\,M\,\epsilon}{3}}\right)\;, \tag{S182}$$

where $|X-\mu| < M$.

## S3.2 Mean and Variance of an MDP Sample of the Return

Since the variance of the estimators of the expectations (arithmetic mean and exponentially weighted arithmetic mean) is governed by the variance of the samples, we compute mean and variance of the return estimate $q^\pi(s,a)$. We follow [73, 79, 80] for deriving the mean and variance.

We consider an MDP with finite horizon $T$, that is, each episode has length $T$. The finite horizon MDP can be generalized to an MDP with absorbing (terminal) state $s = \text{E}$. We only consider proper policies, that is there exists an integer $n$ such that from any initial state the probability of achieving the terminal state E after $n$ steps is strictly positive. $T$ is the time to the first visit of the terminal state: $T = \min k \mid s_k = \text{E}$. The return $G_0$ is:

$$G_0 \;=\; \sum_{k=0}^{T} \gamma^k\, R_{k+1}\;. \tag{S183}$$

The action-value function, the $Q$-function, is the expected return

$$G_t \;=\; \sum_{k=0}^{T-t} \gamma^k\, R_{t+k+1} \tag{S184}$$

if starting in state $S_t = s$ and action $A_t = a$:

$$q^\pi(s,a) \;=\; \text{E}_\pi\left[ G_t \mid s,a \right]\;. \tag{S185}$$

The second moment of the return is:

$$M^\pi(s,a) \;=\; \text{E}_\pi\left[ G_t^2 \mid s,a \right]\;. \tag{S186}$$

The variance of the return is:

$$V^\pi(s,a) \;=\; \text{Var}_\pi\left[ G_t \mid s,a \right] \;=\; M^\pi(s,a) \;-\; \left( q^\pi(s,a) \right)^2\;. \tag{S187}$$

Using $\text{E}_{s',a'}(f(s',a')) = \sum_{s'} p(s' \mid s,a) \sum_{a'} \pi(a' \mid s') f(s',a')$, and analogously $\text{Var}_{s',a'}$ and $\text{Var}_r$, the next Theorem S7 gives mean and variance $V^\pi(s,a) = \text{Var}_\pi\left[ G_t \mid s,a \right]$ of sampling returns from an MDP.

**Theorem S7.** *The mean $q^\pi$ and variance $V^\pi$ of sampled returns from an MDP are*

$$q^\pi(s,a) = \sum_{s',r} p(s',r \mid s,a) \left( r + \gamma \sum_{a'} \pi(a' \mid s') q^\pi(s',a') \right) = r(s,a) + \gamma \text{E}_{s',a'}\left[ q^\pi(s',a') \mid s,a \right],$$

$$V^\pi(s,a) = \text{Var}_r\left[ r \mid s,a \right] + \gamma^2 \left( \text{E}_{s',a'}\left[ V^\pi(s',a') \mid s,a \right] + \text{Var}_{s',a'}\left[ q^\pi(s',a') \mid s,a \right] \right). \tag{S188}$$

*Proof.* The Bellman equation for $Q$-values is

$$q^\pi(s,a) \;=\; \sum_{s',r} p(s',r \mid s,a) \left( r \;+\; \gamma \sum_{a'} \pi(a' \mid s')\, q^\pi(s',a') \right) \tag{S189}$$

$$=\; r(s,a) \;+\; \gamma\, \text{E}_{s',a'}\left[ q^\pi(s',a') \mid s,a \right]\;.$$

This equation gives the mean if drawing one sample. We use

$$r(s,a) \;=\; \sum_r r\, p(r \mid s,a)\;, \tag{S190}$$

$$r^2(s,a) \;=\; \sum_r r^2\, p(r \mid s,a)\;. \tag{S191}$$

For the second moment, we obtain [79]:

$$M^\pi(s,a) = \mathrm{E}_\pi\left[G_t^2 \mid s,a\right] \tag{S192}$$

$$= \mathrm{E}_\pi\left[\left(\sum_{k=0}^{T-t}\gamma^k R_{t+k+1}\right)^2 \mid s,a\right]$$

$$= \mathrm{E}_\pi\left[\left(R_{t+1} + \sum_{k=1}^{T-t}\gamma^k R_{t+k+1}\right)^2 \mid s,a\right]$$

$$= r^2(s,a) + 2\,r(s,a)\,\mathrm{E}_\pi\left[\sum_{k=1}^{T-t}\gamma^k R_{t+k+1} \mid s,a\right]$$

$$+ \mathrm{E}_\pi\left[\left(\sum_{k=1}^{T-t}\gamma^k R_{t+k+1}\right)^2 \mid s,a\right]$$

$$= r^2(s,a) + 2\gamma\,r(s,a)\sum_{s'}p(s' \mid s,a)\sum_{a'}\pi(a' \mid s')\,q^\pi(s',a')$$

$$+ \gamma^2\sum_{s'}p(s' \mid s,a)\sum_{a'}\pi(a' \mid s')\,M^\pi(s',a')$$

$$= r^2(s,a) + 2\gamma\,r(s,a)\,\mathrm{E}_{s',a'}\left[q^\pi(s',a') \mid s,a\right] + \gamma^2\,\mathrm{E}_{s',a'}\left[M^\pi(s',a') \mid s,a\right].$$

For the variance, we obtain:

$$V^\pi(s,a) = M^\pi(s,a) - \left(q^\pi(s,a)\right)^2 \tag{S193}$$

$$= r^2(s,a) - (r(s,a))^2 + \gamma^2\,\mathrm{E}_{s',a'}\left[M^\pi(s',a') \mid s,a\right] - \gamma^2\,\mathrm{E}_{s',a'}^2\left[q^\pi(s',a') \mid s,a\right]$$

$$= \mathrm{Var}_r\left[r \mid s,a\right] + \gamma^2\left(\mathrm{E}_{s',a'}\left[M^\pi(s',a') - \left(q^\pi(s',a')\right)^2 \mid s,a\right]\right.$$

$$\left. - \mathrm{E}_{s',a'}^2\left[q^\pi(s',a') \mid s,a\right] + \mathrm{E}_{s',a'}\left[\left(q^\pi(s',a')\right)^2 \mid s,a\right]\right)$$

$$= \mathrm{Var}_r\left[r \mid s,a\right] + \gamma^2\left(\mathrm{E}_{s',a'}\left[V^\pi(s',a') \mid s,a\right] + \mathrm{Var}_{s',a'}\left[q^\pi(s',a') \mid s,a\right]\right).$$

$$\square$$

For deterministic reward, that is, $\mathrm{Var}_r\left[r \mid s,a\right] = 0$, the corresponding result is given as Equation (4) in Sobel 1982 [73] and as Proposition 3.1 (c) in Tamar et al. 2012 [79].
For temporal difference (TD) learning, the next $Q$-values are fixed to $\hat{q}^\pi(s',a')$ when drawing a sample. Therefore, TD is biased, that is, both SARSA and $Q$-learning are biased. During learning with according updates of $Q$-values, $\hat{q}^\pi(s',a')$ approaches $q^\pi(s',a')$, and the bias is reduced. However, this reduction of the bias is exponentially small in the number of time steps between reward and updated $Q$-values, as we will see later. The reduction of the bias is exponentially small for eligibility traces, too.
The variance recursion Eq. (S188) of sampled returns consists of three parts:

- (1) the immediate variance $\mathrm{Var}_r\left[r \mid s,a\right]$ of the immediate reward stemming from the probabilistic reward $p(r \mid s,a)$,
- (2) the local variance $\gamma^2\mathrm{Var}_{s',a'}\left[q^\pi(s',a') \mid s,a\right]$ from state transitions $p(s' \mid s,a)$ and new actions $\pi(a' \mid s')$,
- (3) the expected variance $\gamma^2\mathrm{E}_{s',a'}\left[V^\pi(s',a') \mid s,a\right]$ of the next $Q$-values.

For different settings the following parts may be zero:

- (1) the immediate variance $\mathrm{Var}_r\left[r \mid s,a\right]$ is zero for deterministic immediate reward,
- (2) the local variance $\gamma^2\mathrm{Var}_{s',a'}\left[q^\pi(s',a') \mid s,a\right]$ is zero for (i) deterministic state transitions and deterministic policy and for (ii) $\gamma = 0$ (only immediate reward),
- (3) the expected variance $\gamma^2\mathrm{E}_{s',a'}\left[V^\pi(s',a') \mid s,a\right]$ of the next $Q$-values is zero for (i) temporal difference (TD) learning, since the next $Q$-values are fixed and set to their current estimates (if just one sample is drawn) and for (ii) $\gamma = 0$ (only immediate reward).

The local variance $\mathrm{Var}_{s',a'}\left[q^\pi(s',a') \mid s,a\right]$ is the variance of a linear combination of $Q$-values weighted by a multinomial distribution $\sum_{s'} p(s' \mid s,a) \sum_{a'} \pi(a' \mid s')$. The local variance is

$$\mathrm{Var}_{s',a'}\left[q^\pi(s',a') \mid s,a\right] = \sum_{s'} p(s' \mid s,a) \sum_{a'} \pi(a' \mid s') \left(q^\pi(s',a')\right)^2 \tag{S194}$$

$$- \left(\sum_{s'} p(s' \mid s,a) \sum_{a'} \pi(a' \mid s') \, q^\pi(s',a')\right)^2 .$$

This result is Equation (6) in Sobel 1982 [73]. Sobel derived these formulas also for finite horizons and an analog formula if the reward depends also on the next state, that is, for $p(r \mid s,a,s')$.
Monte Carlo uses the accumulated future rewards for updates, therefore its variance is given by the recursion in Eq. (S188). TD, however, fixes $q^\pi(s',a')$ to the current estimates $\hat{q}^\pi(s',a')$, which do not change in the current episode. Therefore, TD has $\mathrm{E}_{s',a'}\left[V^\pi(s',a') \mid s,a\right] = 0$ and only the local variance $\mathrm{Var}_{s',a'}\left[q^\pi(s',a') \mid s,a\right]$ is present. For $n$-step TD, the recursion in Eq. (S188) must be applied $(n-1)$ times. Then, the expected next variances are zero since the future reward is estimated by $\hat{q}^\pi(s',a')$.

**Delayed rewards**. For TD and delayed rewards, information on new data is only captured by the last step of an episode that receives a reward. This reward is used to update the estimates of the $Q$-values of the last state $\hat{q}(s_T, a_T)$. Subsequently, the reward information is propagated one step back via the estimates $\hat{q}$ for each sample. The drawn samples (state action sequences) determine where information is propagated back. Therefore, delayed reward introduces a large bias for TD over a long period of time, since the estimates $\hat{q}(s,a)$ need a long time to reach their true $Q$-values.
For Monte Carlo and delayed rewards, the immediate variance $\mathrm{Var}_r\left[r \mid s,a\right] = 0$ except for the last step of the episode. The delayed reward increases the variance of $Q$-values according to Eq. (S188).

**Sample Distribution Used by Temporal Difference and Monte Carlo.** Monte Carlo (MC) sampling uses the true mean and true variance, where the true mean is

$$q^\pi(s,a) = r(s,a) + \gamma \, \mathrm{E}_{s',a'}\left[q^\pi(s',a') \mid s,a\right] \tag{S195}$$

and the true variance is

$$V^\pi(s,a) = \mathrm{Var}_r\left[r \mid s,a\right] + \gamma^2 \left(\mathrm{E}_{s',a'}\left[V^\pi(s',a') \mid s,a\right] + \mathrm{Var}_{s',a'}\left[q^\pi(s',a') \mid s,a\right]\right) . \tag{S196}$$

Temporal difference (TD) methods replace $q^\pi(s',a')$ by $\hat{q}^\pi(s',a')$ which does not depend on the drawn sample. The mean which is used by temporal difference is

$$q^\pi(s,a) = r(s,a) + \gamma \, \mathrm{E}_{s',a'}\left[\hat{q}^\pi(s',a') \mid s,a\right] . \tag{S197}$$

This mean is biased by

$$\gamma \left(\mathrm{E}_{s',a'}\left[\hat{q}^\pi(s',a') \mid s,a\right] - \mathrm{E}_{s',a'}\left[q^\pi(s',a') \mid s,a\right]\right) . \tag{S198}$$

The variance used by temporal difference is

$$V^\pi(s,a) = \mathrm{Var}_r\left[r \mid s,a\right] + \gamma^2 \, \mathrm{Var}_{s',a'}\left[\hat{q}^\pi(s',a') \mid s,a\right] , \tag{S199}$$

since $V^\pi(s',a') = 0$ if $\hat{q}^\pi(s',a')$ is used instead of the future reward of the sample. The variance of TD is smaller than for MC, since variances are not propagated back.

### S3.3 TD corrects Bias exponentially slowly with Respect to Reward Delay

**Temporal Difference.** We show that TD updates for delayed rewards are exponentially small, fading exponentially with the number of delay steps. $Q$-learning with learning rates $1/i$ at the $i$th update leads to an arithmetic mean as estimate, which was shown to be exponentially slow [4]. If for a fixed learning rate the agent always travels along the same sequence of states, then TD is superquadratic [4]. We, however, consider the general case where the agent travels along random sequences due to a random environment or due to exploration. For a fixed learning rate, the information of the delayed reward has to be propagated back either through the Bellman error or via eligibility traces. We first consider backpropagation of reward information via the Bellman error. For each episode the reward information is propagated back one step at visited state-action pairs via the TD update rule. We denote the $Q$-values of episode $i$ as $q^i$ and assume that the state action pairs

$(s_t, a_t)$ are the most visited ones. We consider the update of $q^i(s_t, a_t)$ of a state-action pair $(s_t, a_t)$ that is visited at time $t$ in the $i$th episode:

$$q^{i+1}(s_t, a_t) \; = \; q^i(s_t, a_t) \; + \; \alpha \, \delta_t \; , \tag{S200}$$

$$\delta_t \; = \; r_{t+1} \; + \; \max_{a'} q^i(s_{t+1}, a') \; - \; q^i(s_t, a_t) \quad \text{(Q-learning)} \tag{S201}$$

$$\delta_t \; = \; r_{t+1} \; + \; \sum_{a'} \pi(a' \mid s_{t+1}) \, q^i(s_{t+1}, a') \; - \; q^i(s_t, a_t) \quad \text{(expected SARSA)} \; . \tag{S202}$$

**Temporal Difference with Eligibility Traces.**   Eligibility traces have been introduced to propagate back reward information of an episode and are now standard for TD($\lambda$) [72]. However, the eligibility traces are exponentially decaying when propagated back. The accumulated trace is defined as [72]:

$$e_{t+1}(s, a) \; = \; \begin{cases} \gamma \, \lambda \, e_t(s, a) & \text{for } s \neq s_t \text{ or } a \neq a_t \; , \\ \gamma \, \lambda \, e_t(s, a) \; + \; 1 & \text{for } s = s_t \text{ and } a = a_t \; , \end{cases} \tag{S203}$$

while the replacing trace is defined as [72]:

$$e_{t+1}(s, a) \; = \; \begin{cases} \gamma \, \lambda \, e_t(s, a) & \text{for } s \neq s_t \text{ or } a \neq a_t \; , \\ 1 & \text{for } s = s_t \text{ and } a = a_t \; . \end{cases} \tag{S204}$$

With eligibility traces using $\lambda \in [0, 1]$, the $\lambda$-return $G_t^\lambda$ is [78]

$$G_t^\lambda \; = \; (1 - \lambda) \sum_{n=1}^{\infty} \lambda^{n-1} \, G_t^{(n)} \; , \tag{S205}$$

$$G_t^{(n)} \; = \; r_{t+1} \; + \; \gamma \, r_{t+2} \; + \; \ldots \; + \; \gamma^{n-1} r_{t+n} \; + \; \gamma^{n-1} \, V(s_{t+n}) \; . \tag{S206}$$

We obtain

$$
\begin{aligned}
G_t^\lambda \; &= \; (1 - \lambda) \sum_{n=1}^{\infty} \lambda^{n-1} \, G_t^{(n)} \\
&= \; (1 - \lambda) \left( r_{t+1} \; + \; \gamma \, V(s_{t+1}) \; + \; \sum_{n=2}^{\infty} \lambda^{n-1} \, G_t^{(n)} \right) \\
&= \; (1 - \lambda) \left( r_{t+1} \; + \; \gamma \, V(s_{t+1}) \; + \; \sum_{n=1}^{\infty} \lambda^{n} \, G_t^{(n+1)} \right) \\
&= \; (1 - \lambda) \left( r_{t+1} \; + \; \gamma \, V(s_{t+1}) \; + \; \lambda \, \gamma \sum_{n=1}^{\infty} \lambda^{n-1} \, G_{t+1}^{(n)} \; + \; \sum_{n=1}^{\infty} \lambda^{n} \, r_{t+1} \right) \\
&= \; (1 - \lambda) \sum_{n=0}^{\infty} \lambda^{n} \, r_{t+1} \; + \; (1 - \lambda)\gamma \, V(s_{t+1}) \; + \; \lambda \, \gamma \, G_{t+1}^\lambda \\
&= \; r_{t+1} \; + \; (1 - \lambda)\gamma \, V(s_{t+1}) \; + \; \lambda \, \gamma \, G_{t+1}^\lambda \; .
\end{aligned}
\tag{S207}
$$

We use the naive $Q(\lambda)$, where eligibility traces are not set to zero. In contrast, Watkins' $Q(\lambda)$ [87] zeros out eligibility traces after non-greedy actions, that is, if not the $\max_a$ is chosen. Therefore, the decay is even stronger for Watkin's $Q(\lambda)$. Another eligibility trace method is Peng's $Q(\lambda)$ [52] which also does not zero out eligibility traces.

The next Theorem S8 states that the decay of TD is exponential for $Q$-value updates in an MDP with delayed reward, even for eligibility traces. Thus, for delayed rewards TD requires exponentially many updates to correct the bias, where the number of updates is exponential in the delay steps.

**Theorem S8.** *For initialization $q^0(s_t, a_t) = 0$ and delayed reward with $r_t = 0$ for $t \leqslant T$, $q(s_{T-i}, a_{T-i})$ receives its first update not earlier than at episode $i$ via $q^i(s_{T-i}, a_{T-i}) = \alpha^{i+1} r_{T+1}^1$, where $r_{T+1}^1$ is the reward of episode 1. Eligibility traces with $\lambda \in [0, 1)$ lead to an exponential decay of $(\gamma\lambda)^k$ when the reward is propagated $k$ steps back.*

*Proof.* If we assume that $Q$-values are initialized with zero, then $q^0(s_t, a_t) = 0$ for all $(s_t, a_t)$. For delayed rewards we have $r_t = 0$ for $t \leqslant T$. The $Q$-value $q(s_{T-i}, a_{T-i})$ at time $T - i$ can receive an update for the first time at episode $i$. Since all $Q$-values have been initialized with zero, the update is

$$q^i(s_{T-i}, a_{T-i}) \; = \; \alpha^{i+1} \, r_{T+1}^1 \; , \tag{S208}$$

where $r_{T+1}^1$ is the reward at time $T + 1$ for episode 1.

We move on to eligibility traces, where the update for a state $s$ is

$$q_{t+1}(s, a) = q_t(s, a) + \alpha \, \delta_t \, e_t(s, a) \,, \tag{S209}$$

$$\delta_t = r_{t+1} + \max_{a'} q_t(s_{t+1}, a') - q_t(s_t, a_t) \,. \tag{S210}$$

If states are not revisited, the eligiblity trace at time $t + k$ for a visit of state $s_t$ at time $t$ is:

$$e_{t+k}(s_t, a_t) = (\gamma \, \lambda)^k \,. \tag{S211}$$

If all $\delta_{t+i}$ are zero except for $\delta_{t+k}$, then the update of $q(s, a)$ is

$$q_{t+k+1}(s, a) = q_{t+k}(s, a) + \alpha \, \delta_{t+k} \, e_{t+k}(s, a) = q_{t+k}(s, a) + \alpha \, (\gamma \, \lambda)^k \, \delta_{t+k} \,. \tag{S212}$$

$\square$

A learning rate of $\alpha = 1$ does not work since it would imply to forget all previous learned estimates, and therefore no averaging over episodes would exist. Since $\alpha < 1$, we observe exponential decay backwards in time for online updates.

### S3.4 MC affects the Variance of Exponentially Many Estimates with Delayed Reward

The variance for Monte Carlo is

$$V^\pi(s, a) = \mathrm{Var}_r \left[ r \mid s, a \right] + \gamma^2 \left( \mathrm{E}_{s',a'} \left[ V^\pi(s', a') \mid s, a \right] + \mathrm{Var}_{s',a'} \left[ q^\pi(s', a') \mid s, a \right] \right) \,. \tag{S213}$$

This is a Bellman equation of the variance. For undiscounted reward $\gamma = 1$, we obtain

$$V^\pi(s, a) = \mathrm{Var}_r \left[ r \mid s, a \right] + \mathrm{E}_{s',a'} \left[ V^\pi(s', a') \mid s, a \right] + \mathrm{Var}_{s',a'} \left[ q^\pi(s', a') \mid s, a \right] \,. \tag{S214}$$

If we define the "on-site" variance $\omega$ as

$$\omega(s, a) = \mathrm{Var}_r \left[ r \mid s, a \right] + \mathrm{Var}_{s',a'} \left[ q^\pi(s', a') \mid s, a \right] \,, \tag{S215}$$

we get

$$V^\pi(s, a) = \omega(s, a) + \mathrm{E}_{s',a'} \left[ V^\pi(s', a') \mid s, a \right] \,. \tag{S216}$$

This is the solution of the general formulation of the Bellman operator. The Bellman operator is defined component-wise for any variance $V$ as

$$\mathrm{T}^\pi \left[ V \right] (s, a) = \omega(s, a) + \mathrm{E}_{s',a'} \left[ V(s', a') \mid s, a \right] \,. \tag{S217}$$

For proper policies $\pi$ a unique fixed point $V^\pi$ exists:

$$V^\pi = \mathrm{T}^\pi \left[ V^\pi \right] \tag{S218}$$

$$V^\pi = \lim_{k \to \infty} \left( \mathrm{T}^\pi \right)^k V \,, \tag{S219}$$

where $V$ is any initial variance. The operator $\mathrm{T}^\pi$ is continuous, monotonically increasing (component-wise larger or smaller), and a contraction mapping for a weighted sup-norm. If we define the operator $\mathrm{T}^\pi$ as depending on the on-site variance $\omega$, that is $\mathrm{T}_\omega^\pi$, then it is monotonically in $\omega$. We obtain component-wise for $\omega > \tilde\omega$:

$$\mathrm{T}_\omega^\pi \left[ q \right] (s, a) - \mathrm{T}_{\tilde\omega}^\pi \left[ q \right] (s, a) \tag{S220}$$
$$= \left( \omega(s, a) + \mathrm{E}_{s',a'} \left[ q(s', a') \right] \right) - \left( \tilde\omega(s, a) + \mathrm{E}_{s',a'} \left[ q(s', a') \right] \right)$$
$$= \omega(s, a) - \tilde\omega(s, a) \geqslant 0 \,.$$

It follows for the fixed points $V^\pi$ of $\mathrm{T}_\omega^\pi$ and $\widetilde V^\pi$ of $\mathrm{T}_{\tilde\omega}^\pi$:

$$V^\pi(s, a) \geqslant \widetilde V^\pi(s, a) \,. \tag{S221}$$

Therefore if

$$\omega(s, a) = \mathrm{Var}_r \left[ r \mid s, a \right] + \mathrm{Var}_{s',a'} \left[ q^\pi(s', a') \mid s, a \right] \geqslant \tag{S222}$$

$$\tilde\omega(s, a) = \widetilde{\mathrm{Var}}_r \left[ r \mid s, a \right] + \widetilde{\mathrm{Var}}_{s',a'} \left[ q^\pi(s', a') \mid s, a \right]$$

then

$$V^\pi(s, a) \geqslant \widetilde V^\pi(s, a) \,. \tag{S223}$$

**Theorem S9.** *Starting from the sequence end at $t = T$, as long as $\omega(s_t, a_t) \geqslant \tilde{\omega}(s_t, a_t)$ holds also the following holds:*

$$V(s_t, a_t) \geqslant \widetilde{V}(s_t, a_t) . \tag{S224}$$

*If for $(s_t, a_t)$ the strict inequality $\omega(s_t, a_t) > \tilde{\omega}(s_t, a_t)$ holds, then we have the strict inequality*

$$V(s_t, a_t) > \widetilde{V}(s_t, a_t) . \tag{S225}$$

*If $p(s_t, a_t \mid s_{t-1}, a_{t-1}) \neq 0$ for some $(s_{t-1}, a_{t-1})$ then*

$$\mathrm{E}_{s_t, a_t} \left[ V(s_t, a_t) \mid s_{t-1}, a_{t-1} \right] > \mathrm{E}_{s_t, a_t} \left[ \widetilde{V}(s_t, a_t) \mid s_{t-1}, a_{t-1} \right] . \tag{S226}$$

*Therefore, the strict inequality $\omega(s_t, a_t) > \tilde{\omega}(s_t, a_t)$ is propagated back as a strict inequality of variances.*

*Proof.* Proof by induction: Induction base: $V(s_{T+1}, a_{T+1}) = \widetilde{V}(s_{T+1}, a_{T+1}) = 0$ and $\omega(s_T, a_T) = \tilde{\omega}(s_T, a_T) = 0$.
Induction step $((t+1) \to t)$: The induction hypothesis is that for all $(s_{t+1}, a_{t+1})$ we have

$$V(s_{t+1}, a_{t+1}) \geqslant \widetilde{V}(s_{t+1}, a_{t+1}) \tag{S227}$$

and $\omega(s_t, a_t) \geqslant \tilde{\omega}(s_t, a_t)$. It follows that

$$\mathrm{E}_{s_{t+1}, a_{t+1}} \left[ V(s_{t+1}, a_{t+1}) \right] \geqslant \mathrm{E}_{s_{t+1}, a_{t+1}} \left[ \widetilde{V}(s_{t+1}, a_{t+1}) \right] . \tag{S228}$$

We obtain

$$V(s_t, a_t) - \widetilde{V}(s_t, a_t) \tag{S229}$$

$$= \left( \omega(s_t, a_t) + \mathrm{E}_{s_{t+1}, a_{t+1}} \left[ V(s_{t+1}, a_{t+1}) \right] \right) - \left( \tilde{\omega}(s_t, a_t) + \mathrm{E}_{s_{t+1}, a_{t+1}} \left[ \widetilde{V}(s_{t+1}, a_{t+1}) \right] \right)$$

$$= \omega(s_t, a_t) - \tilde{\omega}(s_t, a_t) \geqslant 0 .$$

If for $(s_t, a_t)$ the strict inequality $\omega(s_t, a_t) > \tilde{\omega}(s_t, a_t)$ holds, then we have the strict inequality $V(s_t, a_t) > \widetilde{V}(s_t, a_t)$. If $p(s_t, a_t \mid s_{t-1}, a_{t-1}) \neq 0$ for some $(s_{t-1}, a_{t-1})$ then

$$\mathrm{E}_{s_t, a_t} \left[ V(s_t, a_t) \mid s_{t-1}, a_{t-1} \right] > \mathrm{E}_{s_t, a_t} \left[ \widetilde{V}(s_t, a_t) \mid s_{t-1}, a_{t-1} \right] . \tag{S230}$$

Therefore, the strict inequality $\omega(s_t, a_t) > \tilde{\omega}(s_t, a_t)$ is propagated back as a strict inequality of variances as long as $p(s_t, a_t \mid s_{t-1}, a_{t-1}) \neq 0$ for some $(s_{t-1}, a_{t-1})$.
The induction goes through as long as $\omega(s_t, a_t) \geqslant \tilde{\omega}(s_t, a_t)$. $\qquad \square$

In Stephen Patek's PhD thesis, [51] Lemma 5.1 on page 88-89 and proof thereafter state that if $\tilde{\omega}(s, a) = \omega(s, a) - \lambda$, then the solution $\widetilde{V}^\pi$ is continuous and decreasing in $\lambda$. From the inequality above it follows that

$$V^\pi(s, a) - \widetilde{V}^\pi(s, a) = (\mathrm{T}_\omega^\pi V^\pi)(s, a) - \left( \mathrm{T}_{\tilde{\omega}}^\pi \widetilde{V}^\pi \right)(s, a) \tag{S231}$$

$$= \omega(s, a) - \tilde{\omega}(s, a) + \mathrm{E}_{s', a'} \left[ V^\pi(s', a') - \widetilde{V}^\pi(s', a') \mid s, a \right]$$

$$\geqslant \omega(s, a) - \tilde{\omega}(s, a) .$$

**Time-Agnostic States.** We defined a Bellman operator as

$$\mathrm{T}^\pi \left[ \boldsymbol{V}^\pi \right](s, a) = \omega(s, a) + \sum_{s'} p(s' \mid s, a) \sum_{a'} \pi(a' \mid s') V^\pi(s', a') \tag{S232}$$

$$= \omega(s, a) + (\boldsymbol{V}^\pi)^T \boldsymbol{p}(s, a) ,$$

where $\boldsymbol{V}^\pi$ is the vector with value $V^\pi(s', a')$ at position $(s', a')$ and $\boldsymbol{p}(s, a)$ is the vector with value $p(s' \mid s, a)\pi(a' \mid s')$ at position $(s', a')$. The fixed point equation is known as the *Bellman equation*. In vector and matrix notation the Bellman equation reads

$$\mathrm{T}^\pi \left[ \boldsymbol{V}^\pi \right] = \boldsymbol{\omega} + \boldsymbol{P} \boldsymbol{V}^\pi , \tag{S233}$$

where $\boldsymbol{P}$ is the row-stochastic matrix with $p(s' \mid s, a)\pi(a' \mid s')$ at position $((s, a), (s', a'))$. We assume that the set of state-actions $\{(s, a)\}$ is equal to the set of next state-actions $\{(s', a')\}$, therefore $\boldsymbol{P}$ is a square row-stochastic matrix. This Bellman operator has the same characteristics as the Bellman operator for the action-value function $q^\pi$.

Since $\boldsymbol{P}$ is a row-stochastic matrix, the Perron-Frobenius theorem says that (1) $\boldsymbol{P}$ has as largest eigenvalue 1 for which the eigenvector corresponds to the steady state and (2) the absolute value of each (complex) eigenvalue is smaller equal 1. Only the eigenvector to eigenvalue 1 has purely positive real components. Equation 7 of Bertsekas and Tsitsiklis, 1991, [7] states that

$$(\mathrm{T}^\pi)^t [\boldsymbol{V}^\pi] = \sum_{k=0}^{t-1} \boldsymbol{P}^k \, \boldsymbol{\omega} \, + \, \boldsymbol{P}^t \, \boldsymbol{V}^\pi \, . \tag{S234}$$

Applying the operator $\mathrm{T}^\pi$ recursively $t$ times can be written as [7]:

$$(\mathrm{T}^\pi)^t [\boldsymbol{V}^\pi] = \sum_{k=0}^{t-1} \boldsymbol{P}^k \, \boldsymbol{\omega} \, + \, \boldsymbol{P}^t \, \boldsymbol{V}^\pi \, . \tag{S235}$$

In particular for $\boldsymbol{V}^\pi = \boldsymbol{0}$, we obtain

$$(\mathrm{T}^\pi)^t [\boldsymbol{0}] = \sum_{k=0}^{t-1} \boldsymbol{P}^k \, \boldsymbol{\omega} \, . \tag{S236}$$

For finite horizon MDPs, the values $\boldsymbol{V}^\pi = \boldsymbol{0}$ are correct for time step $T + 1$ since no reward for $t > T + 1$ exists. Therefore, the "backward induction algorithm" [55, 56] gives the correct solution:

$$\boldsymbol{V}^\pi = (\mathrm{T}^\pi)^T [\boldsymbol{0}] = \sum_{k=0}^{T-1} \boldsymbol{P}^k \, \boldsymbol{\omega} \, . \tag{S237}$$

The product of square stochastic matrices is a stochastic matrix, therefore $\boldsymbol{P}^k$ is a stochastic matrix. Perron-Frobenius theorem states that the spectral radius $\mathcal{R}(\boldsymbol{P}^k)$ of the stochastic matrix $\boldsymbol{P}^k$ is: $\mathcal{R}(\boldsymbol{P}^k) = 1$. Furthermore, the largest eigenvalue is 1 and all eigenvalues have absolute values smaller or equal one. Therefore, $\boldsymbol{\omega}$ can have large influence on $\boldsymbol{V}^\pi$ at every time step.

**Time-Aware States.** Next we consider time-aware MDPs, where transitions occur only from states $s_t$ to $s_{t+1}$. The transition matrix from states $s_t$ to $s_{t+1}$ is denoted by $\boldsymbol{P}_t$. We assume that $\boldsymbol{P}_t$ are row-stochastic matrices which are rectangular, that is $\boldsymbol{P}_t \in \mathbb{R}^{m \times n}$.

**Definition S12.** *A row-stochastic matrix $\boldsymbol{A} \in \mathbb{R}^{m \times n}$ has non-negative entries and the entries of each row sum up to one.*

It is known that the product of square stochastic matrices $\boldsymbol{A} \in \mathbb{R}^{n \times n}$ is a stochastic matrix. We show in next theorem that this holds also for rectangular matrices.

**Lemma S4.** *The product $\boldsymbol{C} = \boldsymbol{A}\boldsymbol{B}$ with $\boldsymbol{C} \in \mathbb{R}^{m \times k}$ of a row-stochastic matrix $\boldsymbol{A} \in \mathbb{R}^{m \times n}$ and a row-stochastic matrix $\boldsymbol{B} \in \mathbb{R}^{n \times k}$ is row-stochastic.*

*Proof.* All entries of $\boldsymbol{C}$ are non-negative since they are sums and products of non-negative entries of $\boldsymbol{A}$ and $\boldsymbol{B}$. The row-entries of $\boldsymbol{C}$ sum up to one:

$$\sum_k C_{ik} = \sum_k \sum_j A_{ij} \, B_{jk} = \sum_j A_{ij} \sum_k B_{jk} = \sum_j A_{ij} = 1 \, . \tag{S238}$$

$\square$

We will use the $\infty$-norm and the 1-norm of a matrix, which are defined based on the $\infty$-norm $\|\boldsymbol{x}\|_\infty = \max_i |x_i|$ and 1-norm $\|\boldsymbol{x}\|_1 = \sum_i |x_i|$ of a vector $\boldsymbol{x}$.

**Definition S13.** *The $\infty$-norm of a matrix is the maximum absolute row sum:*

$$\|\boldsymbol{A}\|_\infty = \max_{\|\boldsymbol{x}\|_\infty = 1} \|\boldsymbol{A}\,\boldsymbol{x}\|_\infty = \max_i \sum_j |A_{ij}| \, . \tag{S239}$$

*The 1-norm of a matrix is the maximum absolute column sum:*

$$\|\boldsymbol{A}\|_1 = \max_{\|\boldsymbol{x}\|_1 = 1} \|\boldsymbol{A}\,\boldsymbol{x}\|_1 = \max_j \sum_i |A_{ij}| \, . \tag{S240}$$

The statements of next theorem are known as Perron-Frobenius theorem for square stochastic matrices $\boldsymbol{A} \in \mathbb{R}^{n \times n}$, e.g. that the spectral radius $\mathcal{R}$ is $\mathcal{R}(\boldsymbol{A}) = 1$. We extend the theorem to a "$\infty$-norm equals one" property for rectangular stochastic matrices $\boldsymbol{A} \in \mathbb{R}^{m \times n}$.

**Lemma S5** (Perron-Frobenius). *If $\boldsymbol{A} \in \mathbb{R}^{m \times n}$ is a row-stochastic matrix, then*

$$\|\boldsymbol{A}\|_\infty = 1 , \quad \|\boldsymbol{A}^T\|_1 = 1 , \text{ and for } n = m \quad \mathcal{R}(\boldsymbol{A}) = 1 . \tag{S241}$$

*Proof.* $\boldsymbol{A} \in \mathbb{R}^{m \times n}$ is a row-stochastic matrix, therefore $A_{ij} = |A_{ij}|$. Furthermore, the rows of $\boldsymbol{A}$ sum up to one. Thus, $\|\boldsymbol{A}\|_\infty = 1$. Since the column sums of $\boldsymbol{A}^T$ are the row sums of $\boldsymbol{A}$, it follows that $\|\boldsymbol{A}^T\|_1 = 1$.

For square stochastic matrices, that is $m = n$, Gelfand's Formula (1941) says that for any matrix norm $\|.\|$, for the spectral norm $\mathcal{R}(\boldsymbol{A})$ of a matrix $\boldsymbol{A} \in \mathbb{R}^{n \times n}$ we obtain:

$$\mathcal{R}(\boldsymbol{A}) = \lim_{k \to \infty} \|\boldsymbol{A}^k\|^{1/k} . \tag{S242}$$

Since the product of row-stochastic matrices is a row-stochastic matrix, $\boldsymbol{A}^k$ is a row-stochastic matrix. Consequently $\|\boldsymbol{A}^k\|_\infty = 1$ and $\|\boldsymbol{A}^k\|_\infty^{1/k} = 1$. Therefore, the spectral norm $\mathcal{R}(\boldsymbol{A})$ of a row-stochastic matrix $\boldsymbol{A} \in \mathbb{R}^{n \times n}$ is

$$\mathcal{R}(\boldsymbol{A}) = 1 . \tag{S243}$$

The last statement follows from Perron-Frobenius theorem, which says that the spectral radius of $\boldsymbol{P}$ is 1. $\qquad\square$

Using random matrix theory, we can guess how much the spectral radius of a rectangular matrix deviates from that of a square matrix. Let $\boldsymbol{A} \in \mathbb{R}^{m \times n}$ be a matrix whose entries are independent copies of some random variable with zero mean, unit variance, and finite fourth moment. The Marchenko-Pastur quarter circular law for rectangular matrices says that for $n = m$ the maximal singular value is $2\sqrt{m}$ [41]. Asymptotically we have for the maximal singular value $s_{\max}(\boldsymbol{A}) \propto \sqrt{m} + \sqrt{n}$ [62]. A bound on the largest singular value is given by [74]:

$$s_{\max}^2(\boldsymbol{A}) \leqslant (\sqrt{m} + \sqrt{n})^2 + O(\sqrt{n} \log(n)) \text{ a.s.} \tag{S244}$$

Therefore, a rectangular matrix modifies the largest singular value by a factor of $a = 0.5(1 + \sqrt{n/m})$ compared to a $m \times m$ square matrix. In the case that tstates are time aware, transitions only occur from states $s_t$ to $s_{t+1}$. The transition matrix from states $s_t$ to $s_{t+1}$ is denoted by $\boldsymbol{P}_t$.

**States affected by the on-site variance $\boldsymbol{\omega}_k$ (reachable states).** Typically, states in $s_t$ have only few predecessor states in $s_{t-1}$ compared to $N_{t-1}$, the number of possible states in $s_{t-1}$. Only for those states in $s_{t-1}$ the transition probability to the state in $s_t$ is larger than zero. That is, each $i \in s_{t+1}$ has only few $j \in s_t$ for which $p_t(i \mid j) > 0$. We now want to know how many states have increased variance due to $\boldsymbol{\omega}_k$, that is how many states are affected by $\boldsymbol{\omega}_k$. In a general setting, we assume random connections.

Let $N_t$ be the number of all states $s_t$ that are reachable after $t$ time steps of an episode. $\bar{N} = 1/k \sum_{t=1}^k N_t$ is the arithmetic mean of $N_t$. Let $c_t$ be the average connectivity of a state in $s_t$ to states in $s_{t-1}$ and $\bar{c} = \left(\prod_{t=1}^k c_t\right)^{1/k}$ the geometric mean of the $c_t$. Let $n_t$ be the number of states in $s_t$ that are affected by the on-site variance $\boldsymbol{\omega}_k$ at time $k$ for $t \leqslant k$. The number of states affected by $\boldsymbol{\omega}_k$ is $a_k = \sum_{t=0}^k n_t$. We assume that $\boldsymbol{\omega}_k$ only has one component larger than zero, that is, only one state at time $t = k$ is affected: $n_k = 1$. The number of affected edges from $s_t$ to $s_{t-1}$ is $c_t n_t$. However, states in $s_{t-1}$ may be affected multiple times by different affected states in $s_t$. Figure S1 shows examples of how affected states affect states in a previous time step. The left panel shows no overlap since affected states in $s_{t-1}$ connect only to one affected state in $s_t$. The right panel shows some overlap since affected states in $s_{t-1}$ connect to multiple affected states in $s_t$.

The next theorem states that the on-site variance $\boldsymbol{\omega}_k$ can have large effect on the variance of each previous state-action pair. Furthermore, for small $k$ the number of affected states grows exponentially, while for large $k$ it grows only linearly after some time $\hat{t}$. Figure S2 shows the function which determines how much $a_k$ grows with $k$.

**Theorem S10.** *For $t \leqslant k$, $\boldsymbol{\omega}_k$ contributes to $\boldsymbol{V}_t^\pi$ by the term $\boldsymbol{P}_{t \leftarrow k} \boldsymbol{\omega}_k$, where $\|\boldsymbol{P}_{t \leftarrow k}\|_\infty = 1$. The number $a_k$ of states affected by the on-site variance $\boldsymbol{\omega}_k$ is*

$$a_k = \sum_{t=0}^k \left(1 - \left(1 - \frac{c_t}{N_{t-1}}\right)^{n_t}\right) N_{t-1} . \tag{S245}$$

Figure S1: Examples of how affected states (cyan) affect states in a previous time step (indicated by cyan edges) starting with $n_5 = 1$ (one affected state). The left panel shows no overlap since affected states in $s_{t-1}$ connect only to one affected state in $s_t$. The right panel shows some overlap since affected states in $s_{t-1}$ connect to multiple affected states in $s_t$.

*Proof.* The "backward induction algorithm" [55, 56] gives with $\boldsymbol{V}_{T+1}^{\pi} = \boldsymbol{0}$ and on-site variance $\boldsymbol{\omega}_{T+1} = \boldsymbol{0}$:

$$\boldsymbol{V}_t^{\pi} = \sum_{k=t}^{T} \prod_{\tau=t}^{k-1} \boldsymbol{P}_{\tau}\,\boldsymbol{\omega}_k \,, \tag{S246}$$

where we define $\prod_{\tau=t}^{t-1} \boldsymbol{P}_{\tau} = \boldsymbol{I}$ and $[\boldsymbol{\omega}_k]_{(s_k,a_k)} = \omega(s_k,a_k)$.
Since the product of two row-stochastic matrices is a row-stochastic matrix according to Lemma S4, $\boldsymbol{P}_{t\leftarrow k} = \prod_{\tau=t}^{k-1} \boldsymbol{P}_{\tau}$ is a row-stochastic matrix. Since $\|\boldsymbol{P}_{t\leftarrow k}\|_{\infty} = 1$ according to Lemma S5, each on-site variance $\boldsymbol{\omega}_k$ with $t \leqslant k$ can have large effects on $\boldsymbol{V}_t^{\pi}$. Using the row-stochastic matrices $\boldsymbol{P}_{t\leftarrow k}$, we can reformulate the variance:

$$\boldsymbol{V}_t^{\pi} = \sum_{k=t}^{T} \boldsymbol{P}_{t\leftarrow k}\,\boldsymbol{\omega}_k \,, \tag{S247}$$

with $\|\boldsymbol{P}_{t\leftarrow k}\|_{\infty} = 1$. The on-site variance $\boldsymbol{\omega}_k$ at step $k$ increases all variances $\boldsymbol{V}_t^{\pi}$ with $t \leqslant k$.
Next we proof the second part of the theorem, which considers the growth of $a_k$. To compute $a_k$ we first have to know $n_t$. For computing $n_{t-1}$ from $n_t$, we want to know how many states are affected in $s_{t-1}$ if $n_t$ states are affected in $s_t$. The answer to this question is the expected coverage when searching a document collection using a set of independent computers [11]. We follow the approach of Cox et al. [11]. The minimal number of affected states in $s_{t-1}$ is $c_t$, where each of the $c_t$ affected states in $s_{t-1}$ connects to each of the $n_t$ states in $s_t$ (maximal overlap). The maximal number of affected states in $s_{t-1}$ is $c_t n_t$, where each affected state in $s_{t-1}$ connects to only one affected state in $s_t$ (no overlap). We consider a single state in $s_t$. The probability of a state in $s_{t-1}$ being connected to this single state in $s_t$ is $c_t/N_{t-1}$ and being not connected to this state in $s_t$ is $1 - c_t/N_{t-1}$. The probability of a state in $s_{t-1}$ being not connected to any of the $n_t$ affected states in $s_t$ is

$$\left(1 - \frac{c_t}{N_{t-1}}\right)^{n_t} \,. \tag{S248}$$

The probability of a state in $s_{t-1}$ being at least connected to one of the $n_t$ affected states in $s_t$ is

$$1 - \left(1 - \frac{c_t}{N_{t-1}}\right)^{n_t} \,. \tag{S249}$$

Figure S2: The function $\left(1 - \left(1 - \frac{c_t}{N_{t-1}}\right)^{n_t}\right)$ which scales $N_{t-1}$ in Theorem S10. This function determines the growth of $a_k$, which is exponentially at the beginning, and then linearly when the function approaches 1.

Thus, the expected number of distinct states in $s_{t-1}$ being connected to one of the $n_t$ affected states in $s_t$ is

$$n_{t-1} = \left(1 - \left(1 - \frac{c_t}{N_{t-1}}\right)^{n_t}\right) N_{t-1} . \tag{S250}$$

The number $a_k$ of affected states by $\boldsymbol{\omega}_k$ is

$$a_k = \sum_{t=0}^{k} \left(1 - \left(1 - \frac{c_t}{N_{t-1}}\right)^{n_t}\right) N_{t-1} . \tag{S251}$$

$\square$

**Corollary S2.** *For small $k$, the number $a_k$ of states affected by the on-site variance $\boldsymbol{\omega}_k$ at step $k$ grows exponentially with $k$ by a factor of $\bar{c}$:*

$$a_k > \bar{c}^k . \tag{S252}$$

*For large $k$ and after some time $t > \hat{t}$, the number $a_k$ of states affected by $\boldsymbol{\omega}_k$ grows linearly with $k$ with a factor of $\bar{N}$:*

$$a_k \approx a_{\hat{t}-1} + (k - \hat{t} + 1)\,\bar{N} . \tag{S253}$$

*Proof.* For small $n_t$ with $\frac{c_t n_t}{N_{t-1}} \ll 1$, we have

$$\left(1 - \frac{c_t}{N_{t-1}}\right)^{n_t} \approx 1 - \frac{c_t\,n_t}{N_{t-1}} , \tag{S254}$$

thus

$$n_{t-1} \approx c_t \, n_t \, . \tag{S255}$$

For large $N_{t-1}$ compared to the number of connections $c_t$ of a single state in $s_t$ to states in $s_{t-1}$, we have the approximation

$$\left(1 - \frac{c_t}{N_{t-1}}\right)^{n_t} = \left(\left(1 + \frac{-c_t}{N_{t-1}}\right)^{N_{t-1}}\right)^{n_t/N_{t-1}} \approx \exp(-(c_t \, n_t)/N_{t-1}) \, . \tag{S256}$$

We obtain

$$n_{t-1} = (1 - \exp(-(c_t \, n_t)/N_{t-1})) \, N_{t-1} \, . \tag{S257}$$

For small $n_t$, we again have

$$n_{t-1} \approx c_t \, n_t \, . \tag{S258}$$

Therefore, for small $k - t$, we obtain

$$n_t \approx \prod_{\tau=t}^{k} c_\tau \approx \bar{c}^{k-t} \, . \tag{S259}$$

Thus, for small $k$ the number $a_k$ of states affected by $\boldsymbol{\omega}_k$ is

$$a_k = \sum_{t=0}^{k} n_t \approx \sum_{t=0}^{k} \bar{c}^{k-t} = \sum_{t=0}^{k} \bar{c}^t = \frac{\bar{c}^{k+1} - 1}{\bar{c} - 1} > \bar{c}^k \, . \tag{S260}$$

Consequently, for small $k$ the number $a_k$ of states affected by $\boldsymbol{\omega}_k$ grows exponentially with $k$ by a factor of $\bar{c}$. For large $k$, at a certain time $t > \hat{t}$, $n_t$ has grown such that $c_t n_t > N_{t-1}$, yielding $\exp(-(c_t n_t)/N_{t-1}) \approx 0$, and thus

$$n_t \approx N_t \, . \tag{S261}$$

Therefore

$$a_k - a_{\hat{t}-1} = \sum_{t=\hat{t}}^{k} n_t \approx \sum_{t=\hat{t}}^{k} N_t \approx (k - \hat{t} + 1) \, \bar{N} \, . \tag{S262}$$

Consequently, for large $k$ the number $a_k$ of states affected by $\boldsymbol{\omega}_k$ grows linearly with $k$ by a factor of $\bar{N}$. $\qquad\square$

Therefore, we aim for decreasing the on-site variance $\boldsymbol{\omega}_k$ for large $k$, in order to reduce the variance. In particular, we want to avoid delayed rewards and provide the reward as soon as possible in each episode. Our goal is to give the reward as early as possible in each episode to reduce the variance of action-values that are affected by late rewards and their associated immediate and local variances.

## S4 Experiments

### S4.1 Artificial Tasks

This section provides more details for the artificial tasks (I), (II) and (III) in the main paper. Additionally, we include artificial task (IV) characterized by deterministic reward and state transitions, and artificial task (V) which is solved using policy gradient methods.

#### S4.1.1 Task (I): Grid World

This environment is characterized by probabilistic delayed rewards. It illustrates a situation, where a time bomb explodes at episode end. The agent has to defuse the bomb and then run away as far as possible since defusing fails with a certain probability. Alternatively, the agent can immediately run away, which, however, leads to less reward on average since the bomb always explodes. The Grid World is a quadratic $31 \times 31$ grid with *bomb* at coordinate $[30, 15]$ and *start* at $[30 - d, 15]$, where $d$ is the delay of the task. The agent can move in four different directions (*up*, *right*, *left*, and *down*). Only moves are allowed that keep the agent on the grid. The episode finishes after $1.5d$ steps. At the end of the episode, with a given probability of 0.5, the agent receives a reward of 1000 if it has visited *bomb*. At each time step the agent receives an immediate reward of $c \cdot t \cdot h$, where the factor $c$ depends on the chosen action, $t$ is the current time step, and $h$ is the Hamming distance to *bomb*. Each move of the agent, which reduces the Hamming distance to *bomb*, is penalized by the immediate reward via $c = -0.09$. Each move of the agent, which increases the Hamming distance to *bomb*, is rewarded by the immediate reward via $c = 0.1$. The agent is forced to learn the $Q$-values precisely, since the immediate reward of directly running away hints at a sub-optimal policy.

For non-deterministic reward, the agent receives the delayed reward for having visited *bomb* with probability $p(r_{T+1} = 100 \mid s_T, a_T)$. For non-deterministic transitions, the probability of transiting to next state $s'$ is $p(s' \mid s, a)$. For the deterministic environment these probabilities were either 1 or zero.

**Policy evaluation: learning the action-value estimator for a fixed policy.** First, the theoretical statements on bias and variance of estimating the action-values by TD in Theorem S8 and by MC in Theorem S10 are experimentally verified for a fixed policy. Secondly, we consider the bias and variance of TD and MC estimators of the transformed MDP with optimal reward redistribution according to Theorem S5.

The new MDP with an optimal reward redistribution has advantages over the original MDP both for TD and MC. For TD, the new MDP corrects the bias exponentially faster and for MC it has fewer number of action-values with high variance. Consequently, estimators for the new MDP learn faster than the same estimators in the original MDP.

Since the bias-variance analysis is defined for a particular number of samples drawn from a fixed distribution, we need to fix the policy for sampling. We use an $\epsilon$-greedy version of the optimal policy, where $\epsilon$ is chosen such that on average in 10% of the episodes the agent visits *bomb*. For the analysis, the delay ranges from 5 to 30 in steps of 5. The true $Q$-table for each delay is computed by backward induction and we use 10 different action-value estimators for computing bias and variance.

For the TD update rule we use the exponentially weighted arithmetic mean that is sample-updates, with initial value $q^0(s, a) = 0$. We only monitor the mean and the variance for action-value estimators at the first time step, since we are interested in the time required for correcting the bias. 10 different estimators are run for 10,000 episodes. Figure S3a shows the bias correction for different delays, normalized by the first error.

For the MC update rule we use the arithmetic mean for policy evaluation (later we will use constant-$\alpha$ MC for learning the optimal policy). For each delay, a test set of state-actions for each delay is generated by drawing 5,000 episodes with the $\epsilon$-greedy optimal policy. For each action-value estimator the mean and the variance is monitored every 10 visits. If every action-value has 500 updates (visits), learning is stopped. Bias and variance are computed based on 10 different action-value estimators. As expected from Section S3.1, in Figure S3b the variance decreases by $1/n$, where $n$ is the number of samples. Figure S3b shows that the number of state-actions with a variance larger than a threshold increases exponentially with the delay. This confirms the statements of Theorem S10.

**Learning the optimal policy.** For finding the optimal policy for the Grid World task, we apply Monte Carlo Tree Search (MCTS), $Q$-learning, and Monte Carlo (MC). We train until the greedy policy reaches 90% of the return of the optimal policy. The learning time is measured by the number of episodes. We use *sample updates* for $Q$-learning and MC [78]. For MCTS the greedy policy uses 0 for the exploration constant in UCB1 [33]. The greedy policy is evaluated in 100 episodes

(a)&emsp;&emsp;&emsp;&emsp;&emsp;&emsp;&emsp;&emsp;&emsp;&emsp;&emsp;&emsp;&emsp;&emsp;&emsp;(b)

Figure S3: (a) Experimental evaluation of bias and variance of different $Q$-value estimators on the Grid World. (b) Normalized bias reduction for different delays. Right: Average variance reduction for the 10th highest values.

intervals. The MCTS selection step begins in the start state, which is the root of the game tree that is traversed using UCB1 [33] as the tree policy. If a tree-node gets visited the first time, it is expanded with an initial value obtained by 100 simulated trajectories that start at this node. These simulations use a uniform random policy whose average Return is calculated. The backpropagation step uses the MCTS(1) update rule [32]. The tree policies exploration constant is $\sqrt{2}$. $Q$-learning and MC use a learning rate of $0.3$ and an $\epsilon$-greedy policy with $\epsilon = 0.3$. For RUDDER the optimal reward redistribution using a return decomposition as stated in Section S2.6.1 is used. For each *delay* and each method, 300 runs with different seeds are performed to obtain statistically relevant results.

**Estimation of the median learning time and quantiles.** The performance of different methods is measured by the median learning time in terms of episodes. We stop training at 100 million episodes. Some runs, especially for long delays, have taken too long and have thus been stopped. To resolve this bias the quantiles of the learning time are estimated by fitting a distribution using right censored data [17] .The median is still robustly estimated if more than 50% of runs have finished, which is the case for all plotted datapoints. We find that for delays where all runs have finished the learning time follows a Log-normal distribution. Therefore, we fit a Log-normal distribution on the right censored data. We estimate the median from the existing data, and use maximum likelihood estimation to obtain the second distribution parameter $\sigma^2$. The start value of the $\sigma^2$ estimation is calculated by the measured variance of the existing data which is algebraically transformed to get the $\sigma$ parameter.

### S4.1.2&emsp;Task (II): The Choice

In this experiment we compare RUDDER, temporal difference (TD) and Monte Carlo (MC) in an environment with delayed deterministic reward and probabilistic state transitions to investigate how reward information is transferred back to early states. This environment is a variation of our introductory pocket watch example and reveals problems of TD and MC, while contribution analysis excels. In this environment, only the first action at the very beginning determines the reward at the end of the episode.

**The environment** is an MDP consisting of two actions $a \in \mathcal{A} = \{+, -\}$, an initial state $s^0$, two *charged* states $s^+$, $s^-$, two *neutral* states $s^\oplus$, $s^\ominus$, and a final state $s^f$. After the first action $a_0 \in \mathcal{A} = \{+, -\}$ in state $s^0$, the agent transits to state $s^+$ for action $a_0 = +$ and to $s^-$ for action $a_0 = -$. Subsequent state transitions are probabilistic and independent on actions. With probability $p_C$ the agent stays in the *charged* states $s^+$ or $s^-$, and with probability $(1 - p_C)$ it transits from $s^+$ or $s^-$ to the *neutral* states $s^\oplus$ or $s^\ominus$, respectively. The probability to go from *neutral* states to *charged* states is $p_C$, and the probability to stay in *neutral* states is $(1 - p_C)$. Probabilities to transit from $s^+$

Figure S4: State transition diagram for The Choice task. The diagram is a simplification of the actual MDP.

or $s^{\oplus}$ to $s^-$ or $s^{\ominus}$ or vice versa are zero. Thus, the first action determines whether that agent stays in "+"-states or "−"-states. The reward is determined by how many times the agent visits *charged* states plus a bonus reward depending on the agent's first action. The accumulative reward is given at sequence end and is deterministic. After $T$ time steps, the agent is in the final state $s^{\mathrm{f}}$, in which the reward $R_{T+1}$ is provided. $R_{T+1}$ is the sum of 3 deterministic terms:

1. $R_0$, the baseline reward associated to the first action;
2. $R_C$, the collected reward across states, which depends on the number of visits $n$ to the *charged* states;
3. $R_b$, a bonus if the first action $a_0 = +$.

The expectations of the accumulative rewards for $R_0$ and $R_C$ have the same absolute value but opposite signs, therefore they cancel in expectation over episodes. Thus, the expected return of an episode is the expected reward $R_b$: $p(a_0 = +)b$. The rewards are defined as follows:

$$c_0 = \begin{cases} 1 & \text{if } a_0 = + \\ -1 & \text{if } a_0 = - \,, \end{cases} \tag{S263}$$

$$R_b = \begin{cases} b & \text{if } a_0 = + \\ 0 & \text{if } a_0 = - \,, \end{cases} \tag{S264}$$

$$R_C = c_0 \, C \, n \,, \tag{S265}$$

$$R_0 = - c_0 \, C \, p_C \, T \,, \tag{S266}$$

$$R_{T+1} = R_C + R_0 + R_b \,, \tag{S267}$$

where $C$ is the baseline reward for *charged* states, and $p_C$ the probability of staying in or transiting to *charged* states. The expected visits of charged states is $\mathrm{E}[n] = p_C T$ and $\mathrm{E}[R_{T+1}] = \mathrm{E}[R_b] = p(a_0 = +)b$.

**Methods compared:**    The following methods are compared:

1. $Q$-learning with eligibility traces according to Watkins [87],
2. Monte Carlo,
3. RUDDER with reward redistribution.

For RUDDER, we use an LSTM without lessons buffer and without safe exploration. Contribution analysis is realized by differences of return predictions. For MC, $Q$-values are the exponential moving average of the episode return. For RUDDER, the $Q$-values are estimated by an exponential moving average of the reward redistribution.

**Performance evaluation and results.**    The task is considered as solved when the exponential moving average of the selection of the desired action at time $t = 0$ is equal to $1 - \epsilon$, where $\epsilon$ is the exploration rate. The performances of the compared methods are measured by the average learning time in the number of episodes required to solve the task. A Wilcoxon signed-rank test is performed between the learning time of RUDDER and those of the other methods. Statistical significance p-values are obtained by Wilcoxon signed-rank test. RUDDER with reward redistribution is significantly faster than all other methods with p-values $< 10^{-8}$. Table S1 reports the number of episodes required by different methods to solve the task. RUDDER with reward redistribution clearly outperforms all other methods.

Table S1: Number of episodes required by different methods to solve the grid world task with delayed reward. Numbers give the mean and the standard deviation over 100 trials. RUDDER with reward redistribution clearly outperforms all other TD methods.

| Method | Delay 10 | | | Delay 15 | | | Delay 20 | | |
|---|---|---|---|---|---|---|---|---|---|
| RUDDER | 3520.06 | ± 2343.79 | p = 5.00E-01 | 3062.07 | ± 1278.92 | p = 5.00E-01 | 3813.96 | ± 2738.18 | p = 5.00E-01 |
| MC | 10920.64 | ± 7550.04 | p = 5.03E-24 | 17102.89 | ± 12640.09 | p = 1.98E-30 | 22910.85 | ± 19149.02 | p = 1.25E-28 |
| Q | 66140.76 | ± 1455.33 | p = 1.28E-34 | 115352.25 | ± 1962.20 | p = 1.28E-34 | 171571.94 | ± 2436.25 | p = 1.28E-34 |

| Method | Delay 25 | | | Delay 30 | | | Delay 35 | | |
|---|---|---|---|---|---|---|---|---|---|
| MC | 39772 | ± 47460 | p < 1E-29 | 41922 | ± 36618 | p < 1E-30 | 50464 | ± 60318 | p < 1E-30 |
| Q | 234912 | ± 2673 | p < 1E-33 | 305894 | ± 2928 | p < 1E-33 | 383422 | ± 4346 | p < 1E-22 |
| RUDDER | 4112 | ± 3769 | | 3667 | ± 1776 | | 3850 | ± 2875 | |

| Method | Delay 40 | | | Delay 45 | | | Delay 50 | | |
|---|---|---|---|---|---|---|---|---|---|
| MC | 56945 | ± 54150 | p < 1E-30 | 69845 | ± 79705 | p < 1E-31 | 73243 | ± 70399 | p = 1E-31 |
| Q | 466531 | ± 3515 | p = 1E-22 | | | | | | |
| RUDDER | 3739 | ± 2139 | | 4151 | ± 2583 | | 3884 | ± 2188 | |

| Method | Delay 100 | | | Delay 500 | | |
|---|---|---|---|---|---|---|
| MC | 119568 | ± 110049 | p < 1E-11 | 345533 | ± 320232 | p < 1E-16 |
| RUDDER | 4147 | ± 2392 | | 5769 | ± 4309 | |

### S4.1.3 Task(III): Trace-Back

This section supports the artificial task (III) – **Trace-Back** – in the main paper. RUDDER is compared to potential-based reward shaping methods. In this experiment, we compare reinforcement learning methods that have to transfer back information about a delayed reward. These methods comprise RUDDER, TD($\lambda$) and potential-based reward shaping approaches. For potential-based reward shaping we compare the original *reward shaping* [49], *look-forward advice*, and *look-back advice* [90] with three different potential functions. Methods that transfer back reward information are characterized by low variance estimates of the value function or the action-value function, since they use an estimate of the future return instead of the future return itself. To update the estimates of the future returns, reward information has to be transferred back. The task in this experiment can be solved by Monte Carlo estimates very fast, which do not transfer back information but use samples of the future return for the estimation instead. However, Monte Carlo methods have high variance, which is not considered in this experiment.

**The environment** is a 15×15 grid, where actions move the agent from its current position in 4 adjacent positions (*up, down, left, right*), except the agent would be moved outside the grid. The number of steps (moves) per episode is $T = 20$. The starting position is $(7, 7)$ in the middle of the grid. The maximal return is a combination of negative immediate reward and positive delayed reward. To obtain the maximum return, the policy must move the agent *up* in the time step $t = 1$ and *right* in the following time step $t = 2$. In this case, the agent receives an immediate reward of -50 at $t = 2$ and a delayed reward of 150 at the end of the episode at $t = 20$, that is, a return of 100. Any other combination of actions gives the agent immediate reward of 50 at $t = 2$ without any delayed reward, that is, a return of 50. To ensure Markov properties the position of the agent, the time, as well as the delayed reward are coded in the state. The future reward discount rate $\gamma$ is set to 1. The state transition probabilities are deterministic for the first two moves. For $t > 2$ and for each action, state transition probabilities are equal for each possible next state (uniform distribution), meaning that actions after $t = 2$ do not influence the return. For comparisons of long delays, both the size of the grid and the length of the episode are increased. For a delay of $n$, a $(3n/4) \times (3n/4)$ grid is used with an episode length of $n$, and starting position $(3n/8, 3n/8)$.

**Compared methods.** We compare different TD($\lambda$) and potential-based reward shaping methods. For TD($\lambda$), the baseline is $Q(\lambda)$, with eligibility traces $\lambda = 0.9$ and $\lambda = 0$ and Watkins' implementation [87]. The potential-based reward shaping methods are the original reward shaping, look-ahead advice as well as look-back advice. For look-back advice, we use SARSA($\lambda$) [63] instead of $Q(\lambda)$ as suggested by the authors [90]. $Q$-values are represented by a state-action table, that is, we consider only tabular methods. In all experiments an $\epsilon$-greedy policy with $\epsilon = 0.2$ is used. All three reward shaping methods require a potential function $\phi$, which is based on the reward redistribution ($\tilde{r}_t$) in three different ways:

(I) The Potential function $\phi$ is the difference of LSTM predictions, which is the redistributed reward $R_t$:

$$\phi(s_t) = \mathrm{E}\left[R_{t+1} \mid s_t\right] \quad \text{or} \tag{S268}$$

$$\phi(s_t, a_t) = \mathrm{E}\left[R_{t+1} \mid s_t, a_t\right] . \tag{S269}$$

(II) The potential function $\phi$ is the sum of future redistributed rewards, i.e. the q-value of the redistributed rewards. In the optimal case, this coincides with implementation (I):

$$\phi(s_t) = \mathrm{E}\left[\sum_{\tau=t}^{T} R_{\tau+1} \mid s_t\right] \quad \text{or} \tag{S270}$$

$$\phi(s_t, a_t) = \mathrm{E}\left[\sum_{\tau=t}^{T} R_{\tau+1} \mid s_t, a_t\right] . \tag{S271}$$

(III) The potential function $\phi$ corresponds to the LSTM predictions. In the optimal case this corresponds to the accumulated reward up to $t$ plus the q-value of the delayed MDP:

$$\phi(s_t) = \mathrm{E}\left[\sum_{\tau=0}^{T} \tilde{R}_{\tau+1} \mid s_t\right] \quad \text{or} \tag{S272}$$

$$\phi(s_t, a_t) = \mathrm{E}\left[\sum_{\tau=0}^{T} \tilde{R}_{\tau+1} \mid s_t, a_t\right] . \tag{S273}$$

The following methods are compared:

1. $Q$-learning with eligibility traces according to Watkins ($Q(\lambda)$),

2. SARSA with eligibility traces (SARSA($\lambda$)),

3. Reward Shaping with potential functions (I), (II), or (III) according to $Q$-learning and eligibility traces according to Watkins,

4. Look-ahead advise with potential functions (I), (II), or (III) with $Q(\lambda)$,

5. Look-back advise with potential functions (I), (II), or (III) with SARSA($\lambda$),

6. RUDDER with reward redistribution for $Q$-value estimation and RUDDER applied on top of $Q$-learning.

RUDDER is implemented with an LSTM architecture without output gate nor forget gate. For this experiments, RUDDER does not use lessons buffer nor safe exploration. For contribution analysis we use differences of return predictions. For RUDDER, the $Q$-values are estimated by an exponential moving average (RUDDER $Q$-value estimation) or alternatively by $Q$-learning.

**Performance evaluation:** The task is considered solved when the exponential moving average of the return is above 90, which is 90% of the maximum return. Learning time is the number of episodes required to solve the task. The first evaluation criterion is the average learning time. The $Q$-value differences at time step $t = 2$ are monitored. The $Q$-values at $t = 2$ are the most important ones, since they have to predict whether the maximal return will be received or not. At $t = 2$ the immediate reward acts as a distraction since it is -50 for the action leading to the maximal return ($a^+$) and 50 for all other actions ($a^-$). At the beginning of learning, the $Q$-value difference between $a^+$ and $a^-$ is about -100, since the immediate reward is -50 and 50, respectively. Once the $Q$-values converge to the optimal policy, the difference approaches 50. However, the task will already be correctly solved as soon as this difference is positive. The second evaluation criterion is the $Q$-value differences at time step $t = 2$, since it directly shows to what extend the task is solved.

**Results:** Table S1 reports the number of episodes required by different methods to solve the task. The mean and the standard deviation over 100 trials are given. A Wilcoxon signed-rank test is performed between the learning time of RUDDER and those of the other methods. Statistical significance p-values are obtained by Wilcoxon signed-rank test. RUDDER with reward redistribution is significantly faster than all other methods with p-values $< 10^{-17}$. Tables S2,S3 report the results for all methods.

Table S2: Number of episodes required by different methods to solve the Trace-Back task with delayed reward. The numbers represent the mean and the standard deviation over 100 trials. RUDDER with reward redistribution significantly outperforms all other methods.

| Method | Delay 6 | | | Delay 8 | | | Delay 10 | | |
|---|---|---|---|---|---|---|---|---|---|
| Look-back I | 6074 | ± 952 | p = 1E-22 | 13112 | ± 2024 | p = 1E-22 | 21715 | ± 4323 | p = 1E-06 |
| Look-back II | 4584 | ± 917 | p = 1E-22 | 9897 | ± 2083 | p = 1E-22 | 15973 | ± 4354 | p = 1E-06 |
| Look-back III | 4036.48 | ± 1424.99 | p = 5.28E-17 | 7812.72 | ± 2279.26 | p = 1.09E-23 | 10982.40 | ± 2971.65 | p = 1.03E-07 |
| Look-ahead I | 14469.10 | ± 1520.81 | p = 1.09E-23 | 28559.32 | ± 2104.91 | p = 1.09E-23 | 46650.20 | ± 3035.78 | p = 1.03E-07 |
| Look-ahead II | 12623.42 | ± 1075.25 | p = 1.09E-23 | 24811.62 | ± 1986.30 | p = 1.09E-23 | 43089.00 | ± 2511.18 | p = 1.03E-07 |
| Look-ahead III | 16050.30 | ± 1339.69 | p = 1.09E-23 | 30732.00 | ± 1871.07 | p = 1.09E-23 | 50340.00 | ± 2102.78 | p = 1.03E-07 |
| Reward Shaping I | 14686.12 | ± 1645.02 | p = 1.09E-23 | 28223.94 | ± 3012.81 | p = 1.09E-23 | 46706.50 | ± 3649.57 | p = 1.03E-07 |
| Reward Shaping II | 11397.10 | ± 905.59 | p = 1.09E-23 | 21520.98 | ± 2209.63 | p = 1.09E-23 | 37033.40 | ± 1632.24 | p = 1.03E-07 |
| Reward Shaping III | 12125.48 | ± 1209.59 | p = 1.09E-23 | 23680.98 | ± 1994.07 | p = 1.09E-23 | 40828.70 | ± 2748.82 | p = 1.03E-07 |
| $Q(\lambda)$ | 14719.58 | ± 1728.19 | p = 1.09E-23 | 28518.70 | ± 2148.01 | p = 1.09E-23 | 44017.20 | ± 3170.08 | p = 1.03E-07 |
| SARSA($\lambda$) | 8681.94 | ± 704.02 | p = 1.09E-23 | 23790.40 | ± 836.13 | p = 1.09E-23 | 48157.50 | ± 1378.38 | p = 1.03E-07 |
| RUDDER $Q(\lambda)$ | 726.72 | ± 399.58 | p = 3.49E-04 | 809.86 | ± 472.27 | p = 3.49E-04 | 906.13 | ± 514.55 | p = 3.36E-02 |
| RUDDER | 995.59 | ± 670.31 | p = 5.00E-01 | 1128.82 | ± 741.29 | p = 5.00E-01 | 1186.34 | ± 870.02 | p = 5.00E-01 |

| Method | Delay 12 | | | Delay 15 | | | Delay 17 | | |
|---|---|---|---|---|---|---|---|---|---|
| Look-back I | 33082.56 | ± 7641.57 | p = 1.09E-23 | 49658.86 | ± 8297.85 | p = 1.28E-34 | 72115.16 | ± 21221.78 | p = 1.09E-23 |
| Look-back II | 23240.16 | ± 9060.15 | p = 1.09E-23 | 29293.94 | ± 7468.94 | p = 1.28E-34 | 42639.38 | ± 17178.81 | p = 1.09E-23 |
| Look-back III | 15647.40 | ± 4123.20 | p = 1.09E-23 | 20478.06 | ± 5114.44 | p = 1.28E-34 | 26946.92 | ± 10360.21 | p = 1.09E-23 |
| Look-ahead I | 66769.02 | ± 4333.47 | p = 1.09E-23 | 105336.74 | ± 4977.84 | p = 1.28E-34 | 136660.12 | ± 5688.32 | p = 1.09E-23 |
| Look-ahead II | 62220.56 | ± 3139.87 | p = 1.09E-23 | 100505.05 | ± 4987.16 | p = 1.28E-34 | 130271.88 | ± 5397.61 | p = 1.09E-23 |
| Look-ahead III | 72804.44 | ± 4232.40 | p = 1.09E-23 | 115616.59 | ± 5648.99 | p = 1.28E-34 | 149064.68 | ± 7895.48 | p = 1.09E-23 |
| Reward Shaping I | 68428.04 | ± 3416.12 | p = 1.09E-23 | 107399.17 | ± 5242.88 | p = 1.28E-34 | 137032.14 | ± 6663.12 | p = 1.09E-23 |
| Reward Shaping II | 56225.24 | ± 3778.86 | p = 1.09E-23 | 93091.44 | ± 5233.02 | p = 1.28E-34 | 122224.20 | ± 5545.63 | p = 1.09E-23 |
| Reward Shaping III | 60071.52 | ± 3809.29 | p = 1.09E-23 | 99476.40 | ± 5607.08 | p = 1.28E-34 | 130103.50 | ± 6005.61 | p = 1.09E-23 |
| $Q(\lambda)$ | 66952.16 | ± 4137.67 | p = 1.09E-23 | 107438.36 | ± 5327.95 | p = 1.28E-34 | 135601.26 | ± 6385.76 | p = 1.09E-23 |
| SARSA($\lambda$) | 78306.28 | ± 1813.31 | p = 1.09E-23 | 137561.92 | ± 2350.84 | p = 1.28E-34 | 186679.12 | ± 3146.78 | p = 1.09E-23 |
| RUDDER $Q(\lambda)$ | 1065.16 | ± 661.71 | p = 3.19E-01 | 972.73 | ± 702.92 | p = 1.13E-04 | 1101.24 | ± 765.76 | p = 1.54E-01 |
| RUDDER | 1121.70 | ± 884.35 | p = 5.00E-01 | 1503.08 | ± 1157.04 | p = 5.00E-01 | 1242.88 | ± 1045.15 | p = 5.00E-01 |

Table S3: Cont. Number of episodes required by different methods to solve the Trace-Back task with delayed reward. The numbers represent the mean and the standard deviation over 100 trials. RUDDER with reward redistribution significantly outperforms all other methods.

| Method | Delay 20 | | | Delay 25 | | |
|--------|----------|---|---|----------|---|---|
| Look-back I | 113873.30 | ± 31879.20 | p = 1.03E-07 | | | |
| Look-back II | 56830.30 | ± 19240.04 | p = 1.03E-07 | 111693.34 | ± 73891.21 | p = 1.09E-23 |
| Look-back III | 35852.10 | ± 11193.80 | p = 1.03E-07 | | | |
| Look-ahead I | 187486.50 | ± 5142.87 | p = 1.03E-07 | | | |
| Look-ahead II | 181974.30 | ± 5655.07 | p = 1.03E-07 | 289782.08 | ± 11984.94 | p = 1.09E-23 |
| Look-ahead III | 210029.90 | ± 6589.12 | p = 1.03E-07 | | | |
| Reward Shaping I | 189870.30 | ± 7635.62 | p = 1.03E-07 | 297993.28 | ± 9592.30 | p = 1.09E-23 |
| Reward Shaping II | 170455.30 | ± 6004.24 | p = 1.03E-07 | 274312.10 | ± 8736.80 | p = 1.09E-23 |
| Reward Shaping III | 183592.60 | ± 6882.93 | p = 1.03E-07 | 291810.28 | ± 10114.97 | p = 1.09E-23 |
| $Q(\lambda)$ | 186874.40 | ± 7961.62 | p = 1.03E-07 | | | |
| SARSA$(\lambda)$ | 273060.70 | ± 5458.42 | p = 1.03E-07 | 454031.36 | ± 5258.87 | p = 1.09E-23 |
| RUDDER I | 1048.97 | ± 838.26 | p = 5.00E-01 | 1236.57 | ± 1370.40 | p = 5.00E-01 |
| RUDDER II | 1159.30 | ± 731.46 | p = 8.60E-02 | 1195.75 | ± 859.34 | p = 4.48E-01 |

### S4.1.4 Task (IV): Charge-Discharge

The Charge-Discharge task depicted in Figure S5 is characterized by deterministic reward and state transitions. The environment consists of two states: *charged* C / *discharged* D and two actions *charge* c / *discharge* d. The deterministic reward is $r(\text{D}, \text{d}) = 1, r(\text{C}, \text{d}) = 10, r(\text{D}, \text{c}) = 0$, and $r(\text{C}, \text{c}) = 0$. The reward $r(\text{C}, \text{d})$ is accumulated for the whole episode and given only at time $T + 1$, where $T$ corresponds to the maximal delay of the reward. The optimal policy alternates between charging and discharging to accumulate a reward of 10 every other time step. The smaller immediate reward of 1 distracts the agent from the larger delayed reward. The distraction forces the agent to learn the value function well enough to distinguish between the contribution of the immediate and the delayed reward to the final return.

Figure S5: The Charge-Discharge task with two basic states: *charged* C and *discharged* D. In each state the actions charge c leading to the charged state C and discharge d leading to discharged state D are possible. Action d in the discharged state D leads to a small immediate reward of 1 and in the charged state C to a delayed reward of 10. After sequence end $T = 4$, the accumulated delayed reward $r_{T+1} = r_5$ is given.

For this task, the RUDDER backward analysis is based on monotonic LSTMs and on layer-wise relevance propagation (LRP). The reward redistribution provided by RUDDER uses an LSTM which consists of 5 memory cells and is trained with Adam and a learning rate of $0.01$. The reward redistribution is used to learn an optimal policy by $Q$-learning and by MC with a learning rate of $0.1$ and an exploration rate of $0.1$. Again, we use *sample updates* for $Q$-learning and MC [78]. The learning is stopped either if the agent achieves $90\%$ of the reward of the optimal policy or after a maximum number of 10 million episodes. For each $T$ and each method, 100 runs with different seeds are performed to obtain statistically relevant results. For delays with runs which did not finish within 100m episodes we estimate parameters like described in Paragraph S4.1.1.

### S4.1.5 Task (V): Solving Trace-Back using policy gradient methods

In this experiment, we compare policy gradient methods instead of $Q$-learning based methods. These methods comprise RUDDER on top of PPO with and without GAE, and a baseline PPO using GAE. The environment and performance evaluation are the same as reported in Task III. Again, RUDDER is exponentially faster than PPO. RUDDER on top of PPO is slightly better with GAE than without.

### S4.2 Atari Games

In this section we describe the implementation of RUDDER for Atari games. The implementation is largely based on the *OpenAI baselines* package [13] for the RL components and our package for the LSTM reward redistribution model, which will be announced upon publication. If not specified otherwise, standard input processing, such as skipping 3 frames and stacking 4 frames, is performed by the *OpenAI baselines* package.

We consider the 52 Atari games that were compatible with OpenAI baselines, Arcade Learning Environment (ALE) [6], and OpenAI Gym [10]. Games are divided into episodes, i.e. the loss of a life or the exceeding of 108k frames trigger the start of a new episode without resetting the environment. Source code will be made available at upon publication.

Figure S6: Comparison of performance of RUDDER with GAE (RUDDER+GAE) and without GAE (RUDDER) and PPO with GAE (PPO) on artificial task V with respect to the learning time in episodes (median of 100 trials) in log scale vs. the delay of the reward. The shadow bands indicate the $40\%$ and $60\%$ quantiles. Again, RUDDER significantly outperforms all other methods.

### S4.2.1 Architecture

We use a modified PPO architecture and a separate reward redistribution model. While parts of the two could be combined, this separation allows for better comparison between the PPO baseline with and without RUDDER.

**PPO architecture.** The design of the policy and the value network relies on the *ppo2* implementation [13], which is depicted in Figure S7 and summarized in Table S4. The network input, 4 stacked Atari game frames [44], is processed by 3 convolution layers with ReLU activation functions, followed by a fully connected layer with ReLU activation functions. For PPO with RUDDER 2 output units, for the original and redistributed reward value function, and another set of output units for the policy prediction are applied. For the PPO baseline without RUDDER the output unit for the redistributed reward value function is omitted.

**Reward redistribution model.** Core of the reward redistribution model is an LSTM layer containing 64 memory cells with sigmoid gate activations, tanh input nonlinearities, and identity output activation functions, as illustrated in Figure S7 and summarized in Table S4. This LSTM implementation omits output gate and forget gate to simplify the network dynamics. Identity output activation functions were chosen to support the development of linear counting dynamics within the LSTM layer, as is required to count the reward pieces during an episode chunk. Furthermore, the input gate is only connected recurrently to other LSTM blocks and the cell input is only connected to forward connections from the lower layer. For the vision system the same architecture was used as with the PPO network, with the first convolution layer being doubled to process $\Delta$ frames and full frames separately in the first layer. Additionally, the memory cell layer receives the vision feature activations of the PPO network, the current action, and the approximate in-game time as inputs. No gradients from the reward redistribution network are propagated over the connections to the PPO network. After the LSTM layer, the reward redistribution model has one output node for the prediction $\widehat{\mathbf{g}}$ of the return realization $\mathbf{g}$ of the return variable $G_0$. The reward redistribution model has 4 additional output nodes for the auxiliary tasks as described in Section S4.2.3.

v  π  $\widehat{\mathbf{g}}$

Dense Layer
n=512

LSTM Layer

$a_t$

Conv.Layer2
3x3x64, strides=1

Conv.Layer6
3x3x64, strides=1

Conv.Layer1
4x4x64, strides=2

Conv.Layer5
4x4x64, strides=2

Conv.Layer0
8x8x32, strides=4

Conv.Layer3
8x8x32, strides=4

Conv.Layer4
8x8x32, strides=4

Stacked
Frames

Single
Frame

Delta-
Frame

Figure S7: RUDDER architecture for Atari games as described in Section S4.2.1. Left: The *ppo2* implementation [13]. Right: LSTM reward redistribution architecture. The reward redistribution network has access to the PPO vision features (dashed lines) but no gradient is propagated between the networks. The LSTM layer receives the current action and an approximate in-game-time as additional input. The PPO outputs $v$ for value function prediction and $\pi$ for policy prediction each represent multiple output nodes: the original and redistributed reward value function prediction for $v$ and the outputs for all of the available actions for $\pi$. Likewise, the reward redistribution network output $\widehat{\mathbf{g}}$ represents multiple outputs, as described in Section S4.2.3 Details on layer configuration are given in Table S4.

| Layer | Specifications | | Layer | Specifications | |
|---|---|---|---|---|---|
| Conv.Layer 0 | features | 32 | Conv.Layer 4 | features | 32 |
| | kernelsize | 8x8 | | kernelsize | 8x8 |
| | striding | 4x4 | | striding | 4x4 |
| | act | ReLU | | act | ReLU |
| | initialization | orthogonal, gain=$\sqrt{2}$ | | initialization | orthogonal, gain=0.1 |
| Conv.Layer 1 | features | 64 | Conv.Layer 5 | features | 64 |
| | kernelsize | 4x4 | | kernelsize | 4x4 |
| | striding | 2x2 | | striding | 2x2 |
| | act | ReLU | | act | ReLU |
| | initialization | orthogonal, gain=$\sqrt{2}$ | | initialization | orthogonal, gain=0.1 |
| Conv.Layer 2 | features | 64 | Conv.Layer 6 | features | 64 |
| | kernelsize | 3x3 | | kernelsize | 3x3 |
| | striding | 1x1 | | striding | 1x1 |
| | act | ReLU | | act | ReLU |
| | initialization | orthogonal, gain=$\sqrt{2}$ | | initialization | orthogonal, gain=0.1 |
| Dense Layer | features | 512 | LSTM Layer | cells | 64 |
| | act | ReLU | | gate act. | sigmoid |
| | initialization | orthogonal, gain=$\sqrt{2}$ | | ci act. | tanh |
| Conv.Layer 3 | features | 32 | | output act. | linear |
| | kernelsize | 8x8 | | bias ig | trunc.norm., mean= $-5$ |
| | striding | 4x4 | | bias ci | trunc.norm., mean= 0 |
| | act | ReLU | | fwd.w. ci | trunc.norm., scale= 0.0001 |
| | initialization | orthogonal, gain=0.1 | | fwd.w. ig | omitted |
| | | | | rec.w. ci | omitted |
| | | | | rec.w. ig | trunc.norm., scale= 0.001 |
| | | | | og | omitted |
| | | | | fg | omitted |

Table S4: Specifications of PPO and RUDDER architectures as shown in Figure S7. Truncated normal initialization has the default values mean= 0, stddev= 1 and is optionally multiplied by a factor scale.

### S4.2.2 Lessons Replay Buffer

The lessons replay buffer is realized as a priority-based buffer containing up to 128 samples. New samples are added to the buffer if (i) the buffer is not filled or if (ii) the new sample is considered more important than the least important sample in the buffer, in which case the new sample replaces the least important sample.

Importance of samples for the buffer is determined based on a combined ranking of (i) the reward redistribution model error and (ii) the difference of the sample return to the mean return of all samples in the lessons buffer. Each of these two rankings contributes equally to the final ranking of the sample. Samples with higher loss and greater difference to the mean return achieve a higher ranking.

Sampling from the lessons buffer is performed as a sampling from a softmax function on the sample-losses in the buffer. Each sample is a sequence of 512 consecutive transitions, as described in the last paragraph of Section S4.2.3.

### S4.2.3 Game Processing, Update Design, and Target Design

Reward redistribution is performed in an online fashion as new transitions are sampled from the environment. This allows to keep the original update schema of the PPO baseline, while still using the redistributed reward for the PPO updates. Training of the reward redistribution model is done separately on the lessons buffer samples from Section S4.2.2. These processes are described in more detail in the following paragraphs.

**Reward Scaling.** As described in the main paper, rewards for the PPO baseline and RUDDER are scaled based on the maximum return per sample encountered during training so far. With $i$ samples sampled from the environment and a maximum return of $\mathbf{g}_i^{\max} = \max_{1 \leqslant j \leqslant i}\{|\mathbf{g}_j|\}$ encountered, the scaled reward $r_{\mathrm{new}}$ is

$$r_{\mathrm{new}} = \frac{10\,r}{\mathbf{g}_i^{\max}}. \tag{S274}$$

Goal of this scaling is to normalize the reward $r$ to range $[-10, 10]$ with a linear scaling, suitable for training the PPO and reward redistribution model. Since the scaling is linear, the original proportions between rewards are kept. Downside to this approach is that if a new maximum return is encountered, the scaling factor is updated, and the models have to readjust.

**Reward redistribution.** Reward redistribution is performed using differences of return predictions of the LSTM network. That is, the differences of the reward redistribution model prediction $\widehat{\mathbf{g}}$ at time step $t$ and $t-1$ serve as contribution analysis and thereby give the redistributed reward $r_t = \widehat{\mathbf{g}}_t - \widehat{\mathbf{g}}_{t-1}$. This allows for online reward redistribution on the sampled transitions before they are used to train the PPO network, without waiting for the game sequences to be completed.

To assess the current quality of the reward redistribution model, a quality measure based on the relative absolute error of the prediction $\widehat{\mathbf{g}}_T$ at the last time step $T$ is introduced:

$$\texttt{quality} = 1 - \frac{|\mathbf{g} - \widehat{\mathbf{g}}_T|}{\mu}\,\frac{1}{1-\epsilon}, \tag{S275}$$

with $\epsilon$ as quality threshold of $\epsilon = 80\%$ and the maximum possible error $\mu$ as $\mu = 10$ due to the reward scaling applied. `quality` is furthermore clipped to be within range $[0, 1]$.

**PPO model.** The *ppo2* implementation [13] samples from the environment using multiple agents in parallel. These agents play individual environments but share all weights, i.e. they are distinguished by random effects in the environment or by exploration. The value function and policy network is trained online on a batch of transitions sampled from the environment. Originally, the policy/value function network updates are adjusted using a policy loss, a value function loss, and an entropy term, each with dedicated scaling factors [68]. To decrease the number of hyperparameters, the entropy term scaling factor is adjusted automatically using Proportional Control to keep the policy entropy in a predefined range.

We use two value function output units to predict the value functions of the original and the redistributed reward. For the PPO baseline without RUDDER, the output unit for the redistributed reward is omitted. Analogous to the *ppo2* implementation, these two value function predictions serve to compute the advantages used to scale the policy gradient updates. For this, the advantages for original reward $a_o$ and redistributed reward $a_r$ are combined as a weighted sum $a = a_o\,(1 - \texttt{qualityv}) + a_r\,\texttt{quality}$. The PPO value function loss term $L_v$ is replaced by the sum of the value function $v_o$ loss $L_o$ for the original reward and the scaled value function $v_r$ loss

$L_r$ for the redistributed reward, such that $L_v = L_o + L_r$ `quality`. Parameter values were taken from the original paper [68] and implementation [13]. Additionally, a coarse hyperparameter search was performed with value function coefficients $\{0.1, 1, 10\}$ and replacing the static entropy coefficient by a Proportional Control scaling of the entropy coefficient. The Proportional Control target entropy was linearly decreased from 1 to 0 over the course of training. PPO baseline hyperparamters were used for PPO with RUDDER without changes.
Parameter values are listed in Table S5.

**Reward redistribution model.** The loss of the reward redistribution model for a sample is composed of four parts. (i) The main loss $L_m$, which is the squared prediction loss of $\mathbf{g}$ at the last time step $T$ of the episode

$$L_m = (\mathbf{g} - \widehat{\mathbf{g}}_T)^2 \ , \tag{S276}$$

(ii) the continuous prediction loss $L_c$ of $\mathbf{g}$ at each time step

$$L_c = \frac{1}{T+1} \sum_{t=0}^{T} (\mathbf{g} - \widehat{\mathbf{g}}_t)^2 \ , \tag{S277}$$

(iii) the loss $L_e$ of the prediction of the output at $t + 10$ at each time step $t$

$$L_e = \frac{1}{T-9} \sum_{t=0}^{T-10} \left( \widehat{\mathbf{g}}_{t+10} - \widehat{(\mathbf{g}_{t+10})}_t \right)^2 \ , \tag{S278}$$

as well as (iv) the loss on 3 auxiliary tasks. At every time step $t$, these auxiliary tasks are (1) the prediction of the action-value function $\widehat{q}_t$, (2) the prediction of the accumulated original reward $\tilde{r}$ in the next 10 frames $\sum_{i=t}^{t+10} \tilde{r}_i$, and (3) the prediction of the accumulated reward in the next 50 frames $\sum_{i=t}^{t+50} \tilde{r}_i$, resulting in the final auxiliary loss $L_a$ as

$$L_{a1} = \frac{1}{T+1} \sum_{t=0}^{T} (q_t - \widehat{q}_t)^2 \ , \tag{S279}$$

$$L_{a2} = \frac{1}{T-9} \sum_{t=0}^{T-10} \left( \sum_{i=t}^{t+10} \tilde{r}_i - \left( \widehat{\sum_{i=t}^{t+10} \tilde{r}_i} \right)_t \right)^2 \ , \tag{S280}$$

$$L_{a3} = \frac{1}{T-49} \sum_{t=0}^{T-50} \left( \sum_{i=t}^{t+50} \tilde{r}_i - \left( \widehat{\sum_{i=t}^{t+50} \tilde{r}_i} \right)_t \right)^2 \ , \tag{S281}$$

$$L_a = \frac{1}{3} \left( L_{a1} + L_{a2} + L_{a3} \right) \ . \tag{S282}$$

The final loss for the reward redistribution model is then computed as

$$L = L_m + \frac{1}{10} \left( L_c + L_e + L_a \right) \ . \tag{S283}$$

The continuous prediction and earlier prediction losses $L_c$ and $L_e$ push the reward redistribution model toward performing an optimal reward redistribution. This is because important events that are redundantly encoded in later states are stored as early as possible. Furthermore, the auxiliary loss $L_a$ speeds up learning by adding more information about the original immediate rewards to the updates. The reward redistribution model is only trained on the lessons buffer. Training epochs on the lessons buffer are performed every $10^4$ PPO updates or if a new sample was added to the lessons buffer. For each such training epoch, 8 samples are sampled from the lessons buffer. Training epochs are repeated until the reward redistribution quality is sufficient (`quality > 0`) for all replayed samples in the last 5 training epochs.
The reward redistribution model is not trained or used until the lessons buffer contains at least 32 samples and samples with different return have been encountered.
Parameter values are listed in Table S5.

| PPO | | RUDDER | |
|---|---|---|---|
| learning rate | $2.5 \cdot 10^{-4}$ | learning rate | $10^{-4}$ |
| policy coefficient | 1.0 | $L_2$ weight decay | $10^{-7}$ |
| initial entropy coefficient | 0.01 | gradient clipping | 0.5 |
| value function coefficient | 1.0 | optimization | ADAM |

Table S5: Left: Update parameters for PPO model. Entropy coefficient is scaled via Proportional Control with the target entropy linearly annealed from 1 to 0 over the course of learning. Unless stated otherwise, default parameters of *ppo2* implementation [13] are used. Right: Update parameters for reward redistribution model of RUDDER.

**Sequence chunking and Truncated Backpropagation Through Time (TBPTT).**　Ideally, RUD-DER would be trained on completed game sequences, to consequently redistribute the reward within a completed game. To shorten computational time for learning the reward redistribution model, the model is not trained on completed game sequences but on sequence chunks consisting of 512 time steps. The beginning of such a chunk is treated as beginning of a new episode for the model and ends of episodes within this chunk reset the state of the LSTM, so as to not redistribute rewards between episodes. To allow for updates on sequence chunks even if the game sequence is not completed, the PPO value function prediction is used to estimate the expected future reward at the end of the chunk. Utilizing TBPTT to further speed up LSTM learning, gradients for the reward redistribution LSTM are cut after every 128 time steps.

### S4.2.4　Exploration

Safe exploration to increase the likelihood of observing delayed rewards is an important feature of RUDDER. We use a safe exploration strategy, which is realized by normalizing the output of the policy network to range $[0, 1]$ and randomly picking one of the actions that is above a threshold $\theta$. Safe exploration is activated once per sequence at a random sequence position for a random duration between 0 and the average game length $\bar{l}$. Thereby we encourage long but safe off-policy trajectories within parts of the game sequences. Only 2 of the 8 parallel actors use safe exploration with $\theta_1 = 0.001$ and $\theta_1 = 0.5$, respectively. All actors sample from the softmax policy output.

To avoid policy lag during safe exploration transitions, we use those transitions only to update the reward redistribution model but not the PPO model.

### S4.2.5　Results

Training curves for 3 random seeds for PPO baseline and PPO with RUDDER are shown in Figure S8 and scores are listed in Table S6 for all 52 Atari games. Training was conducted over 200M game frames (including skipped frames), as described in the experiments section of the main paper.

We investigated failures and successes of RUDDER in different Atari games. RUDDER failures were observed to be mostly due to LSTM failures and comprise e.g. slow learning in Breakout, explaining away in Double Dunk, spurious redistributed rewards in Hero, overfitting to the first levels in Qbert, and exploration problems in MontezumaRevenge. RUDDER successes were observed to be mostly due to redistributing rewards to important key actions that would otherwise not receive reward, such as moving towards the built igloo in Frostbite, diving up for refilling oxygen in Seaquest, moving towards the treasure chest in Venture, and shooting at the shield of the enemy boss UFO, thereby removing its shield.

Figure S8: Training curves for PPO baseline and PPO with RUDDER over 200M game frames, 3 runs with different random seeds each. Curves show scores during training of a single agent that does not use safe exploration, smoothed using Locally Weighted Scatterplot Smoothing (y-value estimate using 20% of data with 10 residual-based re-weightings).

|  | *average* | | | *final* | | |
|---|---|---|---|---|---|---|
|  | baseline | RUDDER | % | baseline | RUDDER | % |
| Alien | 1,878 | 3,087 | 64.4 | 3,218 | 5,703 | 77.3 |
| Amidar | 787 | 724 | -8.0 | 1,242 | 1,054 | -15.1 |
| Assault | 5,788 | 4,242 | -26.7 | 10,373 | 11,305 | 9.0 |
| Asterix | 10,554 | 18,054 | 71.1 | 29,513 | 102,930 | 249 |
| Asteroids | 22,065 | 4,905 | -77.8 | 310,505 | 154,479 | -50.2 |
| Atlantis | 1,399,753 | 1,655,464 | 18.3 | 3,568,513 | 3,641,583 | 2.0 |
| BankHeist | 936 | 1,194 | 27.5 | 1,078 | 1,335 | 23.8 |
| BattleZone | 12,870 | 17,023 | 32.3 | 24,667 | 28,067 | 13.8 |
| BeamRider | 2,372 | 4,506 | 89.9 | 3,994 | 6,742 | 68.8 |
| Berzerk | 1,261 | 1,341 | 6.4 | 1,930 | 2,092 | 8.4 |
| Bowling | 61.5 | 179 | 191 | 56.3 | 192 | 241 |
| Boxing | 98.0 | 94.7 | -3.4 | 100 | 99.5 | -0.5 |
| Breakout | 217 | 153 | -29.5 | 430 | 352 | -18.1 |
| Centipede | 25,162 | 23,029 | -8.5 | 53,000 | 36,383 | -31.4 |
| ChopperCommand | 6,183 | 5,244 | -15.2 | 10,817 | 9,573 | -11.5 |
| CrazyClimber | 125,249 | 106,076 | -15.3 | 140,080 | 132,480 | -5.4 |
| DemonAttack | 28,684 | 46,119 | 60.8 | 464,151 | 400,370 | -13.7 |
| DoubleDunk | -9.2 | -13.1 | -41.7 | -0.3 | -5.1 | -1,825 |
| Enduro | 759 | 777 | 2.5 | 2,201 | 1,339 | -39.2 |
| FishingDerby | 19.5 | 11.7 | -39.9 | 52.0 | 36.3 | -30.3 |
| Freeway | 26.7 | 25.4 | -4.8 | 32.0 | 31.4 | -1.9 |
| Frostbite | 3,172 | 4,770 | 50.4 | 5,092 | 7,439 | 46.1 |
| Gopher | 8,126 | 4,090 | -49.7 | 102,916 | 23,367 | -77.3 |
| Gravitar | 1,204 | 1,415 | 17.5 | 1,838 | 2,233 | 21.5 |
| Hero | 22,746 | 12,162 | -46.5 | 32,383 | 15,068 | -53.5 |
| IceHockey | -3.1 | -1.9 | 39.4 | -1.4 | 1.0 | 171 |
| Kangaroo | 2,755 | 9,764 | 254 | 5,360 | 13,500 | 152 |
| Krull | 9,029 | 8,027 | -11.1 | 10,368 | 8,202 | -20.9 |
| KungFuMaster | 49,377 | 51,984 | 5.3 | 66,883 | 78,460 | 17.3 |
| MontezumaRevenge | 0.0 | 0.0 | 38.4 | 0.0 | 0.0 | 0.0 |
| MsPacman | 4,096 | 5,005 | 22.2 | 6,446 | 6,984 | 8.3 |
| NameThisGame | 8,390 | 10,545 | 25.7 | 10,962 | 17,242 | 57.3 |
| Phoenix | 15,013 | 39,247 | 161 | 46,758 | 190,123 | 307 |
| Pitfall | -8.4 | -5.5 | 34.0 | -75.0 | 0.0 | 100 |
| Pong | 19.2 | 18.5 | -3.9 | 21.0 | 21.0 | 0.0 |
| PrivateEye | 102 | 34.1 | -66.4 | 100 | 33.3 | -66.7 |
| Qbert | 12,522 | 8,290 | -33.8 | 28,763 | 16,631 | -42.2 |
| RoadRunner | 20,314 | 27,992 | 37.8 | 35,353 | 36,717 | 3.9 |
| Robotank | 24.9 | 32.7 | 31.3 | 32.2 | 47.3 | 46.9 |
| Seaquest | 1,105 | 2,462 | 123 | 1,616 | 4,770 | 195 |
| Skiing | -29,501 | -29,911 | -1.4 | -29,977 | -29,978 | 0.0 |
| Solaris | 1,393 | 1,918 | 37.7 | 616 | 1,827 | 197 |
| SpaceInvaders | 778 | 1,106 | 42.1 | 1,281 | 1,860 | 45.2 |
| StarGunner | 6,346 | 29,016 | 357 | 18,380 | 62,593 | 241 |
| Tennis | -13.5 | -13.5 | 0.2 | -4.0 | -5.3 | -32.8 |
| TimePilot | 3,790 | 4,208 | 11.0 | 4,533 | 5,563 | 22.7 |
| Tutankham | 123 | 151 | 22.7 | 140 | 163 | 16.3 |
| Venture | 738 | 885 | 20.1 | 820 | 1,350 | 64.6 |
| VideoPinball | 19,738 | 19,196 | -2.7 | 15,248 | 16,836 | 10.4 |
| WizardOfWor | 3,861 | 3,024 | -21.7 | 6,480 | 5,950 | -8.2 |
| YarsRevenge | 46,707 | 60,577 | 29.7 | 109,083 | 178,438 | 63.6 |
| Zaxxon | 6,900 | 7,498 | 8.7 | 12,120 | 10,613 | -12.4 |

Table S6: Scores on all 52 considered Atari games for the PPO baseline and PPO with RUDDER and the improvement by using RUDDER in percent (%). Agents are trained for 200M game frames (including skipped frames) with *no-op starting condition*, i.e. a random number of up to 30 no-operation actions at the start of each game. Episodes are prematurely terminated if a maximum of 108K frames is reached. Scoring metrics are (a) *average*, the average reward per completed game throughout training, which favors fast learning [68] and (b) *final*, the average over the last 10 consecutive games at the end of training, which favors consistency in learning. Scores are shown for one agent without safe exploration.

**Visual Confirmation of Detecting Relevant Events by Reward Redistribution.** We visually confirm a meaningful and helpful redistribution of reward in both Bowling and Venture during training. As illustrated in Figure S9, RUDDER is capable of redistributing a reward to key events in a game, drastically shortening the delay of the reward and quickly steering the agent toward good policies. Furthermore, it enriches sequences that were sparse in reward with a dense reward signal. Video demonstrations are available at `https://goo.gl/EQerZV`.

Figure S9: Observed return decomposition by RUDDER in two Atari games with long delayed rewards. **Left:** In the game Bowling, reward is only given after a turn which consist of multiple rolls. RUDDER identifies the actions that guide the ball in the right direction to hit all pins. Once the ball hit the pins, RUDDER detects the delayed reward associated with striking the pins down. In the figure only 100 frames are represented but the whole turn spans more than 200 frames. In the original game, the reward is given only at the end of the turn. **Right:** In the game Venture, reward is only obtained after picking up the treasure. RUDDER guides the agent (red) towards the treasure (golden) via reward redistribution. Reward is redistributed to entering a room with treasure. Furthermore, the redistributed reward gradually increases as the agent approaches the treasure. The environment only gives reward at the event of collecting the treasure.

## S5    Discussion and Frequent Questions

**RUDDER and reward rescaling.**    RUDDER works with no rescaling, various rescalings, and sign function as we have confirmed in additional experiments. Rescaling ensures similar reward magnitudes across different Atari games, therefore the same hyperparameters can be used for all games. For LSTM and PPO, we only scale the original return by a constant factor, therefore do not change the problem and do not simplify it. The sign function, in contrast, may simplify the problem but may change the optimal policy.

**RUDDER for infinite horizon: Continual Learning.**    RUDDER assumes a finite horizon problem. For games and for most tasks in real world these assumptions apply: "did you solve the task?" (make tax declaration, convince a customer to buy, design a drug, drive a car to a location, assemble a car, build a building, clean the room, cook a meal, pass the Turing test). In general our approach can be extended to continual learning with discounted reward. Only the transformation of an immediate reward MDP to an MDP with episodic reward is no longer possible. However the delayed reward problem becomes more obvious and also more serious when not discounting the reward.

**Is the LSTM in RUDDER a state-action value function?**    For reward redistribution we assume an MDP with one reward (=return) at sequence end which can be predicted from the last state-action pair. When introducing the $\Delta$-states, the reward cannot be predicted from the last $\Delta$ and the task is no longer Markov. However the return can be predicted from the sequence of $\Delta$s. Since the $\Delta$s are mutually independent, the contribution of each $\Delta$ to the return must be stored in the hidden states of the LSTM to predict the final reward. The $\Delta$ can be generic as states and actions can be numbered and then the difference of this numbers can be used for $\Delta$.
In applications like Atari games with immediate rewards we give the accumulated reward at the end of the episode without enriching the states. This has a similar effect as using $\Delta$. We force the LSTM to build up an internal state which tracks the already accumulated reward.
Indeed, the LSTM is the value function at time $t$ based on the $\Delta$ sub-sequence up to t. The LSTM prediction can be decomposed into two sub-predictions. The first sub-prediction is the contribution of the already known $\Delta$ sub-sequence up to t to the return (backward view). The second sub-prediction is the expected contribution of the unknown future sequence from t+1 onwards to the return (forward view). However, we are not interested in the second sub-prediction but only in the contribution of $\Delta_t$ to the prediction of the expected return. The second sub-prediction is irrelevant for our approach. We cancel the second sub-prediction via the differences of predictions. The difference at time t gives the contribution of $\Delta_t$ to the expected return.
Empirical confirmation: Four years ago, we started this research project with using LSTM as a value function but we failed. This was the starting point for RUDDER. In the submission, we used LSTM predictions in artificial task (IV) as potential function for reward shaping, look-ahead advice, and look-back advice. Furthermore, we investigated LSTM as a value function for artificial task (II) but these results have not been included. At the time where RUDDER already solved the task, the LSTM error was too large to allow learning via a value function. Problem is the large variance of the returns at the beginning of the sequence which hinders LSTM learning (forward view). RUDDER LSTM learning was initiated by propagating back prediction errors at the sequence end, where the variance of the return is lower (backward view). These late predictions initiated the storing of key events at the sequence beginning even with high prediction errors. The redistributed reward at the key events led RUDDER solve the task. Concluding: at the time RUDDER solved the task, the early predictions are not learned due to the high variance of the returns. Therefore using the predictions as value function does not help (forward view).
Example: The agent has to take a key to open the door. Since it is an MDP, the agent is always aware to have the key indicated by a key bit to be on. The reward can be predicted in the last step. Using differences $\Delta$ the key bit is zero, except for the step where the agent takes the key. Thus, the LSTM has to store this event and will transfer reward to it.

**Compensation reward.**    The compensation corrects for prediction errors of $g$ ($g$ is the sum of $h$). The prediction error of $g$ can have two sources: (1) the probabilistic nature of the reward, (2) an approximation error of g for the expected reward. We aim to make (2) small and then the correction is only for the probabilistic nature of the reward. The compensation error depends on $g$, which, in turn, depends on the whole sequence. The dependency on state-action pairs from $t = 0$ to $T - 1$ is viewed as random effect, therefore the compensation reward only depends on the last state-action pair.
That $h_t$ and $R_{t+1}$ depends only on $(s_t, a_t, s_{t-1}, a_{t-1})$ is important to prove Theorem 3. Then $a_{t-1}$ cancels and the advantage function remains the same.

**Connection theory and algorithms.** Theorem 1 and Theorem 2 ensure that the algorithms are correct since the optimal policies do not change even for non-optimal return decompositions. In contrast to TD methods which are biased, Theorem 3 shows that the update rule $Q$-value estimation is unbiased when assuming optimal decomposition. Theorem 4 explicitly derives optimality conditions for the expected sum of delayed rewards "kappa" and measures the distance to optimality. This "kappa" is used for learning and is explicitly estimated to correct learning if an optimal decomposition cannot be assured. The theorems are used to justify following learning methods (A) and (B):
(A) Q-value estimation: (i) Direct Q-value estimation (not Q-learning) according to Theorem 3 is given in Eq. (9) when an optimal decomposition is assumed. (ii) Q-value estimation with correction by kappa according to Theorem 4, when optimal decomposition is not assumed. Here kappa is learned by TD as given in Eq. (10). (iii) Q-value estimation using eligibility traces. (B) Policy gradient: Theorems are used as for Q-value estimation as in (A) but now the Q-values serve for policy gradient. (C) Q-learning: Here the properties in Theorem 3 and Theorem 4 are ignored.
We also show variants (not in the main paper) on page 31 and 32 of using kappa "Correction of the reward redistribution" by reward shaping with kappa and "Using kappa as auxiliary task in predicting the return for return decomposition".

**Optimal return decomposition, contributions, and policy.** The Q-value $q^\pi$ depends on a particular policy $\pi$. The function h depends on policy $\pi$ since h predicts the expected return ($E_\pi[\tilde{R}_{T+1}]$) which depends on $\pi$. Thus, both return decomposition and optimal return decomposition are defined for a particular policy $\pi$. A reward redistribution from a return decomposition leads to a return equivalent MDP. Return equivalent MDPs are defined via all policies even if the reward redistribution was derived from a particular policy. A reward redistribution depends only on the state-action sequence but not on the policy that generated this sequence. Also $\Delta$ does not depend on a policy.

**Optimal policies are preserved for every state.** We assume all states are reachable via at least one non-zero transition probability to each state and policies that have a non-zero probability for each action due to exploration. For an MDP being optimal in the initial state is the same as being optimal in every reachable state. This follows from recursively applying the Bellman optimality equation to the initial value function. The values of the following states must be optimal otherwise the initial value function is smaller. Only states to which the transition probability is zero the Bellman optimality equation does not determine the optimality.
All RL algorithms are suitable. For example we applied TD, Monte Carlo, Policy Gradient, which all work faster with the new MDP.

**Limitations.** In all of the experiments reported in this manuscript, we show that RUDDER significantly outperforms other methods for delayed reward problems. However, RUDDER might not be effective when the reward is not delayed since LSTM learning takes extra time and has problems with very long sequences. Furthermore, reward redistribution may introduce disturbing spurious reward signals.

# S6   Additional Related Work

**Delayed Reward.**   To learn delayed rewards there are three phases to consider: (i) discovering the delayed reward, (ii) keeping information about the delayed reward, (iii) learning to receive the delayed reward to secure it for the future. Recent successful reinforcement learning methods provide solutions to one or more of these phases. Most prominent are Deep $Q$-Networks (DQNs) [43, 44], which combine $Q$-learning with convolutional neural networks for visual reinforcement learning [34]. The success of DQNs is attributed to *experience replay* [38], which stores observed state-reward transitions and then samples from them. Prioritized experience replay [64, 27] advanced the sampling from the replay memory. Different policies perform exploration in parallel for the Ape-X DQN and share a prioritized experience replay memory [27]. DQN was extended to double DQN (DDQN) [83, 84] which helps exploration as the overestimation bias is reduced. Noisy DQNs [16] explore by a stochastic layer in the policy network (see [26, 65]). Distributional $Q$-learning [5] profits from noise since means that have high variance are more likely selected. The dueling network architecture [85, 86] separately estimates state values and action advantages, which helps exploration in unknown states. Policy gradient approaches [92] explore via parallel policies, too. A2C has been improved by IMPALA through parallel actors and correction for policy-lags between actors and learners [15]. A3C with asynchronous gradient descent [42] and Ape-X DPG [27] also rely on parallel policies. Proximal policy optimization (PPO) extends A3C by a surrogate objective and a trust region optimization that is realized by clipping or a Kullback-Leibler penalty [68].
Recent approaches aim to solve learning problems caused by delayed rewards. Function approximations of value functions or critics [44, 42] bridge time intervals if states associated with rewards are similar to states that were encountered many steps earlier. For example, assume a function that has learned to predict a large reward at the end of an episode if a state has a particular feature. The function can generalize this correlation to the beginning of an episode and predict already high reward for states possessing the same feature. Multi-step temporal difference (TD) learning [77, 78] improved both DQNs and policy gradients [25, 42]. AlphaGo and AlphaZero learned to play Go and Chess better than human professionals using Monte Carlo Tree Search (MCTS) [69, 70]. MCTS simulates games from a time point until the end of the game or an evaluation point and therefore captures long delayed rewards. Recently, world models using an evolution strategy were successful [21]. These forward view approaches are not feasible in probabilistic environments with a high branching factor of state transition.

**Backward View.**   We propose learning from a backward view, which either learns a separate model or analyzes a forward model. Examples of learning a separate model are to trace back from known goal states [14] or from high reward states [19]. However, learning a backward model is very challenging. When analyzing a forward model that predicts the return then either sensitivity analysis or contribution analysis may be utilized. The best known backward view approach is sensitivity analysis (computing the gradient) like "Backpropagation through a Model" [48, 59, 60, 89, 3]. Sensitivity analysis has several drawbacks: local minima, instabilities, exploding or vanishing gradients, and proper exploration [26, 65]. The major drawback is that the relevance of actions is missed since sensitivity analysis does not consider their contribution to the output but only their effect on the output when slightly perturbing them.
We use contribution analysis since sensitivity analysis has serious drawbacks. Contribution analysis determines how much a state-action pair contributes to the final prediction. To focus on state-actions which are most relevant for learning is known from prioritized sweeping for model-based reinforcement learning [47]. Contribution analysis can be done by computing differences of return predictions when adding another input, by zeroing out an input and then compute the change in the prediction, by contribution-propagation [35], by a contribution approach [54], by excitation backprop [93], by layer-wise relevance propagation (LRP) [1], by Taylor decomposition [1, 45], or by integrated gradients (IG) [76].

**LSTM.**   LSTM was already used in reinforcement learning [66] for advantage learning [2], for constructing a potential function for reward shaping by representing the return by a sum of LSTM outputs across an episode [75], and learning policies [23, 42, 24].

**Reward Shaping, Look-Ahead Advice, Look-Back Advice.**   Redistributing the reward is fundamentally different from reward shaping [49, 90], look-ahead advice, and look-back advice [91]. However, these methods can be viewed as a special case of reward redistribution that result in an MDP that is return-equivalent to the original MDP as is shown in Section S2.2. On the other hand every reward function can be expressed as look-ahead advice [22]. In contrast to these methods,

reward redistribution is not limited to potential functions, where the additional reward is the potential difference, therefore it is a more general concept than shaping reward or look-ahead/look-back advice. The major difference of reward redistribution to reward shaping, look-ahead advice, and look-back advice is that the last three keep the original rewards. Both look-ahead advice and look-back advice have not been designed for replacing for the original rewards. Since the original reward is kept, the reward redistribution is not optimal according to Section S2.6.1. The original rewards may have long delays that cause an exponential slow-down of learning. The added reward improves sampling but a delayed original reward must still be transferred to the $Q$-values of early states that caused the reward. The concept of return-equivalence of SDPs resulting from reward redistributions allows to eliminate the original reward completely. Reward shaping can replace the original reward. However, it only depends on states but not on actions, and therefore, it cannot identify relevant actions without the original reward.

## S7 Reproducibility Checklist

We followed the reproducibility checklist [53] and point to relevant sections.

**For all models and algorithms presented, check if you include:**

- **A clear description of the mathematical setting, algorithm, and/or model.**

  Description of mathematical settings starts at paragraph "MDP Definitions and Return-Equivalent Sequence-Markov Decision Processes (SDPs)".

  Description of novel learning algorithms starts at paragraph "Novel Learning Algorithms Based on Reward Redistributions".

- **An analysis of the complexity (time, space, sample size) of any algorithm.**

  Plots in Figure 1 show the number of episodes, i.e. the sample size, which are needed for convergence to the optimal policies. They are evaluated for different algorithms and delays in all artificial tasks. For Atari games, the number of samples corresponds to the number of game frames. See paragraph "Atari Games". We further present a bias-variance analysis of TD and MC learning in Section S3.1 and Section S3.2 in the Supplements.

- **A link to a downloadable source code, with specification of all dependencies, including external libraries.**

  https://github.com/ml-jku/rudder

**For any theoretical claim, check if you include:**

- **A statement of the result.**

  The main theorems:

  - Theorem 1
  - Theorem 2
  - Theorem 3

  Additional supporting theorems can be found in the Section S2 of the Supplements.

- **A clear explanation of any assumptions.**

  Section S2 in the Supplements covers all the assumptions for the main theorems.

- **A complete proof of the claim.**

  Proof of the main theorems are moved to the Supplements.

  - Proof of Theorem 1 can be found after Theorem S2 in the Supplements.
  - Proof of Theorem 2 can be found after Theorem S4 in the Supplements.
  - Proof of Theorem 3 can be found after Theorem S5 in the Supplements.

  Proofs for additional theorems can also be found in this Supplements.

**For all figures and tables that present empirical results, check if you include:**

- **A complete description of the data collection process, including sample size.**

  For artificial tasks the environment descriptions can be found in the section "Artificial Tasks" in the main paper. For Atari games, we use the standard sampling procedures as in OpenAI Gym [10] (description can be found in paragraph "Atari Games".

- **A link to a downloadable version of the dataset or simulation environment.**

  Link to our repository: https://github.com/ml-jku/rudder

- **An explanation of any data that were excluded, description of any pre-processing step**

  For Atari games, we use the standard pre-processing described in [42].

- **An explanation of how samples were allocated for training / validation / testing.**

  For artificial tasks, description of training and evaluation are included in Section S4.1 in the Supplements. For Atari games, description of training and evaluation are included Section S4.1 in the Supplements.

- **The range of hyper-parameters considered, method to select the best hyper-parameter configuration, and specification of all hyper-parameters used to generate results.**

  A description can be found at paragraph "PPO model" in the Supplements.

- **The exact number of evaluation runs.**

  For artificial tasks evaluation was performed during training runs. See Figure 1. For Atari games see paragraph "Atari Games". We also provide a more detailed description in Section S4.1 and Section S4.2 in the Supplements.

- **A description of how experiments were run**. For artificial task, description can be found at section "Experiments".

  For Atari games, description starts at paragraph "Atari Games". We also provide a more detailed description in Section S4.1 and Section S4.2 in the Supplements.

- **A clear definition of the specific measure or statistics used to report results.**

  For artificial tasks, see Section S4.1. For Atari games, see Section S4.2 and the caption of Table 1. We also provide a more detailed description in Section S4.1 and Section S4.2 in the Supplements.

- **Clearly defined error bars.**

  For artificial tasks, see caption of Figure 1, second line. For Atari games we show all runs in Figure S8 in the Supplements.

- **A description of results with central tendency (e.g. mean) & variation (e.g. stddev).**

  An exhaustive description of the results including mean, variance and significant test, is included in Table S1, Table S2 and Table S3 in Section S4.1 in the Supplements.

- **A description of the computing infrastructure used.**

  We distributed all runs across 2 CPUs per run and 1 GPU per 4 runs for Atari experiments. We used various GPUs including GTX 1080 Ti, TITAN X, and TITAN V. Our algorithm takes approximately 10 days.

## S8   References

[1] S. Bach, A. Binder, G. Montavon, F. Klauschen, K.-R. Müller, and W. Samek. On pixel-wise explanations for non-linear classifier decisions by layer-wise relevance propagation. *PLoS ONE*, 10(7):e0130140, 2015.

[2] B. Bakker. Reinforcement learning with long short-term memory. In T. G. Dietterich, S. Becker, and Z. Ghahramani, editors, *Advances in Neural Information Processing Systems 14*, pages 1475–1482. MIT Press, 2002.

[3] B. Bakker. Reinforcement learning by backpropagation through an lstm model/critic. In *IEEE International Symposium on Approximate Dynamic Programming and Reinforcement Learning*, pages 127–134, 2007.

[4] F. Beleznay, T. Grobler, and C. Szepesvári. Comparing value-function estimation algorithms in undiscounted problems. Technical Report TR-99-02, Mindmaker Ltd., 1999.

[5] M. G. Bellemare, W. Dabney, and R. Munos. A distributional perspective on reinforcement learning. In D. Precup and Y. W. Teh, editors, *Proceedings of the 34th International Conference on Machine Learning*, volume 70 of *Proceedings of Machine Learning Research (ICML)*, pages 449–458. PMLR, 2017.

[6] M. G. Bellemare, Y. Naddaf, J. Veness, and M. Bowling. The Arcade learning environment: An evaluation platform for general agents. *Journal of Artificial Intelligence Research*, 47:253–279, 2013.

[7] D. P. Bertsekas and J. N. Tsitsiklis. An analysis of stochastic shortest path problems. *Math. Oper. Res.*, 16(3), 1991.

[8] I.-J. Bienaymé. Considérations àl'appui de la découverte de laplace. *Comptes Rendus de l'Académie des Sciences*, 37:309–324, 1853.

[9] V. S. Borkar. Stochastic approximation with two time scales. *Systems & Control Letters*, 29(5):291–294, 1997.

[10] G. Brockman, V. Cheung, L. Pettersson, J. Schneider, J. Schulman, J. Tang, and W. Zaremba. Openai gym. *ArXiv*, 1606.01540, 2016.

[11] I. J. Cox, R. Fu, and L. K. Hansen. Probably approximately correct search. In *Advances in Information Retrieval Theory*, pages 2–16. Springer, Berlin, Heidelberg, 2009.

[12] P. Dayan. The convergence of TD($\lambda$) for general $\lambda$. *Machine Learning*, 8:341, 1992.

[13] P. Dhariwal, C. Hesse, O. Klimov, A. Nichol, M. Plappert, A. Radford, J. Schulman, S. Sidor, and Y. Wu. Openai baselines. https://github.com/openai/baselines, 2017.

[14] A. D. Edwards, L. Downs, and J. C. Davidson. Forward-backward reinforcement learning. *ArXiv*, 1803.10227, 2018.

[15] L. Espeholt, H. Soyer, R. Munos, K. Simonyan, V. Mnih, T. Ward, Y. Doron, V. Firoiu, T. Harley, I. Dunning, S. Legg, and K. Kavukcuoglu. IMPALA: Scalable distributed Deep-RL with importance weighted actor-learner architectures. In J. Dy and A. Krause, editors, *Proceedings of the 35th International Conference on Machine Learning*, 2018. ArXiv: 1802.01561.

[16] M. Fortunato, M. G. Azar, B. Piot, J. Menick, I. Osband, A. Graves, V. Mnih, R. Munos, D. Hassabis, O. Pietquin, C. Blundell, and S. Legg. Noisy networks for exploration. *ArXiv*, 1706.10295, 2018. Sixth International Conference on Learning Representations (ICLR).

[17] Irène Gijbels. Censored data. *Wiley Interdisciplinary Reviews: Computational Statistics*, 2(2):178–188, 2010.

[18] R. Givan, T. Dean, and M. Greig. Equivalence notions and model minimization in Markov decision processes. *Artificial Intelligence*, 147(1):163–223, 2003.

[19] A. Goyal, P. Brakel, W. Fedus, T. Lillicrap, S. Levine, H. Larochelle, and Y. Bengio. Recall traces: Backtracking models for efficient reinforcement learning. *ArXiv*, 1804.00379, 2018.

[20] S. Grünewälder and K. Obermayer. The optimal unbiased value estimator and its relation to LSTD, TD and MC. *Machine Learning*, 83(3):289–330, 2011.

[21] D. Ha and J. Schmidhuber. World models. *ArXiv*, 1803.10122, 2018.

[22] A. Harutyunyan, S. Devlin, P. Vrancx, and A. Now'e. Expressing arbitrary reward functions as potential-based advice. In *Proceedings of the Twenty-Ninth AAAI Conference on Artificial Intelligence (AAAI'15)*, pages 2652–2658, 2015.

[23] M. J. Hausknecht and P. Stone. Deep recurrent Q-Learning for partially observable MDPs. *ArXiv*, 1507.06527, 2015.

[24] N. Heess, G. Wayne, Y. Tassa, T. P. Lillicrap, M. A. Riedmiller, and D. Silver. Learning and transfer of modulated locomotor controllers. *ArXiv*, 1610.05182, 2016.

[25] M. Hessel, J. Modayil, H. van Hasselt, T. Schaul, G. Ostrovski, W. Dabney, D. Horgan, B. Piot, M. G. Azar, and D. Silver. Rainbow: Combining improvements in deep reinforcement learning. *ArXiv*, 1710.02298, 2017.

[26] S. Hochreiter. Implementierung und Anwendung eines 'neuronalen' Echtzeit-Lernalgorithmus für reaktive Umgebungen. Practical work, Supervisor: J. Schmidhuber, Institut für Informatik, Technische Universität München, 1990.

[27] D. Horgan, J. Quan, D. Budden, G. Barth-Maron, M. Hessel, H. van Hasselt, and D. Silver. Distributed prioritized experience replay. *ArXiv*, 1803.00933, 2018. Sixth International Conference on Learning Representations (ICLR).

[28] A. Hyvärinen, J. Karhunen, and E. Oja. *Independent Component Analysis*. John Wiley & Sons, New York, 2001.

[29] T. Jaakkola, M. I. Jordan, and S. P. Singh. On the convergence of stochastic iterative dynamic programming algorithms. *Neural Computation*, 6(6):1185–1201, 1994.

[30] G. H. John. When the best move isn't optimal: Q-learning with exploration. In *Proceedings of the 10th Tenth National Conference on Artificial Intelligence, Menlo Park, CA, 1994. AAAI Press.*, page 1464, 1994.

[31] P. Karmakar and S. Bhatnagar. Two time-scale stochastic approximation with controlled Markov noise and off-policy temporal-difference learning. *Mathematics of Operations Research*, 2017.

[32] P. Khandelwal, E. Liebman, S. Niekum, and P. Stone. On the analysis of complex backup strategies in Monte Carlo Tree Search. In *International Conference on Machine Learning*, pages 1319–1328, 2016.

[33] L. Kocsis and C. Szepesvári. Bandit based Monte-Carlo planning. In *European Conference on Machine Learning*, pages 282–293. Springer, 2006.

[34] J. Koutník, G. Cuccu, J. Schmidhuber, and F. Gomez. Evolving large-scale neural networks for vision-based reinforcement learning. In *Proceedings of the 15th Annual Conference on Genetic and Evolutionary Computation*, GECCO '13, pages 1061–1068, 2013.

[35] W. Landecker, M. D. Thomure, L. M. A. Bettencourt, M. Mitchell, G. T. Kenyon, and S. P. Brumby. Interpreting individual classifications of hierarchical networks. In *IEEE Symposium on Computational Intelligence and Data Mining (CIDM)*, pages 32–38, 2013.

[36] T. Lattimore and C. Szepesvá. *Bandit Algorithms*. Cambridge University Press, 2018. Draft of 28th July, Revision 1016.

[37] L. Li, T. J. Walsh, and M. L. Littman. Towards a unified theory of state abstraction for MDPs. In *Ninth International Symposium on Artificial Intelligence and Mathematics (ISAIM)*, 2006.

[38] L. Lin. *Reinforcement Learning for Robots Using Neural Networks*. PhD thesis, Carnegie Mellon University, Pittsburgh, 1993.

[39] G. Lugosi. Concentration-of-measure inequalities. In *Summer School on Machine Learning at the Australian National University,Canberra*, 2003. Lecture notes of 2009.

[40] S. Mannor, D. Simester, P. Sun, and J. N. Tsitsiklis. Bias and variance approximation in value function estimates. *Management Science*, 53(2):308–322, 2007.

[41] V. A. Marǒenko and L. A. Pastur. Distribution of eigenvalues or some sets of random matrices. *Mathematics of the USSR-Sbornik*, 1(4):457, 1967.

[42] V. Mnih, A. P. Badia, M. Mirza, A. Graves, T. Lillicrap, T. Harley, D. Silver, and K. Kavukcuoglu. Asynchronous methods for deep reinforcement learning. In M. F. Balcan and K. Q. Weinberger, editors, *Proceedings of the 33rd International Conference on Machine Learning (ICML)*, volume 48 of *Proceedings of Machine Learning Research*, pages 1928–1937. PMLR, 2016.

[43] V. Mnih, K. Kavukcuoglu, D. Silver, A. Graves, I. Antonoglou, D. Wierstra, and M. A. Riedmiller. Playing Atari with deep reinforcement learning. *ArXiv*, 1312.5602, 2013.

[44] V. Mnih, K. Kavukcuoglu, D. Silver, A. A. Rusu, J. Veness, M. G. Bellemare, A. Graves, M. Riedmiller, A. K. Fidjeland, G. Ostrovski, S. Petersen, C. Beattie, A. Sadik, I. Antonoglou, H. King, D. Kumaran, D. Wierstra, S. Legg, , and D. Hassabis. Human-level control through deep reinforcement learning. *Nature*, 518(7540):529–533, 2015.

[45] G. Montavon, S. Lapuschkin, A. Binder, W. Samek, and K.-R. Müller. Explaining nonlinear classification decisions with deep Taylor decomposition. *Pattern Recognition*, 65:211 – 222, 2017.

[46] G. Montavon, W. Samek, and K.-R. Müller. Methods for interpreting and understanding deep neural networks. *Digital Signal Processing*, 73:1–15, 2017.

[47] A. W. Moore and C. G. Atkeson. Prioritized sweeping: Reinforcement learning with less data and less time. *Machine Learning*, 13(1):103–130, 1993.

[48] P. W. Munro. A dual back-propagation scheme for scalar reinforcement learning. In *Proceedings of the Ninth Annual Conference of the Cognitive Science Society, Seattle, WA*, pages 165–176, 1987.

[49] A. Y. Ng, D. Harada, and S. J. Russell. Policy invariance under reward transformations: Theory and application to reward shaping. In *Proceedings of the Sixteenth International Conference on Machine Learning (ICML'99)*, pages 278–287, 1999.

[50] B. O'Donoghue, I. Osband, R. Munos, and V. Mnih. The uncertainty Bellman equation and exploration. *ArXiv*, 1709.05380, 2017.

[51] S. D. Patek. *Stochastic and shortest path games: theory and algorithms*. PhD thesis, Massachusetts Institute of Technology. Dept. of Electrical Engineering and Computer Science, 1997.

[52] J. Peng and R. J. Williams. Incremental multi-step $Q$-learning. *Machine Learning*, 22(1):283–290, 1996.

[53] J. Pineau. The machine learning reproducibility checklist, 2018.

[54] B. Poulin, R. Eisner, D. Szafron, P. Lu, R. Greiner, D. S. Wishart, A. Fyshe, B. Pearcy, C. MacDonell, and J. Anvik. Visual explanation of evidence in additive classifiers. In *Proceedings of the 18th Conference on Innovative Applications of Artificial Intelligence (IAAI)*, volume 2, pages 1822–1829, 2006.

[55] M. L. Puterman. Markov decision processes. In *Stochastic Models*, volume 2 of *Handbooks in Operations Research and Management Science*, chapter 8, pages 331–434. Elsevier, 1990.

[56] M. L. Puterman. *Markov Decision Processes*. John Wiley & Sons, Inc., 2005.

[57] B. Ravindran and A. G. Barto. Symmetries and model minimization in Markov decision processes. Technical report, University of Massachusetts, Amherst, MA, USA, 2001.

[58] B. Ravindran and A. G. Barto. SMDP homomorphisms: An algebraic approach to abstraction in semi-Markov decision processes. In *Proceedings of the 18th International Joint Conference on Artificial Intelligence (IJCAI'03)*, pages 1011–1016, San Francisco, CA, USA, 2003. Morgan Kaufmann Publishers Inc.

[59] A. J. Robinson. *Dynamic Error Propagation Networks*. PhD thesis, Trinity Hall and Cambridge University Engineering Department, 1989.

[60] T. Robinson and F. Fallside. Dynamic reinforcement driven error propagation networks with application to game playing. In *Proceedings of the 11th Conference of the Cognitive Science Society, Ann Arbor*, pages 836–843, 1989.

[61] J. Romoff, A. Piché, P. Henderson, V. Francois-Lavet, and J. Pineau. Reward estimation for variance reduction in deep reinforcement learning. *ArXiv*, 1805.03359, 2018.

[62] M. Rudelson and R. Vershynin. Non-asymptotic theory of random matrices: extreme singular values. *ArXiv*, 1003.2990, 2010.

[63] G. A. Rummery and M. Niranjan. On-line $Q$-learning using connectionist systems. Technical Report TR 166, Cambridge University Engineering Department, 1994.

[64] T. Schaul, J. Quan, I. Antonoglou, and D. Silver. Prioritized experience replay. *ArXiv*, 1511.05952, 2015.

[65] J. Schmidhuber. Making the world differentiable: On using fully recurrent self-supervised neural networks for dynamic reinforcement learning and planning in non-stationary environments. Technical Report FKI-126-90 (revised), Institut für Informatik, Technische Universität München, 1990. Experiments by Sepp Hochreiter.

[66] J. Schmidhuber. Deep learning in neural networks: An overview. *Neural Networks*, 61:85–117, 2015.

[67] J. Schulman, P. Moritz, S. Levine, M. I. Jordan, and P. Abbeel. High-dimensional continuous control using generalized advantage estimation. *ArXiv*, 1506.02438, 2015. Fourth International Conference on Learning Representations (ICLR'16).

[68] J. Schulman, F. Wolski, P. Dhariwal, A. Radford, and O. Klimov. Proximal policy optimization algorithms. *ArXiv*, 1707.06347, 2018.

[69] D. Silver, A. Huang, C. J. Maddison, A. Guez, L. Sifre, G. van den Driessche, J. Schrittwieser, I. Antonoglou, V. Panneershelvam, M. Lanctot, S. Dieleman, D. Grewe, J. Nham, N. Kalchbrenner, I. Sutskever, T. P. Lillicrap, M. Leach, K. Kavukcuoglu, T. Graepel, and D. Hassabis. Mastering the game of Go with deep neural networks and tree search. *Nature*, 529(7587):484–489, 2016.

[70] D. Silver, T. Hubert, J. Schrittwieser, I. Antonoglou, M. Lai, A. Guez, M. Lanctot, L. Sifre, D. Kumaran, T. Graepel, T. P. Lillicrap, K. Simonyan, and D. Hassabis. Mastering Chess and Shogi by self-play with a general reinforcement learning algorithm. *ArXiv*, 1712.01815, 2017.

[71] S. Singh, T. Jaakkola, M. Littman, and C. Szepesvári. Convergence results for single-step on-policy reinforcement-learning algorithms. *Machine Learning*, 38:287–308, 2000.

[72] S. P. Singh and R. S. Sutton. Reinforcement learning with replacing eligibility traces. *Machine Learning*, 22:123–158, 1996.

[73] M. J. Sobel. The variance of discounted Markov decision processes. *Journal of Applied Probability*, 19(4):794–802, 1982.

[74] A. Soshnikov. A note on universality of the distribution of the largest eigenvalues in certain sample covariance matrices. *J. Statist. Phys.*, 108(5-6):1033–1056, 2002.

[75] P.-H. Su, D. Vandyke, M. Gasic, N. Mrksic, T.-H. Wen, and S. Young. Reward shaping with recurrent neural networks for speeding up on-line policy learning in spoken dialogue systems. In *Proceedings of the 16th Annual Meeting of the Special Interest Group on Discourse and Dialogue*, pages 417–421. Association for Computational Linguistics, 2015.

[76] M. Sundararajan, A. Taly, and Q. Yan. Axiomatic attribution for deep networks. *ArXiv*, 1703.01365, 2017.

[77] R. S. Sutton. Learning to predict by the methods of temporal differences. *Machine Learning*, 3:9–44, 1988.

[78] R. S. Sutton and A. G. Barto. *Reinforcement Learning: An Introduction*. MIT Press, Cambridge, MA, 2 edition, 2018.

[79] A. Tamar, D. DiCastro, and S. Mannor. Policy gradients with variance related risk criteria. In J. Langford and J. Pineau, editors, *Proceedings of the 29th International Conference on Machine Learning (ICML'12)*, 2012.

[80] A. Tamar, D. DiCastro, and S. Mannor. Learning the variance of the reward-to-go. *Journal of Machine Learning Research*, 17(13):1–36, 2016.

[81] P. Tchebichef. Des valeurs moyennes. *Journal de mathématiques pures et appliquées 2*, 12:177–184, 1867.

[82] J. N. Tsitsiklis. Asynchronous stochastic approximation and $Q$-learning. *Machine Learning*, 16(3):185–202, 1994.

[83] H. van Hasselt. Double $Q$-learning. In J. D. Lafferty, C. K. I. Williams, J. Shawe-Taylor, R. S. Zemel, and A. Culotta, editors, *Advances in Neural Information Processing Systems 23*, pages 2613–2621. Curran Associates, Inc., 2010.

[84] H. van Hasselt, A. Guez, and D. Silver. Deep reinforcement learning with double $Q$-learning. In *Proceedings of the Thirtieth AAAI Conference on Artificial Intelligence*, pages 2094–2100. AAAI Press, 2016.

[85] Z. Wang, N. de Freitas, and M. Lanctot. Dueling network architectures for deep reinforcement learning. *ArXiv*, 1511.06581, 2015.

[86] Z. Wang, T. Schaul, M. Hessel, H. Hasselt, M. Lanctot, and N. de Freitas. Dueling network architectures for deep reinforcement learning. In M. F. Balcan and K. Q. Weinberger, editors, *Proceedings of the 33rd International Conference on Machine Learning (ICML)*, volume 48 of *Proceedings of Machine Learning Research*, pages 1995–2003. PMLR, 2016.

[87] C. J. C. H. Watkins. *Learning from Delayed Rewards*. PhD thesis, King's College, 1989.

[88] C. J. C. H. Watkins and P. Dayan. Q-Learning. *Machine Learning*, 8:279–292, 1992.

[89] P. J. Werbos. A menu of designs for reinforcement learning over time. In W. T. Miller, R. S. Sutton, and P. J. Werbos, editors, *Neural Networks for Control*, pages 67–95. MIT Press, Cambridge, MA, USA, 1990.

[90] E. Wiewiora. Potential-based shaping and $Q$-value initialization are equivalent. *Journal of Artificial Intelligence Research*, 19:205–208, 2003.

[91] E. Wiewiora, G. Cottrell, and C. Elkan. Principled methods for advising reinforcement learning agents. In *Proceedings of the Twentieth International Conference on International Conference on Machine Learning (ICML'03)*, pages 792–799, 2003.

[92] R. J. Williams. Simple statistical gradient-following algorithms for connectionist reinforcement learning. *Machine Learning*, 8(3):229–256, 1992.

[93] J. Zhang, Z. L. Lin, J. Brandt, X. Shen, and S. Sclaroff. Top-down neural attention by excitation backprop. In *Proceedings of the 14th European Conference on Computer Vision (ECCV)*, pages 543–559, 2016. part IV.