[Reviews · NeurIPS 2019]

Reviewer 1



4 ) Originality: The work is highly original and I believe it could lead to novel RL algorithms in the future. Quality: 5) The artificial tasks are well-designed to highlight the strength of RUDDER i.e. agent must be able to attribute reward through time in order to solve the task. 6) I would like to see comments comparing RUDDER to generalized advantage estimation (GAE). GAE can also be interpreted as reward shaping. 7) I would recommend using PPO in combination with GAE as a baseline. I don't think it will substantially change the results, but it will be a fairer comparison and the two works are often coupled. Clarity: 8) The paper is clear and the appendix looks exhaustive. I think the experiments could be reproduce using the paper. Conclusion: 9) I think this work is a very valuable contribution to the field. The main caveat for which I do not recommend the acceptance at the moment is that the work omits to mention and use the GAE baseline. I would like to hear the authors comments and would be happy to increase my score if they provide sufficient explanations for not including this baseline.

Reviewer 2



It's an interesting idea supported by a lot of theoretical derivations. Especially on the artificial tasks I could understand that reward redistribution would indeed help the agent learn the task more quickly. On atari games, things are much more complicated, as the scenes are complex and contribution analysis may fail to capture essential scene elements and actions. Though generally positive, I have a few critical comments: 1. The experimental results were not carefully analyzed. Especially since the results of contribution analysis can be visualized easily (e.g. what frames contribute to most changes of predicted rewards), such visualizations would help us identify when RUDDER works and when it fails. 2. Although theoretically the optimal policies should be unchanged after reward redistribution, RUDDER led to worse performance on a significant portion of atari games. Closer analysis is welcome, for example the authors could use the visualization technique suggested above. 3. The authors should discuss more about the limitations of RUDDER. Apparently it's not for all tasks. There's a trade-off between quicker learning of Q values and errors of contribution analysis.

Reviewer 3



POST-REBUTTAL COMMENT: **** Thank you for the extensive feedback, and congrats on really great work. **** Summary This works proposes a method for converting an MDP with delayed rewards into a decision process with only immediate rewards, which makes learning the resulting process easier. The reward redistribution method is proven to preserve optimal policies and reduce the expected future reward to zero. This is achieved by redistributing the delayed rewards to the salient state-action events (where saliency is determined by contribution analysis methods). Extensive experiments in both toy domains, as well as the suite of Atari games, demonstrate the method's improvements for delayed reward tasks, as well as the shortcomings of MC and TD methods for these types of tasks. Comments: I felt the work presented in the paper is outstanding. There are numerous contributions that could conceivably stand on their own (resulting in an extremely large appendix!). The mathematics is rigorous and extremely detailed (which impacts the readability of the main paper somewhat), and there is no handwaving justifications that one might normally find in an ML paper. Aside from the main theoretical work, the experiments are also extensive. I liked the use of toy domains to illustrate the idea and shortcomings of other approaches, as well as the fact that statistical significance tests were employed. Running experiments across 52 Atari games for millions of frames is a significant undertaking, but it is hard to judge the results of those given that RUDDER sometimes was worse than the baseline. However, they do show that for delayed reward games, RUDDER does offer an improved performance, and the Atari experiments confirm that it can be applied to very high-dimensional problems. One concern I have is that from the appendix, it looks like it was a significant engineering feat to implement the method on Atari, which may hinder its adoption by others. I think adoption will likely depend on the quality of code released. Overall, the paper is original, theoretically justified, can scale to large domains and provides a strong foundation for future work. Some additional questions and comments I have for the authors: On Line 82 it is states "we use contribution analysis instead of sensitivity analysis, and (ii) we use the whole state-action sequence to predict its associated return". Point (i) seems to be ill-motivated here. True that contribution analysis a different approach, but it would be helpful to motivate why it is better than sensitivity analysis. Intuitively, if a small perturbation to some input dramatically affects the output, then does it not hold that this input is relevant to the output? If this is not the case, I think at least some explanation would be appropriate to add. Also, the the second point seems to be more of a restriction in pratice, since you need to collect full episodes. I think the paper is significantly hindered by the limit on space. For instance, I think the mean and variance analysis (S3) are significant contributions on their own, but they are primarily buried in the appendix. Space permitting, I think it would be useful to add a bit more discussion on these results in the main paper. Figure S6 is very useful for explanatory purposes and I think the paper would benefit from having it in the main paper (again, space depending). Questions: 1. In the experiments, RUDDER was placed on top of PPO. But PPO is an on-policy method I think. Would the use of the "lessons" buffer not introduce bias? Or does it not matter in practice? 2. After redistributing rewards optimally, am I correct in saying that now there is no bias in TD updates, and the variance in MC is completely minimised? 3. What are the last two columns in Table S6? Minor comments: 1. L43: "proper reward redistribution" is a vague term at that point in the paper. Similarly, L70 talks about "optimal" reward distributions, but this concept is onyl introduced on L133, which is confusing for the reader. 2. The paragraph starting at L285 can probably be merged into the the descriptions of the tasks on page 7 to save space. 3. I do not think the equation in Theorem 3 is referenced, in which case you do not need to have the (2) label (which cannot fit on the line).

[Author Response · NeurIPS 2019]

Thanks to all reviewers for their motivating comments. We appreciate the deep insights of the
reviewers and their constructive suggestions which helped a lot to improve the paper.
**Overall**: **[new figure and animations → R1 10) & R2 2.1, 2.2, 5.1]:** We include a figure similar
to Fig.2 for Atari game Venture and animations that show reward redistributions during the games.
**[analyzing Atari results → R2 2.1, 2.2, 2.3, 5.1 & R3 failure]:** For Atari games, we investigated
failures and successes of RUDDER. We discuss those in the new version. RUDDER failures (LSTM
problems): Breakout (slow learning), DoubleDunk (explaining away), Hero (spurious redistributed
rewards), Qbert (overfits to first levels), MontezumaRevenge (exploration problem). RUDDER
successes (redistributed rewards): Frostbite (moving to igloo), Seaquest (getting oxygen), Venture
(moving to treasure), Phoenix (shooting at boss shield).
**[immediate reward experiment → R3]:** In grid worlds where the agent has to move to a target, no
reward was redistributed with $Q$-value differences as immediate reward. In probabilistic environments
the reward was larger near the target. For delayed reward, positive (negative) rewards are redistributed
to steps toward (away from) the target. Rewards are redistributed to changes in reward expectations.
For example, losing a queen in the game chess receives negative redistributed reward.
**[heavy code → R3]:** A lightweight RUDDER code for the tabular case ($Q$-table) was already
included in the submission. We extended this code to PPO and used it for additional experiments on
Artificial Task (III). All code will be in a github repository.
**[motivation contribution analysis → R3]:** We placed the motivation of contribution analysis in the
appendix (lines 1506-1514). We now also do it in the main paper. Sensitivity analysis determines how
infinitesimal changes of the input lead to infinitesimal changes at the output. We are not interested in
infinitesimal changes of actions or inputs but in large changes. In [1] disadvantages of sensitivity
analysis are pointed out, which are more severe for reward redistribution and lead to spurious rewards.
**R1**: **[6) and 9) RUDDER and GAE]:** GAE fits perfectly with RUDDER. For optimal reward
redistribution GAE is justified by Theorem 3, which states that the advantage function does not
change. For non-optimal reward redistribution, GAE redistributes the reward further back.
**[7) and 9) GAE baseline]:** For Atari, both RUDDER and the baseline already use GAE based on the
OpenAI PPO implementation. We now move the introduction of GAE [2] and TRPO [3] to the main
paper. We performed additional experiments on Artificial Task (III) but with function approximation
to compare PPO+GAE, RUDDER-PPO, and RUDDER-PPO+GAE. Again RUDDER is exponentially
faster than PPO+GAE. RUDDER-PPO+GAE is slightly better than RUDDER-PPO. We report these
experiments in the new version.
**[10) Fig.2]:** Thanks, we will improve the figure accordingly.
**[11) advantage and future zero reward]:** For optimal reward redistribution, learning the advantage
function $a^\pi$ is simplified to estimating the mean of immediate rewards: $a^\pi(s_t, a_t) = r(s_t, a_t) -$
$\sum_a \pi(a \mid s_t)\, r(s_t, a)$. Consequently, the response function $\chi(l; s_t, a_t)$ is zero except for $l = 0$. The
future expected reward can be expressed as $\kappa = \mathrm{E}\left[A_t - R_{t+1} + v(s_t) \mid s_t, a_t\right]$. For GAE$(\gamma, \lambda)$, $\lambda$
determines to what extend the value function replaces the sum of future rewards in $A_t$. $\lambda = 0$ gives
$\kappa = \gamma \mathrm{E}\left[v(s_{t+1}) \mid s_t, a_t\right]$, while $\lambda = 1$ gives $\kappa = \gamma \mathrm{E}\left[\sum_{l=0}^{\infty} \gamma^l R_{t+l+2} \mid s_t, a_t\right]$.
**R2**: The interpretation, visualization, and analysis of the Atari games is of general interest, therefore
we answered them in "Overall" above.
**[RUDDER limitations]:** We will mention that RUDDER is not effective without delayed rewards
since LSTM learning takes extra time and has problems with very long sequences. Furthermore,
reward redistribution may introduce disturbing spurious reward signals.
**R3**: **[bias-variance analysis]:** Absolutely. We will try to elaborate more on the topic.
**[Fig. S6]:** Probably space limits will not allow this.
**[Q1 off-policy]:** The lessons buffer is only used for LSTM training (reward redistribution) but not for
PPO learning. PPO learning is justified with any reward redistribution but if it is on-policy, learning
is more efficient. Still, off-policy lessons buffers are not critical since they contain episodes with
low and high reward. Low reward is less dependent on the policy. High reward episodes are biased
towards the current policy since it has learned to produce them.
**[Q2 TD bias/ MC variance]:** The estimates are unbiased if the future expected rewards are set to
zero like for direct $Q$-value estimation. The variance of MC is in general smaller even if reward
redistribution introduces variance.
**[Tab. S6]:** Sorry. We corrected the wrongly placed table headings.
**[C1 not defined terms]**, **[C2 merging para.]**, **[C3 Eq.(2) in Th. 3]:** Done, thanks!
**[ensure not to fail]:** It is not obvious to us how to ensure that RUDDER improves the baseline.
Thank you very much for the encouraging words. Especially as we worked hard and for a long time
on this project, this is very rewarding.

[1] G. Montavon et al. Methods for interpreting and understanding DNNs. *Digit Signal Process*, 73:1–15, 2017.
[2] J. Schulman et al. High-dim. continuous control using generalized advantage estimation. *ArXiv*, 2015.
[3] J. Schulman et al. Trust region policy optimization. *ArXiv*, 2015.



[Meta-Review · NeurIPS 2019]

The reviewers were unanimous that this paper presents a novel and potentially significant contribution in reward shaping, with thorough theoretical and empirical results. A strong paper overall.